# Loss of Elp3 blocks intestinal tuft cell differentiation via an mTORC1-Atf4 axis

Caroline Wathieu [1,2], Arnaud Lavergne [3], Xinyi Xu[1,2], Marion Rolot[4], Ivan Nemazanyy [5], Kateryna Shostak[1,2], Najla El Hachem [1,6], Chloé Maurizy [1,2], Charlotte Leemans[1,6], Pierre Close[1,6,7], Laurent Nguyen [1,7,8], Christophe Desmet[1,9], Sylvia Tielens[1,2,10], Benjamin G Dewals [4,10] & Alain Chariot [1,2,7,10 ✉]

## Abstract

**Intestinal tuft cells are critical for anti-helminth parasite immunity because they produce IL-25, which triggers IL-13 secretion by activated group 2 innate lymphoid cells (ILC2s) to expand both goblet and tuft cells. We show that epithelial Elp3, a tRNA-modifying enzyme, promotes tuft cell differentiation and is consequently critical for IL-25 production, ILC2 activation, goblet cell expansion and control of *Nippostrongylus brasiliensis* helminth infection in mice. Elp3 is essential for the generation of intestinal immature tuft cells and for the IL-13-dependent induction of glycolytic enzymes such as Hexokinase 1 and Aldolase A. Importantly, loss of epithelial Elp3 in the intestine blocks the codon-dependent translation of the Gator1 subunit Nprl2, an mTORC1 inhibitor, which consequently enhances mTORC1 activation and stabilizes Atf4 in progenitor cells. Likewise, Atf4 overexpression in mouse intestinal epithelium blocks tuft cell differentiation in response to intestinal helminth infection. Collectively, our data define Atf4 as a negative regulator of tuft cells and provide insights into promotion of intestinal type 2 immune response to parasites through tRNA modifications.**

**Keywords** Tuft Cells; tRNA Modifications; mTORC1; ATF4
**Subject Categories** Immunology; Microbiology, Virology & Host Pathogen Interaction; Translation & Protein Quality

## Introduction

Intestinal helminth infections target more than 1.5 billion people worldwide and remain a major health threat (Loukas et al, 2021). The single layer of intestinal epithelial cells (IECs) acts as a protective barrier surface to separate the host from the external environment and is the first line of defense against invading pathogens, including gastrointestinal helminths (Vacca and Le Gros, 2022). The quick turnover (less than a week) and the plasticity of IECs make them suitable to mount a defense against pathogens (Beumer and Clevers, 2021). IECs orchestrate intestinal homeostasis and immune response by sensing nutrients, metabolites and microbiota and by establishing cross-talks with immune cells.

Intestinal tuft cells, a rare population of IECs, derive from Lgr5[+] stem cells and Gfi1b-expressing progenitors and rely on Pou2f3, Sox4 and Stat6 for their development (Gerbe et al, 2016; Gracz et al, 2018; Howitt et al, 2016). Some studies demonstrated that the transcription factor Atoh1 was dispensable for tuft cell differentiation (Bjerknes et al, 2012; Gracz et al, 2018) while another study actually demonstrated that Atoh1 was required in this process (Gerbe et al, 2011). It is now established that tuft cell differentiation occurs through both Atoh1-dependent and independent pathways (Gerbe et al, 2011; Herring et al, 2018). Indeed, tuft cell differentiation is Atoh1-independent in the small intestine but Atoh1-dependent in the colon. Tuft cell markers include Dclk1, acetylated α-tubulin, α-gustducin, Alox5, Ptgs1, Chat, Hopx, and pEgfr on tyrosine 1068 (Gerbe et al, 2012; McKinley et al, 2017; Nadjsombati et al, 2018). Intestinal tuft cells harbor some microvilli into the lumen and exert sensory functions by expressing members of the taste transduction pathway including Trpm5, some G protein-coupled receptors (GPCRs) such as free fatty acid receptor 3 (Ffar3) and bitter taste receptors (type 2 taste receptors T2Rs) (Bezençon et al, 2008, 2007). Tuft cells also sense protozoa (i.e., *Tritrichomonas*) and microbiota-derived metabolites such as succinate through the expression of the succinate receptor SUCNR1 (Lei et al, 2018; Schneider et al, 2018; Nadjsombati et al, 2018). Intestinal tuft cells express the vomeronasal receptor Vmn2r26 to sense the bacterial metabolite *N*-undecanoylglycine (Xiong et al, 2022). Detection of this metabolite by tuft cells initiates a signaling cascade leading to the production of prostaglandin D2 (PGD2),

[1]Interdisciplinary Cluster for Applied Genoproteomics, Liege, Belgium. [2]Laboratory of Cancer Biology, GIGA, University of Liege, Liege, Belgium. [3]GIGA Genomics Platform, University of Liege, Liege, Belgium. [4]Laboratory of Immunology-Vaccinology, Fundamental and Applied Research in Animals and Health (FARAH), University of Liege, Liege, Belgium. [5]Platform for Metabolic Analyses, Structure Fédérative de Recherche Necker, INSERM US24/CNRS UMS 3633, Paris, France. [6]Laboratory of Cancer Signaling, GIGA, University of Liege, Liege, Belgium. [7]WELBIO department, WEL Research Institute, avenue Pasteur, 6, 1300 Wavre, Belgium. [8]Laboratory of Molecular Regulation of Neurogenesis, University of Liege, Liege, Belgium. [9]Laboratory of Cellular and Molecular Immunology, University of Liege, Liege GIGA-I3, Belgium. [10]These authors contributed equally: Sylvia Tielens, Benjamin G Dewals, Alain Chariot. ✉E-mail: Alain.chariot@uliege.be

which triggers mucus production by goblet cells as part of the antibacterial response (Xiong et al, 2022). Intestinal tuft cells also express CD300lf and consequently act as the physiological target of the murine norovirus (MNoV) (Wilen et al, 2018).

Intestinal tuft cells are essential initiators of type 2 immunity by acting as sensory sentinels to mediate the host response against parasitic protozoa and helminths (Gerbe et al, 2016; Von Moltke et al, 2016; Howitt et al, 2016). Upon infection by *Nippostrongylus brasiliensis* or *Heligmosomoides polygyrus*, tuft cells initiate a positive feed-forward immune signaling circuit by first increasing the production of the "alarmin" IL-25, which triggers the activation of group 2 innate lymphoid cells (ILC2s) in the underlying lamina propria (Von Moltke et al, 2016). Activated ILC2s then produce type 2 cytokines IL-5, IL-9, and IL-13. IL-13 critically contributes to tissue remodeling upon helminth infections by acting on stem cells and by ultimately promoting the lineage commitment of undifferentiated epithelial progenitors to both goblet and tuft cells (Zhu et al, 2019). Newly generated tuft cells further activate ILC2s to create the so-called tuft-ILC2 circuit, which ultimately leads to mucus secretion by goblet cells and to helminth expulsion (Von Moltke et al, 2016; Gerbe et al, 2016; Howitt et al, 2016). Of note, cysteinyl leukotrienes, also produced by tuft cells, cooperate with IL-25 to activate ILC2s in order to mount an immune response to helminths but not to intestinal protists, which suggests a context-specific regulation of tuft-ILC2 circuits (McGinty et al, 2020). Although some key progresses have recently been made on the elucidation of cellular events driving the anti-helminth immune response in the intestine, signaling proteins involved in this process only start to be explored. In this context, the tumor suppressor p53 is critical for the immune response to helminth infections by inducing the transcription of Lrmp, a modulator of $Ca^{2+}$ influx and IL-25 release in tuft cells (Chang et al, 2021). IL-13 exerts a dichotomic role as it promotes tuft cell hyperplasia but also induces the bone morphogenic protein (BMP) pathway to repress Sox4 expression and to prevent uncontrolled tuft cell expansion (Lindholm et al, 2022). Sprouty2, an integrator of signaling acting downstream of tyrosine kinase receptors, also negatively regulates colonic tuft cell expansion by limiting IL-13$^+$ stroma cells (Schumacher et al, 2021). Whether other negative cell-intrinsic regulators of tuft cell expansion upon helminth infections exist is currently unknown.

Translational reprogramming is regulated by tRNA modifications and is expected to occur during tuft cell differentiation, even if molecular mechanisms have not been elucidated yet. The tRNA-modifying complex Elongator (Elp1-Elp6), which includes the catalytic subunit Elp3, promotes the formation of 5-methoxycarbonylmethyl (mcm$^5$) or 5-carbamoylmethyl (ncm$^5$) side chains on wobble uridine ($U_{34}$) of the tRNA anticodon (El Yacoubi et al, 2012). Although the role of Elp3 in both tumor initiation and progression of epithelial cancers, as well as in the resistance to targeted therapy in melanoma, is well established, its role in the immune response remains poorly studied (Ladang et al, 2015; Delaunay et al, 2016; Rapino et al, 2018). Mice lacking *Elp3* in IECs have less tuft cells, but underlying mechanisms are unknown (Ladang et al, 2015).

In this study, we report that *Elp3* deficiency in IECs impairs the positive feed-forward type 2 immune signaling circuit triggered upon helminth infection. When bypassing the ILC2 immune signaling by IL-13 treatment, *Elp3* deficiency still specifically

impairs tuft cell differentiation. We also describe a specific signature of glycolytic enzymes robustly induced by IL-13 in an Elp3-dependent manner. Importantly, Single-cell analyses indicate that the loss of *Elp3* potentiates mTORC1 activation, at least because of a defective synthesis of the mTORC1 inhibitor Nprl2. As a consequence, Atf4 levels are increased in multiple epithelial subtypes in the intestine. Likewise, mice overexpressing Atf4 in intestinal epithelium fail to properly respond to helminth infections due to a defective tuft cell amplification. Collectively, our study provides some insights into mechanisms through which some $U_{34}$ tRNA modifications specifically promote tuft cell fate determination and regulate the immune response in the intestine.

# Results

## *Elp3* deficiency impairs intestinal tuft cell amplification upon helminth infection

We previously reported that mice lacking Elp3 in the intestinal epithelium had less tuft cells (Ladang et al, 2015) (see also Fig. 1A). As tuft cell expansion occurs upon helminth infection, we subjected both Elp3$^{WT}$ and Elp3$^{\Delta IEC}$ mice (i.e. mice lacking Elp3 expression in intestinal epithelial cells) to *N. brasiliensis* infection. Elp3 was dispensable for cell proliferation, in both naive and infected mice (Fig. EV1A). Despite similar infection rates in the lung by day 2 post-infection, Elp3$^{\Delta IEC}$ mice showed higher egg numbers from day 6 to day 11 post-infection as well as higher numbers of worms in their intestine, 5 or 7 days post-infection (Fig. 1B–D, respectively). Nevertheless, no eggs could be detected by day 12 in *Elp3*-deficient mice, suggesting effective but delayed worm clearance (Fig. 1C). The parasite viability, measured using an ATP bioluminescence assay, was higher in Elp3$^{\Delta IEC}$ mice at day 7 and 9 post-infection (Fig. 1E). Therefore, intestinal Elp3 is required for the clearance of *N. brasiliensis*.

Tuft cell amplification has been shown to initiate protective type 2 immune responses to helminth infections in the intestine (Gerbe et al, 2016; Von Moltke et al, 2016; Howitt et al, 2016). Therefore, we assessed tuft cell numbers upon infection with *N. brasiliensis* in both Elp3$^{WT}$ and Elp3$^{\Delta IEC}$ mice. *Elp3* deficiency impaired tuft cell amplification post-infection, as evidenced by anti-Dclk1 immunofluorescence (IF) analyses (Fig. 1F). Consistently, both Dclk1 mRNA and protein levels also failed to be properly induced by *N. brasiliensis* upon *Elp3* deficiency (Fig. 1G and H, respectively). Moreover, mRNA levels of a variety of receptors known to be expressed by tuft cells did not increase in IECs lacking Elp3 at day 7 post-infection (Fig. 1I). Likewise, protein levels of FFAR3 also failed to increase in infected Elp3$^{\Delta IEC}$ mice (Fig. EV1B). In contrast, mRNA levels of Defa5 and ChgA, defined as paneth and enteroendocrine cell markers, respectively, did not change in intestines lacking Elp3 (Fig. EV1C,D).

## *Elp3* deficiency impairs all steps of the type 2 immune cascade triggered upon helminth infection in the intestine

IL-25 is mainly expressed by tuft cells in response to helminth infection to activate ILC2s in the intestine. Interestingly, IL-25 was not properly induced upon *N. brasiliensis* infection in IECs lacking

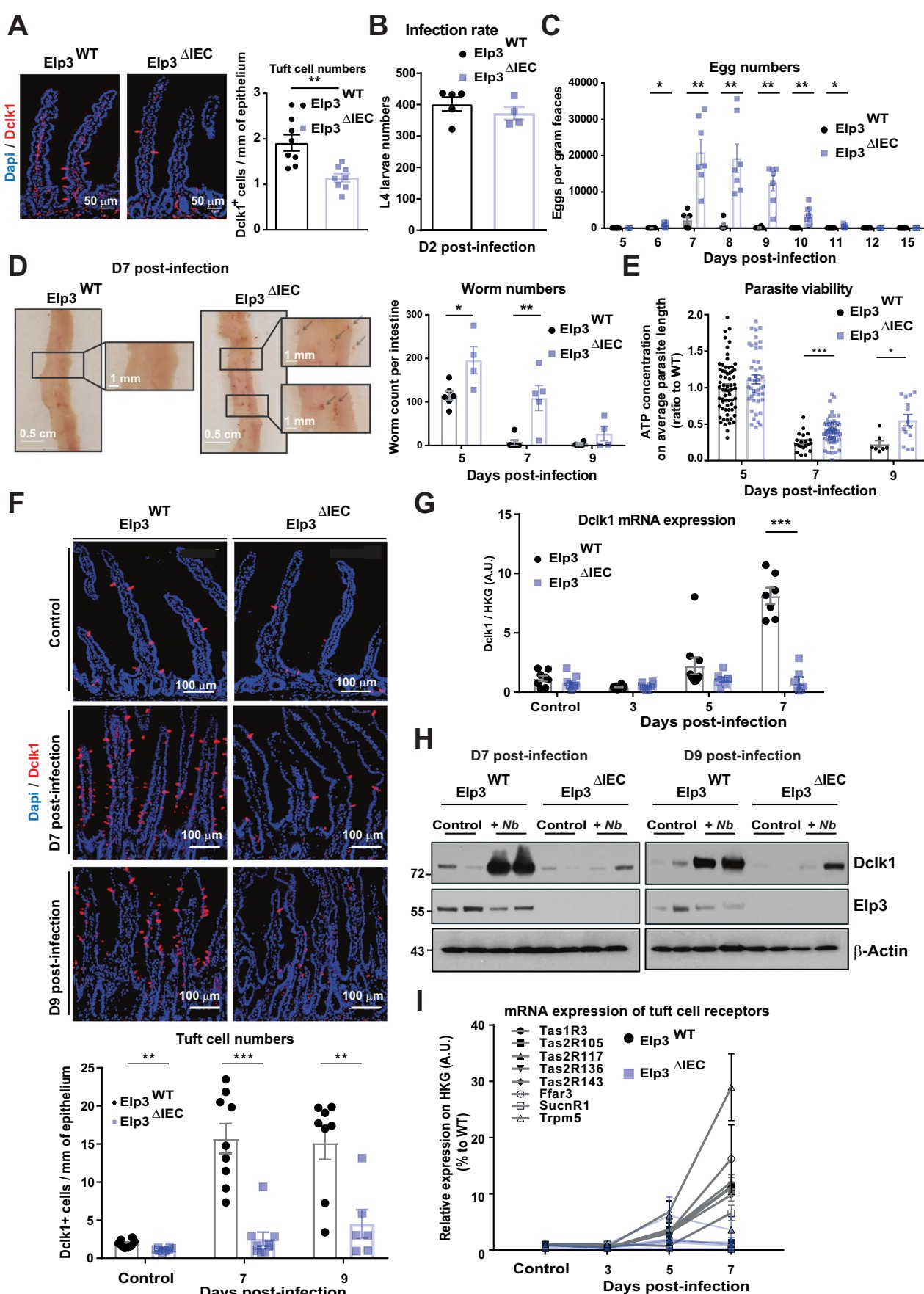

**Figure 1. *Elp3* deficiency impairs intestinal tuft cell amplification upon helminth infection.**

(A) Tuft cell numbers are reduced in intestines lacking Elp3. Immunostainings and corresponding quantifications (left and right panels, respectively) of tuft cells (Dclk1⁺ cells, in red) in the small intestine of naive Elp3^WT and Elp3^ΔIEC mice (mean values ± SEM; Mann–Whitney test; $n = 8$; $**p < 0.01$). (B) *Elp3* deficiency in the intestine does not interfere with the infection rate of *N. brasiliensis* in the lung. The infection rate was assessed by L4 larvae countings at day 2 after *N. brasiliensis* infection in the lungs of both Elp3^WT ($n = 5$) and Elp3^ΔIEC ($n = 4$) mice (mean values ± SEM; Mann–Whitney test). (C, D) Delayed parasite clearance in *Elp3*-deficient mice infected with *N. brasiliensis*. Egg count in faeces from Elp3^WT ($n = 6$) and Elp3^ΔIEC mice ($n = 7$) at indicated time points post-*N. brasiliensis* infection (C) (mean values ± SEM; Mann–Whitney test; $*p < 0.05$, $**p < 0.01$). On the right (D), representative images of proximal small intestines of both genotypes on day 7 post-*N. brasiliensis* infection are shown. On the right (D), worm burden across the entire small intestine at the indicated time points post-*N. brasiliensis* infection are quantified (mean values ± SEM; Mann–Whitney test; $n \geq 4$; $*p < 0.05$, $**p < 0.01$). (E) Parasite viability is enhanced in *Elp3*-deficient mice infected with *N. brasiliensis*. An ATP luciferase assay measuring the parasite viability in Elp3^WT and Elp3^ΔIEC mice at indicated time points post-N. brasiliensis infection is shown (mean values ± SEM; Mann–Whitney test; $n =$ parasites from ≥4 infected mice, $*p < 0.05$, $***p < 0.001$). (F) Tuft cell expansion is impaired in *Elp3*-deficient mice infected with *N. brasiliensis*. Immunostainings and corresponding quantifications (upper and lower panels, respectively) of tuft cells (Dclk1⁺ cells, in red) in the small intestine of Elp3^WT and Elp3^ΔIEC mice naive or at indicated time points post-*N. brasiliensis* infection are shown (mean values ± SEM; Mann–Whitney test; $n \geq 6$; $**p < 0.01$, $***p < 0.001$). Data in naive mice are identical to the ones plotted in panel (A). (G) Impaired induction of Dclk1 mRNA levels in infected intestines lacking Elp3. Real-time PCRs were carried out to quantify Dclk1 mRNA levels with extracts from naive Elp3^WT and Elp3^ΔIEC mice or at indicated time points post-*N. brasiliensis* infection. Normalization was calculated on the average of four housekeeping genes Gapdh, β-Actin, 36b4 and β2-microglobulin (mean values ± SEM; Mann–Whitney test; $n \geq 7$; $***p < 0.001$). (H) Dclk1 protein levels fail to properly increase post-infection in intestines lacking Elp3. Expression levels of Dclk1, Elp3, and β-actin in the small intestine of Elp3^WT and Elp3^ΔIEC mice naive or at indicated time points post-*N. brasiliensis* infection were addressed by western blots. (I) Lack of induction of tuft cell receptors upon infection with *N. brasiliensis* in intestines lacking Elp3. mRNA levels of indicated tuft cell receptors in the small intestine of Elp3^WT and Elp3^ΔIEC mice naive or at indicated time points post-*N. brasiliensis* infection were quantified by real-time PCRs. Normalization was calculated as described in (G). (mean values ± SEM; $n \geq 3$). Source data are available online for this figure.

Elp3 expression, most likely due to a defective tuft cell amplification upon *Elp3* deficiency (Fig. 2A). Reduced IL-25 production further associated with both a significantly reduced IL-13 production by ILC2s in response to *N. brasiliensis* infection in the intestinal lamina propria and the inability of IL-25-induced inflammatory KLRG1⁺ST2⁻ ILC2s (iILC2s) to migrate to the lung of Elp3^ΔIEC mice (Fig. 2B,C) (Huang et al, 2018, 2015). Likewise, mRNA levels of enzymes involved in PGD2 production by tuft cells (Hpgds, Ptgs1/2), were decreased upon *Elp3* deficiency, which ultimately contributes to the impaired PGD2-dependent migration of iILC2s to the lung of Elp3^ΔIEC mice (Fig. 2D) (Xiong et al, 2022; DelGiorno et al, 2020; Wojno et al, 2015). Therefore, *Elp3* deficiency reduces iILC2 activation upon intestinal helminth infection by impairing tuft cell expansion.

Once produced by ILC2s, IL-13 triggers the differentiation of both tuft and goblet cells from Lgr5⁺ stem cells to ultimately expulse the parasites from the intestine (Gerbe et al, 2016; Von Moltke et al, 2016; Howitt et al, 2016). Consistent with an impaired control of worm infection in the intestine of Elp3^ΔIEC mice, mRNA levels of the goblet cell marker Retnlβ did not increase upon infection with *N. brasiliensis* in intestines lacking Elp3 (Fig. 2E) (Herbert et al, 2009). In addition, protein levels of Mucin2 also failed to increase after helminth infection in Elp3^ΔIEC mice, reflecting a lack of amplification of Mucin2⁺ goblet cells (Fig. 2F,G, respectively). Therefore, intestinal *Elp3* deficiency impairs all steps of the local immune response to *N. brasiliensis*, which prevents the proper expulsion of intestinal helminths.

## *Elp3* deficiency impairs IL-13-dependent tuft cell expansion in the intestine

As IL-13 signaling promotes tuft cell differentiation, we assessed the activation of this pathway upon *N. brasiliensis* infection in intestines from both Elp3^WT and Elp3^ΔIEC mice. Jak2 and Stat6 phosphorylation were severely impaired in infected Elp3^ΔIEC mice, most likely due to the decreased number of tuft cells in which Stat6 phosphorylation was detected (Figs. 3A and EV2A, respectively). Of note, IL-13Rα1 but not IL-4Rα mRNA levels were decreased

upon *Elp3* deficiency 7 days post-infection (Fig. 3B). IL-13Rα1 was expressed in intestinal crypts (excluding UEA1⁺ Paneth cells) as well as in Dclk1⁺ tuft cells, as judged by IF analyses (Fig. 3C,D). A defective IL-13-dependent signaling seen upon *Elp3* deficiency can result from impaired levels of IL-13 due to the defective ILC2 expansion. Alternatively, it can be due to an intrinsic defective signaling triggered by IL-13 in *Elp3*-deficient cells. To address this issue, we bypassed ILC2 activation by *N. brasiliensis* and treated both Elp3^WT and Elp3^ΔIEC mice with recombinant IL-13 (rIL-13). IL-13-dependent Dclk1 induction was defective upon *Elp3* deficiency (Fig. 3E). Of note, the expansion of goblet cells upon IL-13 administration, assessed through the quantification of Alcian blue⁺ cells, was similar in intestines from both Elp3^WT and Elp3^ΔIEC mice, indicating that *Elp3* deficiency specifically impairs IL-13-dependent tuft cell differentiation (Fig. EV2B). Importantly, Elp3 expression in IECs was dispensable for the early steps of IL-13 signaling as phosphorylated Stat6 levels after 2 hours of rIL-13 treatment were similar in IECs from both Elp3^WT and Elp3^ΔIEC mice (Fig. 3F). To assess whether Elp3 expression is specifically required in IL-13-dependent tuft cell differentiation, we treated both Elp3^WT and Elp3^ΔIEC mice with Dibenzazepine (DBZ), a γ-Secretase and Notch inhibitor which promotes the differentiation of all types of secretory cells, including goblet, Paneth, tuft and enteroendocrine cells in intestinal epitheliums (van Es et al, 2005; VanDussen et al, 2012; Von Moltke et al, 2016). Elp3 was also required for the differentiation of tuft cells seen upon Notch inhibition, as Dclk1 was not properly induced by DBZ upon *Elp3* deficiency (Fig. 3G). Likewise, DBZ did not properly increase the number of Dclk1⁺ cells upon *Elp3* deficiency (Fig. 3H). On the other hand, levels of Chromogranin A (ChgA), an enteroendocrine cell marker, were similarly induced in IECs of both Elp3^WT and Elp3^ΔIEC mice treated with DBZ, reflecting a proper amplification of ChgA⁺ enteroendocrine cells (Figs. 3G and EV2C, respectively). Moreover, DBZ also similarly induced the number of goblet cells (Alcian blue⁺ cells) in the intestines of both Elp3^WT and Elp3^ΔIEC mice, indicating that Elp3 is dispensable for goblet cell differentiation (Fig. EV2D). Collectively, our results demonstrate that Elp3 is specifically required to transmit signals causing the differentiation of intestinal tuft cells.

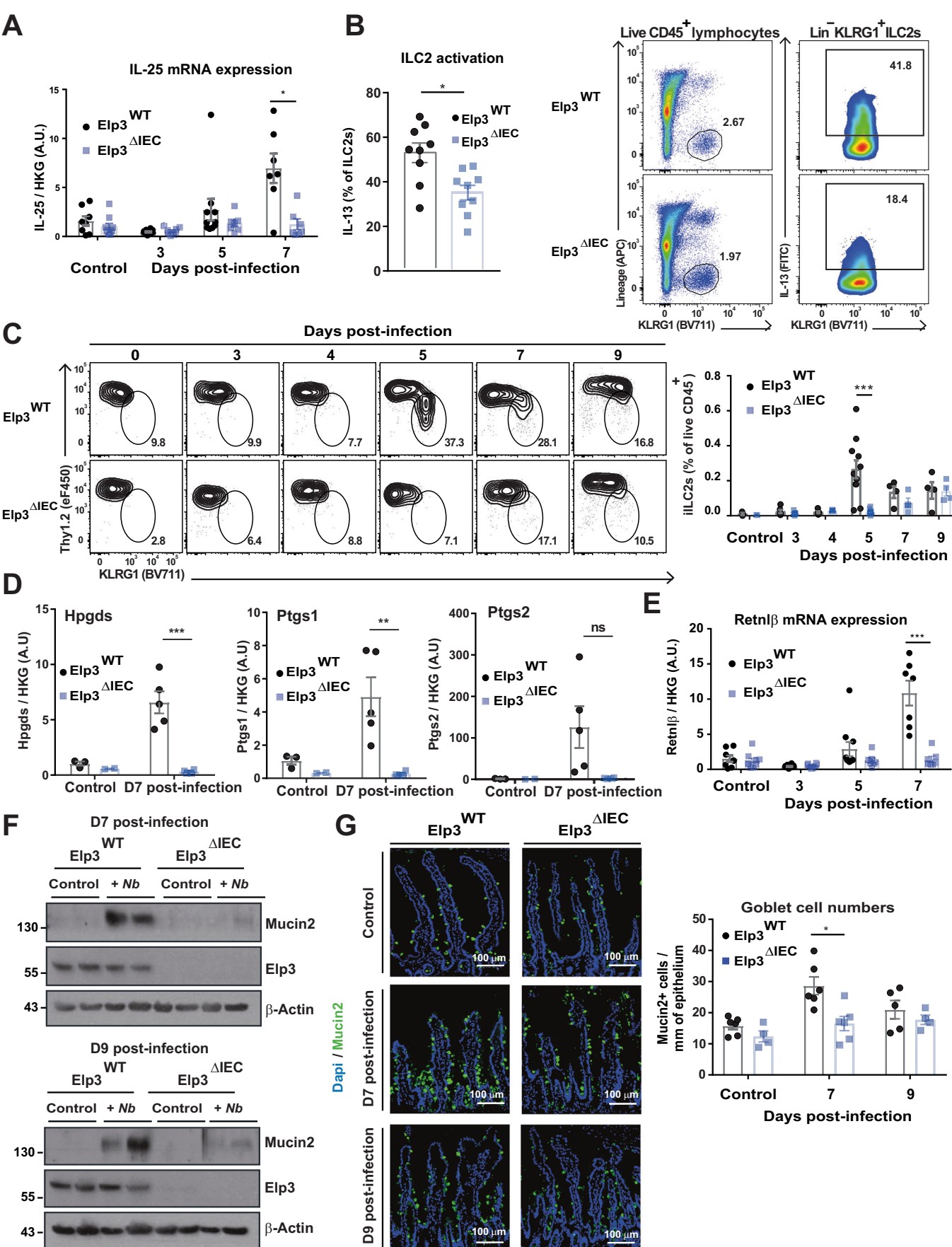

**Figure 2.** *Elp3* **deficiency impairs ILC2 activation and goblet cell expansion upon helminth infection in the intestine.**

(A) IL-25 induction upon helminth infection is impaired in IECs lacking Elp3. IL-25 mRNA levels in the small intestine of Elp3$^{WT}$ and Elp3$^{\Delta IEC}$ mice naive or at indicated time points post-*N. brasiliensis* infection were quantified by real-time PCRs. Normalization was calculated as described in Fig. 1G (mean values ± SEM; Mann–Whitney test; $n \geq 7$; *$p < 0.05$). (B) IL-13 production by intestinal ILC2s is impaired upon *Elp3* deficiency at day 3 after *N. brasiliensis* infection. Lamina propria cells were isolated and subjected to PMA and Ionomycin restimulation in the presence of brefeldin and monsensin before intracellular staining for IL-13. ILC2s were gated as live CD45$^+$Lin$^-$KLRG1$^+$ cells. Lineage included APC-labeled antibodies to CD3ε, CD4, CD8α, CD11c, siglec-F, FcεRI, B220/CD45R, Gr-1, CD5, CD49b, and F4/80 (mean values ± SEM; Mann–Whitney test; $n \geq 9$; *$p < 0.05$). (C) Impaired migration of inflammatory ILC2s (iILC2s) in the lung over time after *N. brasiliensis* infection in Elp3$^{\Delta IEC}$ mice. Total ILC2s were gated as Lin-Gata3$^+$Sca1$^+$ and iILC2s were identified as Klrg1$^{high}$Thy-1$^{low}$ cells (mean values ± SEM; Mann–Whitney test; $n \geq 3$; ***$p < 0.001$). (D) Impaired expression of enzymes involved in PGD2 production in Elp3$^{\Delta IEC}$ mice. mRNA levels of the indicated enzymes in the small intestine of Elp3$^{WT}$ and Elp3$^{\Delta IEC}$ mice naive or 7 days post-*N. brasiliensis* infection were quantified by real-time PCRs. Normalization was calculated on the average of two housekeeping genes Gapdh and 36b4 (mean values ± SEM; Student $t$-test; $n = 3$ and 2 for naive Elp3$^{WT}$ and Elp3$^{\Delta IEC}$ mice; $n = 5$ and 4 for infected Elp3$^{WT}$ and Elp3$^{\Delta IEC}$ mice, **$p < 0.01$, ***$p < 0.001$). (E–G) Goblet cell expansion is impaired in *Elp3*-deficient mice infected with *N. brasiliensis*. Retnlβ mRNA levels in the small intestine of Elp3$^{WT}$ and Elp3$^{\Delta IEC}$ mice naive or at indicated time points post-*N. brasiliensis* infection were quantified (E). Normalization was calculated as described in Fig. 1G (mean values ± SEM; Mann–Whitney test; $n \geq 7$; ***$p < 0.001$). Protein levels of Mucin2 were also quantified by western blot analyses with extracts from the small intestine of Elp3$^{WT}$ and Elp3$^{\Delta IEC}$ mice naive or at indicated time points post-*N. brasiliensis* infection (F). Mucin2$^+$ cells (in green) were also quantified by IF in the small intestine of Elp3$^{WT}$ and Elp3$^{\Delta IEC}$ mice naive or infected with *N. brasiliensis* (mean values ± SEM; Mann–Whitney test; $n \geq 4$; *$p < 0.05$) (G). Source data are available online for this figure.

## *Elp3* deficiency impairs the expression of candidates enriched in immature tuft cells upon IL-13 stimulation

To learn more about the mechanisms through which Elp3 promotes tuft cell differentiation in the intestine, we performed single-cell transcriptomic analyses on IECs from both Elp3$^{WT}$ and Elp3$^{\Delta IEC}$ mice treated with rIL-13 to trigger both tuft and goblet cell differentiation. Transcriptional signatures from all epithelial subtypes found in intestinal epitheliums were identified (Fig. EV3A–C). Unsupervised graph clustering defined 11 distinct populations, which we labeled using known intestinal cell-type markers (Fig. EV3A; Table EV1). For both tuft and goblet cells, we distinguished immature from mature populations. Cells from immature populations share markers with non-differentiated cells (transit-amplifying cells (TA) and enterocyte progenitors) (Fig. EV3C). These cells also expressed some specific tuft/goblet cell markers, although at lower levels than fully differentiated cells (Fig. EV3C). As expected from our initial observation, the signature of tuft, but not other epithelial cell types, was impaired upon *Elp3* deficiency (Fig. 4A,B). Of note, Elp3 expression was mainly detected in both immature tuft and goblet cells as well as in enterocyte progenitors and transit-amplifying cells, as judged by our single-cell RNA sequencing data obtained with extracts of IECs from Elp3$^{WT}$ mice treated with rIL-13 (Fig. 4C). Importantly, we could also establish a transcriptional signature of candidates specifically enriched in immature tuft cells from the intestine of IL-13-stimulated Elp3$^{WT}$ mice (Fig. 4D). The expression of these candidates was severely impaired in Elp3$^{\Delta IEC}$ mice, 7 days post-infection with *N. brasiliensis* (Fig. 4E). Therefore, *Elp3* deficiency in IECs blocks intestinal tuft cell fate at an early stage of differentiation.

## *Elp3* deficiency impairs both oxidative phosphorylation and glycolysis in tuft cells

To further explore molecular mechanisms through which Elp3 promotes intestinal tuft cell differentiation, we carried out a gene set enrichment analysis (GSEA) and found that two metabolic pathways, namely oxidative phosphorylation and glycolysis, were defective in tuft cells from Elp3$^{\Delta IEC}$ mice (Fig. 5A). Interestingly, multiple candidates involved in both metabolic pathways were actually enriched in tuft cells from Elp3$^{WT}$ mice compared to other secretory cells, but this enrichment was less pronounced upon *Elp3* deficiency (Fig. 5B,C). Importantly, some enzymes involved in glycolysis, namely Hexokinase 1 (HK1) and Aldolase A (Aldoa) were not properly induced in IL-13-stimulated Elp3$^{\Delta IEC}$ mice (Fig. 5D). On the other hand, other enzymes involved in glycolysis, namely Phosphoglycerate kinase 1 (Pgk1), Phosphoglycerate mutase 1 (Pgam1) and Enolase 1/2 (Eno1/2) were not induced by IL-13 and did not rely on Elp3 to be properly expressed (Fig. 5D). Therefore, specific glycolytic enzymes enriched in tuft cells are robustly induced by IL-13 in an Elp3-dependent manner.

## *Elp3* deficiency in the intestine potentiates an mTORC1-Atf4 signature in immature cells

To better define signaling pathways that control intestinal tuft cell differentiation, we conducted an additional GSEA with our single-cell RNA analyses and noticed that IL-13-stimulated immature cells lacking Elp3 showed enhanced UPR and mTORC1 signaling signatures (Fig. 6A,B). Indeed, an UPR signaling signature was enhanced in TA cells as well as in enterocyte progenitors and immature tuft cells but not in stem cells lacking Elp3 (Fig. 6A,B). Interestingly, mTORC1 signaling was also potentiated in TA cells and in immature goblet cells but not in stem cells lacking Elp3 (Fig. 6A,B). Likewise, the expression of Atf4, an effector of both UPR and mTORC1 signaling (Torrence et al, 2021), was potentiated upon *Elp3* deficiency in both immature tuft and goblet cells as well as in TA cells and enterocyte progenitors subjected to rIL-13 stimulation (Fig. 6C). Importantly, *Elp3* deficiency did not trigger the canonical UPR as phosphorylated levels of both IRE1α and eIF2α, as well as levels of Xbp1s, Bip, and Chop, remained unchanged (Fig. 6D). Elevated Atf4 levels as well as enhanced S6 phosphorylation, a hallmark of mTORC1 activation, were confirmed through western blot analyses in the intestinal epithelium from Elp3$^{\Delta IEC}$ mice (Fig. 6E). Collectively, our data demonstrate that loss of *Elp3* potentiates mTORC1 activation and enhances Atf4 levels.

Atf4 is an effector of mTORC1 and a specific transcriptional signature can be defined downstream of this pathway (Torrence et al, 2021). Interestingly, this subset of Atf4 target genes was found to be elevated in TA cells, enterocyte progenitors, and immature tuft/goblet cells lacking Elp3, indicating that the mTORC1/Atf4 pathway is indeed potentiated in these cells (Fig. 6F,G). Enhanced

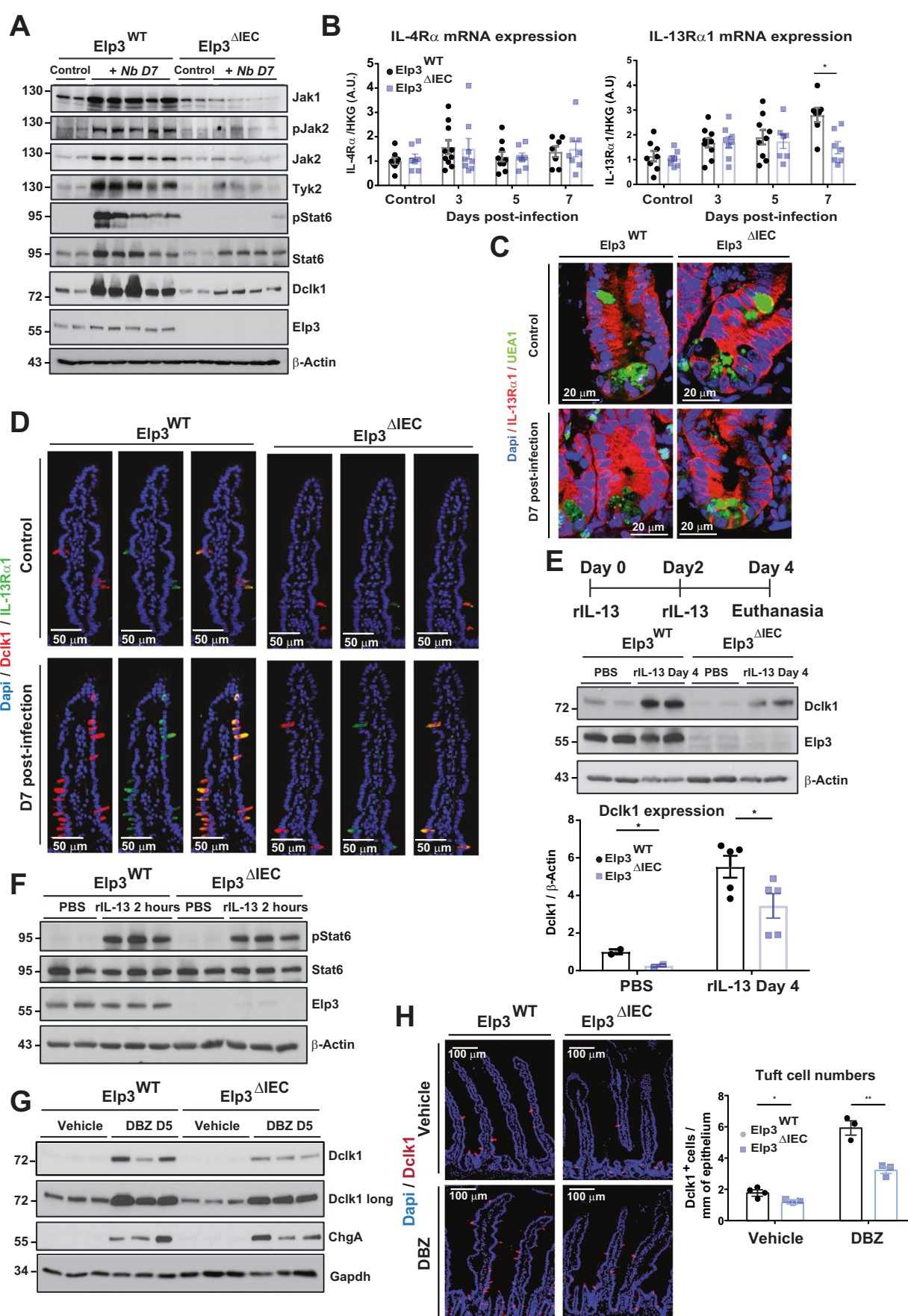

**Figure 3.  *Elp3* deficiency impairs tuft cell differentiation upon IL-13 stimulation or Notch inhibition.**

(A) IL-13 signaling is impaired in the intestine of Elp3$^{\Delta IEC}$ mice upon infection with *N. brasiliensis*. Expression levels of IL-13 signaling effectors, Dclk1, Elp3, and β-Actin in the small intestine of Elp3$^{WT}$ and Elp3$^{\Delta IEC}$ mice naive or at D7 post-*N. brasiliensis* infection were assessed by western blot. (B) Impaired expression of IL-13Rα1 but not IL-4Rα in infected intestines lacking Elp3. IL-13Rα1 and IL-4Rα mRNA levels in the indicated experimental conditions were assessed by real-time PCRs. Data were plotted as described in Fig. 1G (mean values ± SEM; Mann–Whitney test; $n \geq 7$; *$p < 0.05$). (C, D) IL-13Rα1 expression is detected in intestinal crypts (C) and in tuft cells (D). Representative pictures of IL-13Rα1$^+$ cells (in red) by immunostaining in the small intestine of Elp3$^{WT}$ and Elp3$^{\Delta IEC}$ mice naive or infected with *N. brasiliensis* are shown (C). Paneth cells were defined as UEA1$^+$ cells (in green). Representative pictures of IL-13Rα1$^+$/Dclk1$^+$ cells immunostaining (in green and in red, respectively) in the small intestine of Elp3$^{WT}$ and Elp3$^{\Delta IEC}$ mice naive or infected with *N. brasiliensis* are shown (D). (E) Tuft cell differentiation is specifically impaired in intestines lacking Elp3 upon rIL-13 treatment. Elp3$^{WT}$ and Elp3$^{\Delta IEC}$ mice received or not rIL-13 injections every 2 days to promote both tuft and goblet cell differentiation. The resulting extracts from intestines were subjected to western blot analyses. A quantification of the Dclk1/β-Actin ratio in all experimental conditions (mean values ± SEM; Student *t*-test; $n = 2$ for naive mice and $n = 5$ for rIL-13 treated mice; *$p < 0.05$) is shown. (F) IL-13-dependent Stat6 activation is not affected in IECs lacking Elp3 upon rIL-13 treatment. Expression levels of pStat6, Stat6, Elp3, and β-Actin in the small intestine of naive Elp3$^{WT}$ and Elp3$^{\Delta IEC}$ mice or subjected to rIL-13 treatment were assessed by western blot. (G, H) Tuft cell differentiation is specifically impaired in intestines lacking Elp3 upon Notch inhibition. Expression levels of Dclk1, ChgA, and β-Actin in the small intestine of Elp3$^{WT}$ and Elp3$^{\Delta IEC}$ mice naive or after Dibenzazepine (DBZ) treatment assessed by western blot are illustrated (G). Immunostainings and corresponding quantifications (left and right panels, respectively) of tuft cells (Dclk1$^+$ cells, in red) in the small intestine of Elp3$^{WT}$ and Elp3$^{\Delta IEC}$ mice treated or not with DBZ (mean values ± SEM; Student *t*-test ; $n = 3$; *$p < 0.05$, **$p < 0.01$) are illustrated (H). Source data are available online for this figure.

Atf4 expression indeed had some biological consequences as levels of Asparagine synthetase (Asns), an Atf4 target gene (Barbosa-Tessmann et al, 1999), were elevated in naive IECs lacking Elp3 (Fig. 6H). Moreover, Atf5, Eif4ebp1, Phgdh, Psat1 and Psph, which are also parts of this transcriptional signature, showed elevated mRNA levels in intestines lacking Elp3 (Fig. 6H, left panel). Consistently, protein levels of Phgdh and Psat1 were also elevated upon *Elp3* deficiency (Fig. 6H, right panel). To more precisely establish the link between mTORC1 activation and Atf4 protein levels, we treated or not Elp3$^{\Delta IEC}$ mice with the mTORC1 inhibitor Rapamycin and assessed Atf4 protein levels in the intestinal epithelium. As expected, Rapamycin inhibited S6 phosphorylation (Fig. 6I). Interestingly, Atf4 as well as its target gene Asns were both downregulated by Rapamycin (Fig. 6I). Therefore, mTORC1 controls Atf4 protein levels in the intestine lacking Elp3.

To assess the contribution of some Atf4 target genes in the regulation of intestinal tuft cell differentiation, we overexpressed or not Asns in ex-vivo organoids from WT mice and treated or not the resulting organoids with mIL-13. Atf4 levels decreased upon mIL-13 stimulation, which further supports its negative role in tuft cell differentiation (Fig. 6J). Interestingly, Asns levels also decreased in IL-13-stimulated ex-vivo organoids (Fig. 6J). Moreover, Asns overexpression negatively regulated Dclk1 protein levels, thus demonstrating that Asns inhibits intestinal tuft cell differentiation (Fig. 6J). Collectively, our results suggest that the mTORC1-Atf4 branch acts as a negative regulator of intestinal tuft cell differentiation.

## Elp3 promotes the mRNA translation of the mTORC1 inhibitor Nprl2

Elp3 negatively regulates mTORC1 activation in yeast, at least by promoting the mRNA translation of mTORC1 inhibitors (Candiracci et al, 2019). To assess whether this mechanism also applies to our experimental model, we assessed protein levels of a variety of mTORC1 inhibitors in the intestinal epithelium from both Elp3$^{WT}$ and Elp3$^{\Delta IEC}$ mice and found that Nprl2 but not Tsc1/2 was not properly expressed at the protein level upon *Elp3* deficiency (Fig. 7A, left panels). Importantly, Nprl2 mRNA levels remained unchanged in the intestinal epithelium from Elp3$^{\Delta IEC}$ mice, indicating that Elp3 regulates Nprl2 expression at the post-transcriptional level (Fig. 7A, right panel). Likewise, the depletion

of Elp3 in a murine intestinal epithelial cell line ("mIECs") also impaired Nprl2 protein but not mRNA levels (Fig. 7B,C). Therefore, Elp3 limits mTORC1 activation, potentially by promoting Nprl2 expression in the intestine. In support of this conclusion, we also established the polysome profiling on the Nprl2 transcript using extracts from both Elp3$^{WT}$ and Elp3$^{\Delta IEC}$ mice. We observed an increased ribosome density on the Nprl2 transcript upon *Elp3* deficiency (Fig. 7D). Considering that Nprl2 protein but not mRNA levels are reduced in IECs lacking Elp3, these results suggest a ribosome pausing on Nprl2 mRNA, leading to decreased translational efficiency. Elp3 promotes mRNA translation in a codon-dependent manner (Bauer et al, 2012). Therefore, to assess whether Nprl2 is a direct target of Elp3, we generated an Nprl2 mutant in which Lys$^{AAA}$, Gln$^{CAA}$, and Glu$^{GAA}$ codons which rely on Elp3 to be decoded ("Nprl2 WT") were replaced by synonymous Lys$^{AAG}$, Gln$^{CAG}$, and Glu$^{GAG}$ codons ("Nprl2 Mut") which escape from Elp3 regulation (Fig. 7E). Both Nprl2 WT and Nprl2 Mut were expressed at very similar levels in HEK293 cells (Fig. 7E). Importantly, while Nprl2 WT expression was defective in HEK293 cells lacking Elp3, Nprl2 Mut was properly expressed upon Elp3 deficiency (Fig. 7E, top panel). Likewise, the depletion of cytosolic thiouridylase 2 (Ctu2), which acts in the same enzymatic cascade as Elp3, also interfered with the expression of Nprl2 WT but not Nprl2 Mut (Fig. 7E, bottom panel). Therefore, Nprl2 relies on Elp3 for its codon-dependent mRNA translation (El Yacoubi et al, 2012).

In agreement with the notion that Elp3 promotes tuft cell differentiation through Nprl2, we next depleted Nprl2 in ex-vivo organoids from mouse intestinal crypts and noticed that IL-13-dependent Dclk1 induction was robustly impaired, at least through Atf4 stabilization (Fig. 7F). Likewise, the production of Dclk1$^+$ cells upon IL-13 stimulation was severely impaired in ex-vivo organoids in which Nprl2 was depleted (Fig. 7G). Therefore, Elp3 promotes intestinal tuft cell differentiation, at least through Nprl2 mRNA translation.

## Atf4 is a negative regulator of intestinal tuft cell differentiation

To assess whether Atf4 negatively regulates tuft cell differentiation, we carried out several experiments in which Atf4 expression was impaired or potentiated in order to assess consequences on tuft cell differentiation. We first pre-treated or not Elp3$^{\Delta IEC}$ mice with the

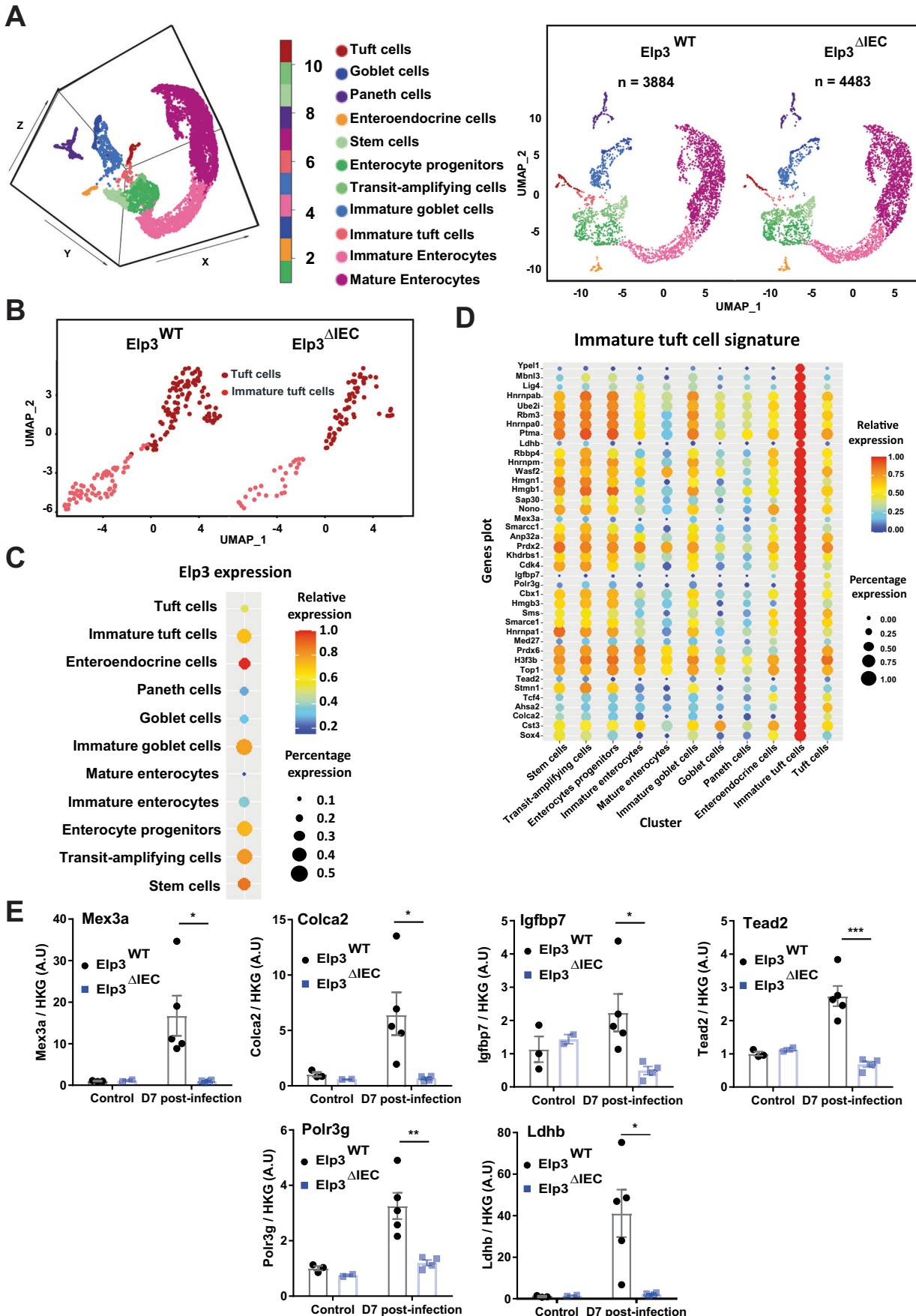

◀ **Figure 4.** *Elp3* deficiency impairs tuft cell fate determination at an early step of differentiation.

(A) Cell-type clustering in intestines of the indicated genotypes revealed through single-cell RNA sequencing. A three-dimensional graphical representation of cell-type clustering in the small intestine of both Elp3^WT and Elp3^ΔIEC mice subjected to rIL-13 treatment overnight is illustrated (left panel; $n = 4$ pooled mice). A two-dimensional graphical representation of cell-type clustering in each genotype is also illustrated (right panel; $n = 2$ pooled littermate mice per genotype). (B) Tuft cell clustering in intestines of the indicated genotypes. The figure shows a tuft cell clustering with the Uniform Manifold Approximation and Projection (UMAP), illustrating the similar distribution of both immature and mature tuft cell populations between Elp3^WT and Elp3^ΔIEC mice subjected to rIL-13 treatment overnight ($n = 2$ pooled mice per genotype). (C) Elp3 expression profile in mouse intestinal epithelial cells revealed through single-cell RNA sequencing. Dotplot illustrating Elp3 expression among epithelial subtypes in the small intestine of Elp3^WT mice subjected to rIL-13 treatment overnight. (D) Identification of the transcriptional signature of immature tuft cells. Dotplot illustrating candidates whose mRNA levels are enriched in immature tuft cells (single-cell RNA sequencing data carried out with extracts from Elp3^WT mice subjected to rIL-13 treatment overnight). (E) Expression of candidates enriched in immature tuft cells is impaired in Elp3^ΔIEC mice infected with *N.brasiliensis*. mRNA levels of the indicated candidates in the small intestine of Elp3^WT and Elp3^ΔIEC mice naive or 7 days post-*N. brasiliensis* infection were quantified by real-time PCRs. Normalization was calculated on the average of two housekeeping genes Gapdh and 36b4 (mean values ± SEM; Student *t*-test; $n = 3$ and 2 for naive Elp3^WT and Elp3^ΔIEC mice; $n = 5$ and 4 for infected Elp3^WT and Elp3^ΔIEC mice; *$p < 0.05$, **$p < 0.01$, ***$p < 0.001$). Source data are available online for this figure.

integrated stress response (ISR) inhibitor Isrib, which inhibits Atf4 translation, and subsequently stimulated them with rIL-13. Isrib potentiated the rIL-13-dependent induction of Dclk1 protein levels as well as the number of Dclk1^+ cells (Fig. 8A,B, respectively). On the other hand, Isrib had no effect on rIL-13-dependent goblet cell differentiation (Fig. 8C). Isrib treatment of ex-vivo organoids generated from WT mouse intestines also potentiated IL-13-dependent tuft cell differentiation, at least by interfering with Atf4 expression (Fig. 8D,E, respectively). To better define the role of Atf4 in this regulation, we next depleted Atf4 in ex-vivo organoids and assessed the consequences on tuft cell differentiation. Atf4 deficiency potentiated IL-13-dependent Dclk1 induction and also increased the number of Dclk1^+ cells upon mIL-13 stimulation (Fig. 8F,G). We next conducted gain-of-function experiments by triggering UPR or by potentiating the Atf4 response. In this context, we pre-incubated or not ex-vivo organoids generated with intestinal crypts from WT mice with Thapsigargin, an UPR inducer. These ex-vivo organoids were subsequently treated with mIL-13 to induce tuft cell differentiation. As expected, Thapsigargin triggered UPR, as evidenced by stabilized Atf4, enhanced Atf6 and Bip levels as well as by IRE1α phosphorylation (Fig. 9A). Importantly, Thapsigargin interfered with the production of Dclk1^+ cells (Fig. 9B). To explore whether Atf4 overexpression alone mimics this effect, we next overexpressed an Atf4 mutant, which escapes from degradation by mutating serine 219 into asparagine, in ex-vivo organoids (Lassot et al, 2001). Overexpression of mutated Atf4 impaired IL-13-dependent tuft cell differentiation, as evidenced by lower levels of Dclk1 and a decreased number of Dclk1^+ cells (Fig. 9C,D). Collectively, our results suggest that Atf4 specifically blocks intestinal tuft cell differentiation.

Yeast strains lacking the U34 tRNA thiolase, which acts in the same enzymatic cascade as Elongator, show defects in amino acid homeostasis and activate the amino acid starvation regulator Gcn4/Atf4 (Zinshteyn and Gilbert, 2013). This yeast strain actually behaves as if it was amino acid-starved (Gupta et al, 2019). Atf4 is typically activated through the amino acid response (AAR) signaling pathway when an imbalance of essential amino acids occurs (Kilberg et al, 2009). Therefore, we assessed whether Atf4 stabilization seen upon *Elp3* deficiency results from an imbalance in amino acid levels. To address this issue, we carried out metabolomic analyses using IECs from both Elp3^WT and Elp3^ΔIEC mice and noticed that levels of some but not all amino acids were elevated in IECs from Elp3^ΔIEC mice (Fig. 9E). More specifically, lysine, arginine and valine were the most upregulated

amino acids in IECs lacking Elp3 (Fig. 9E). This phenomenon may reflect a compensatory mechanism in cells acting as if they were amino acid-starved. Therefore, we decided to feed WT mice with arginine, lysine, and valine (RKV)-deprived diet in order to assess the consequences on intestinal tuft cell differentiation. Mice fed with this RKV-deprived diet lost weight overtime (Fig. 9F). As expected, this RKV-deprived diet triggered Atf4 stabilization in IECs (Fig. 9G). Importantly, tuft cell differentiation was impaired in rIL-13-stimulated mice fed with this RKV-deprived diet (Fig. 9H). Of note, these mice showed slightly more goblet cells (Fig. 9I). Therefore, these data indicate that an amino acid-deprived diet blocks intestinal tuft cell differentiation, at least through Atf4 stabilization, which further supports the demonstration that Atf4 acts as a negative regulator of intestinal tuft cell differentiation.

## Atf4 overexpression in mouse IECs blocks tuft cell differentiation

To explore whether Atf4 overexpression also blocks intestinal tuft cell differentiation in vivo, we generated a mouse model, referred to as "TgAtf4^IEC", in which Aft4 is specifically overexpressed in IECs. TgAtf4^IEC mice looked smaller than WT mice and also had shorter small intestine and colon (Figs. 10A,B, respectively and EV4A,B, respectively). As expected, an enhanced Atf4-dependent transcriptional signature was found in intestines from TgAtf4^IEC mice, as judged by elevated Atf4 target genes such as Sesn2, Asns, Chop, Atf5, Gadd34 and Stc2 mRNA levels, compared to WT mice (Fig. 10C). The Atf4 target gene Asns was also increased at the protein level in intestines from TgAtf4^IEC mice (Fig. EV4C). Of note, S6 phosphorylation was not enhanced in the intestine from these mice, indicating that Atf4 overexpression has no impact on S6 phosphorylation (Fig. EV4C). On the other hand, 4EBP1, whose transcription is known to be induced by Atf4, was more expressed in the intestinal epithelium of TgAtf4^IEC mice (Fig. EV4C) (Yamagushi et al, 2008). We next infected both WT and TgAtf4^IEC mice with *N. brasiliensis* and assessed parasite clearance. Both strains showed similar infection rates in the lung but TgAtf4^IEC mice showed elevated worm numbers in the intestine 7 days post-infection (Fig. 10D,E). Although *N. brasiliensis* infection caused signs of UPR activation, those pathways were not potentiated in TgAtf4^IEC when compared to WT mice (Fig. EV4D). Importantly, Stat6 phosphorylation as well as Dclk1 induction were severely impaired in intestines from TgAtf4^IEC mice 7 days post-infection,

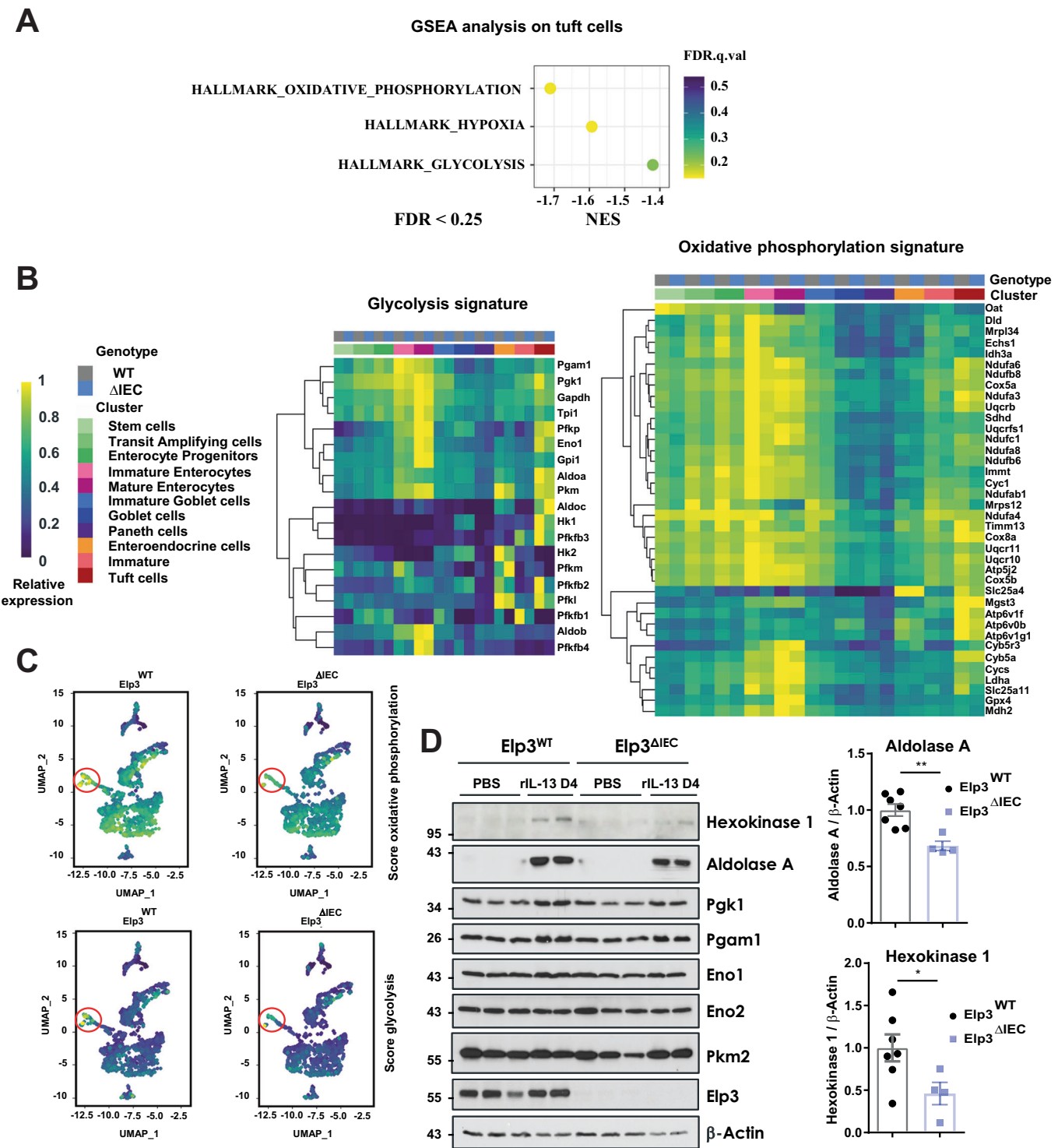

suggesting that tuft cell amplification was abolished upon Atf4 overexpression in vivo (Fig. 10F). Indeed, the amplification of tuft cells post-infection was severely impaired in TgAtf4[IEC] mice (Fig. 10G). Likewise, Dclk1 and IL-25 mRNA levels were not properly induced in infected TgAtf4[IEC] mice (Fig. 10H). The induction of Retnlβ upon infection was also defective in TgAtf4[IEC] mice (Fig. EV4E). To assess the consequences of Atf4

overexpression on IL-13 signaling, we next stimulated both WT and TgAtf4[IEC] mice with rIL-13 and assessed the biological consequences. Although rIL-13-dependent Stat6 phosphorylation after two hours of treatment was comparable in both WT and TgAtf4[IEC] mice, Dclk1 induction 4 days later was impaired upon Atf4 overexpression (Figs. EV4F and 10I, respectively). Of note, the expansion of goblet cells after rIL-13 stimulation was comparable in

**Figure 5.   *Elp3* deficiency impairs some metabolic pathways in tuft cells.**

(A) Identification of defective metabolic pathways upon *Elp3* deficiency in tuft cells. A gene set enrichment analysis (GSEA) was carried out on tuft cells of both Elp3[WT] and Elp3[ΔIEC] mice subjected to rIL-13 treatment and revealed impaired oxidative phosphorylation and glycolysis upon *Elp3* deficiency ($n = 2$ pooled mice per genotype). The statistically significant enrichment of the gene set was selected based on false discovery rate (FDR) ≤0.25. (B, C) Gene candidates involved in both oxidative phosphorylation and glycolysis are enriched in tuft cells compared to other secretory cells and both signatures are impaired upon *Elp3* deficiency. A heatmap revealing candidates involved in glycolysis and oxidative phosphorylation (left and right panels, respectively) based on transcriptomic analyses done with extracts from the small intestine of both Elp3[WT] and Elp3[ΔIEC] mice subjected to rIL-13 stimulation overnight is illustrated ($n = 2$ pooled mice per genotype) (B). A UMAP showing both metabolic pathways enriched in tuft cells ($n = 2$ pooled mice per genotype) is illustrated (data with enterocytes are removed for clarity purposes). Tuft cells are highlighted in red circles (C). (D) *Elp3* deficiency in the intestine interferes with IL-13 induction of specific glycolytic enzymes. Elp3[WT] and Elp3[ΔIEC] mice were treated or not with rIL-13 for 4 days, and the resulting extracts from the intestines were subjected to western blot analyses using the indicated antibodies. A quantification of the relative expression of both glycolytic enzymes (ratio on β-Actin) in IL-13-stimulated mice from several experiments is illustrated (mean values ± SEM; Student *t*-test; $n \geq 4$; *$p < 0.05$, **$p < 0.01$). Source data are available online for this figure.

both WT and TgAtf4[IEC] mice (Fig. EV4G). A similar experiment was conducted with ex-vivo organoids generated with intestinal crypts from both WT and TgAtf4[IEC] mice. As expected, some Atf4 target genes were elevated in ex-vivo organoids from TgAtf4[IEC] mice (Fig. EV4H). On the other hand, mRNA levels of IEC subtype markers were comparable in both WT and TgAtf4[IEC] mice (Fig. EV4I). Here again, the IL-13-dependent induction of Dclk1 was impaired upon Atf4 overexpression, as were Dclk1[+] cell numbers (Fig. 10J,K). Of note, the number of Mucin2[+] cells slightly increased upon mIL-13 stimulation in ex-vivo organoids from TgAtf4[IEC] (Fig. EV4J). Collectively, our data define Atf4 as a negative regulator of intestinal tuft cell differentiation.

# Discussion

Tuft cell expansion, a critical step in the anti-helminth immune response in the intestine, relies on translational reprogramming. We demonstrate here that the loss of the $U_{34}$ tRNA-modifying enzyme Elp3 specifically blocks intestinal tuft cell differentiation, at least through mTORC1 activation and Atf4 stabilization. We further define the Gator1 subunit and mTORC1 inhibitor Nprl2 as a candidate whose codon-dependent mRNA translation relies on Elp3. Therefore, our work defines Atf4 as a negative regulator of intestinal tuft cell differentiation.

$U_{34}$ tRNA modifications regulate translational reprogramming by facilitating the translation of still poorly characterized mRNA candidates. In yeast, phenotypical defects linked to *Elp3* deficiency mainly result from Gcn4 activation (Zinshteyn and Gilbert, 2013). Atf4 activation upon loss of some $U_{34}$ tRNA modifications has also been described in mice. Indeed, *Elp3* deficiency impairs indirect neurogenesis, at least due to enhanced Atf4 levels (Laguesse et al, 2015). Moreover, the loss of *Elp3* in both T cells and in the hematopoietic system also activates an Atf4-dependent pathway (Lemaitre et al, 2021; Rosu et al, 2021). *Elp3* deficiency in hematopoietic progenitors did not trigger the canonical UPR pathway, and we now show that this conclusion also applies to IECs (Rosu et al, 2021). Importantly, myeloid cells lacking Elp3 did not potentiate any Atf4-dependent response, suggesting that Atf4 is not systematically activated in primary cells lacking some $U_{34}$ tRNA modifications (Chen et al, 2022). Transformed epithelial and melanoma cells lacking Elp3 did not show any Atf4 activation (Ladang et al, 2015; Delaunay et al, 2016; Rapino et al, 2021, 2018). In those cases, the lack of Elp3 was linked to a defective translation of mRNAs enriched in Lys[AAA], Gln[CAA], and Glu[GAA] codons known to

rely on Elp3 to be properly decoded, as also shown in yeast (Bauer et al, 2012; Ladang et al, 2015; Delaunay et al, 2016; Rapino et al, 2018, 2021). It is worth mentioning that both consequences seen upon *Elp3* deficiency may not be mutually exclusive. Indeed, Atf4 activation seen in IECs lacking Elp3 was due, at least in part, to mTORC1 activation. As a mechanism of enhanced mTORC1 in IECs lacking Elp3, we showed that the Gator1 subunit and mTORC1 inhibitor Nprl2 relies on Elp3 to be properly translated in a codon-dependent manner, as demonstrated in yeast (Candiracci et al, 2019). In support of the notion that Nprl2 is a direct target of Elp3, we observed a ribosome pausing on the Nprl2 transcript in IECs lacking Elp3. This result is in line with previous studies showing an elongation slowdown leading to an increased ribosome density on specific transcripts upon *Elp3* deficiency, with consequences on translation efficiency (Rapino et al, 2018; Rosu et al, 2021). Therefore, the initial event(s) through which Elp3 controls a variety of physiological and pathological processes is a codon-dependent mRNA translation of one or multiple direct targets whose identity is context-dependent.

The functional links between $U_{34}$ tRNA modifications and mTORC1/2 activation appear to be relevant in multiple experimental models and conserved throughout evolution. Mechanistically, Elp1 is activated by phosphorylation through an mTORC2-dependent pathway in melanoma cells (Rapino et al, 2018). In yeast, TORC2 also activates Elongator in a phospho-dependent manner, while Elongator promotes the translation of key components of TORC2 (Candiracci et al, 2019). On the other hand, Elongator promotes the expression of mTORC1 repressors in yeast, suggesting that $U_{34}$ tRNA modifications exert opposite roles on mTORC1 and mTORC2 activation (Candiracci et al, 2019). Our studies carried out in mice support these findings as we demonstrated that Elp3, and by extension Elongator, promote mTORC2 activation in macrophages but limit mTORC1 activity in IECs (Chen et al, 2022).

Mice lacking Elp3 in the intestinal epithelium fail to promote tuft cell differentiation, at least due to the activation of the mTORC1-Atf4 branch, suggesting that mTORC1 acts as a negative regulator of tuft cell differentiation in this context. Raptor, which is part of the mTORC1 complex, promotes tuft cell differentiation (Aladegbami et al, 2017). Therefore, the consequences of mTORC1 activation on tuft cell differentiation critically rely on Elp3 status. This unexpected fact is currently not understood. It remains to be seen whether protein synthesis downstream of mTORC1 changes in cells lacking some $U_{34}$ tRNA modifications.

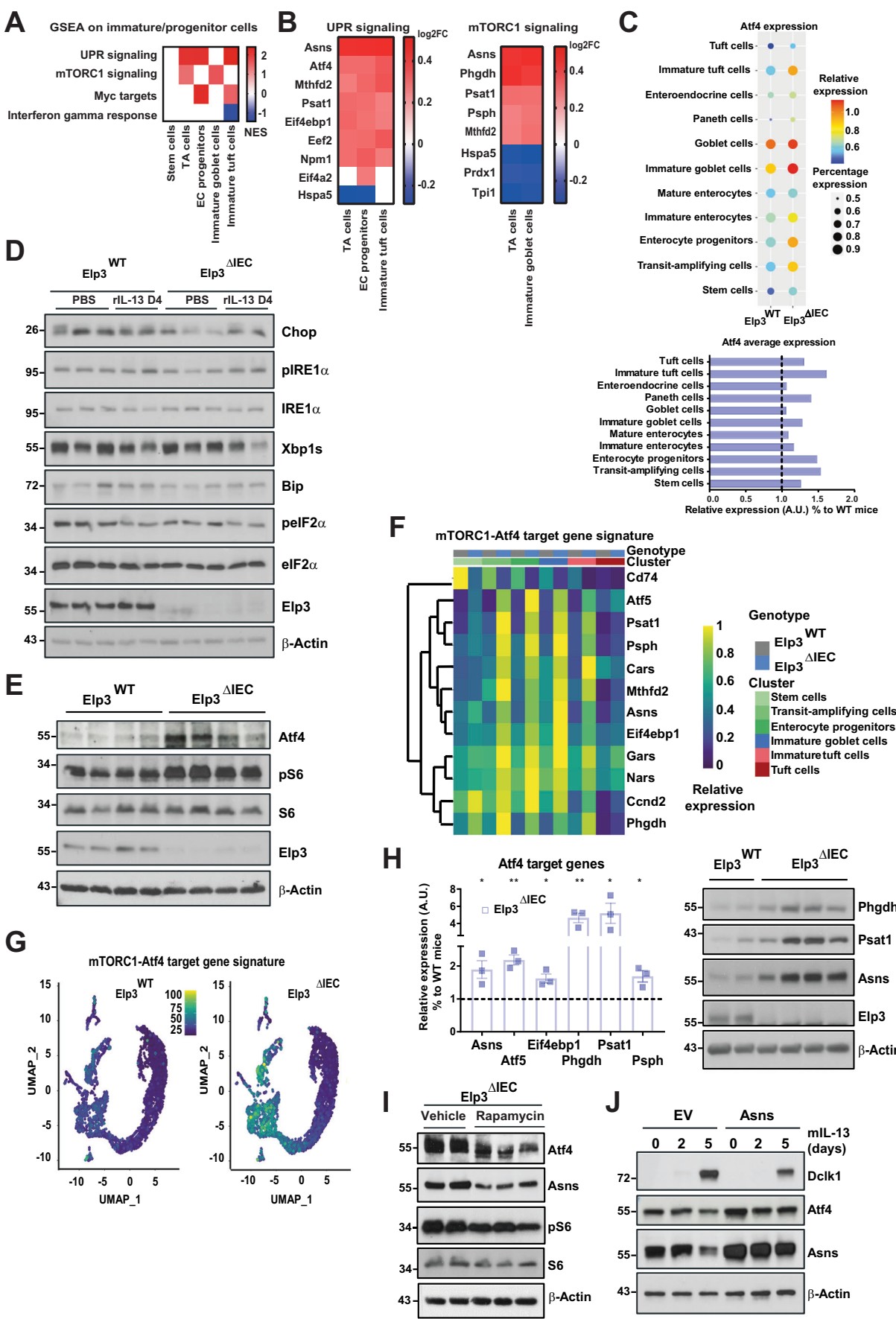

**Figure 6.  *Elp3* deficiency potentiates the mTORC1-Atf4 signature in intestinal immature epithelial cells.**

(A, B) Identification of activated signaling pathways in progenitor cells lacking Elp3. A GSEA on intestinal epithelial cell populations was carried out for both genotypes. A heatmap showing differentially regulated signaling pathways among indicated epithelial subtypes in the small intestine of both Elp3^WT and Elp3^ΔIEC mice subjected to rIL-13 treatment overnight is illustrated (n = 2 pooled mice per genotype) (A). The statistically significant enrichment of the gene set was selected based on false discovery rate (FDR) ≤0.25. Heatmaps depicting signatures highlighted in the GSEA analysis for both mTORC1 and UPR signaling pathways are shown (B). (C) Enhanced Atf4 expression in intestinal epithelial subtypes lacking Elp3. Dot plots and corresponding quantifications of Atf4 expression among epithelial subtypes in the small intestine of both Elp3^WT and Elp3^ΔIEC mice subjected to rIL-13 treatment overnight are illustrated (top and bottom panel, respectively). (D) The canonical UPR signaling is not defective in intestines lacking Elp3. Expression levels of UPR signaling effectors, Elp3 and β-Actin in the small intestine of both naive Elp3^WT and Elp3^ΔIEC mice (PBS) or treated with rIL-13 for 4 days were assessed by western blot. (E) mTORC1 signaling is upregulated in IECs lacking Elp3. Expression levels of mTORC1 signaling effectors, Elp3 and β-Actin in the small intestine of both naive Elp3^WT and Elp3^ΔIEC mice were assessed by western blot. (F, G) The mTORC1-Atf4 target gene signature is upregulated in immature populations of intestines lacking Elp3. A heatmap revealing the differentially expressed genes from the mTORC1-Atf4 target gene signature based on transcriptomic analyses done with extracts from the small intestine of both Elp3^WT and Elp3^ΔIEC mice subjected to rIL-13 stimulation overnight is illustrated (n = 2 pooled mice per genotype). Differentially expressed genes were identified using FindMarkers/FindAllMarkers functions from the Seurat package (Wilcoxon rank-sum test). Only genes that were significantly up or down-regulated in one or more of the represented populations were selected, based on p_val_adj <0.05 and logFC >0.25 (F). An UMAP showing the mTORC1-Atf4 target gene signature upregulated in immature intestinal cells lacking Elp3 (n = 2 pooled mice per genotype) is illustrated (G). (H) Upregulation of some target genes of the mTORC1-Atf4 axis in intestines lacking Elp3. mRNA levels of some mTORC1-Atf4 target genes in the small intestine of both naive Elp3^WT and Elp3^ΔIEC mice (ratio to Elp3^WT mice) were quantified by real-time PCRs (left panel). Normalization was calculated on the average of the three housekeeping genes Gapdh, 36b4 and β2m (mean values ± SEM; Student t-test; n = 3; *p < 0.05, **p < 0.01). Expression levels of some mTORC1-Atf4 target genes, Elp3 and β-Actin in the small intestine of both naive Elp3^WT and Elp3^ΔIEC mice were assessed by western blot (right panel). (I) Pharmacological inhibition of mTORC1 decreases Atf4 protein levels in the intestine. Elp3^ΔIEC mice were treated or not with Rapamycin (see methods for details), and protein extracts from the intestinal epithelium of these mice were subjected to western blot analyses using the indicated antibodies. (J) Asns overexpression inhibits IL-13-dependent tuft cell differentiation. Ex-vivo organoids generated with intestinal crypts from WT mice were infected with a control lentivirus (EV empty vector) or with an Asns-overexpressing lentiviral construct. The resulting ex-vivo organoids were then stimulated or not with mIL-13 for 2 or 5 days, and cell extracts were subjected to western blot analyses. Source data are available online for this figure.

Our study demonstrates that *Elp3* deficiency specifically impairs the differentiation of tuft but not any other epithelial cells of mouse intestines. *Elp3* deficiency potentiates Atf4 in TA cells, in both immature tuft and goblet cells as well as in enterocyte progenitors. Based on these observations, one may expect *Elp3* deficiency to interfere with the differentiation of several epithelial lineages. This specific defect on tuft cells resulting from the loss of *Elp3* is currently unclear. The answer may come from key Atf4 target genes that could preferentially be expressed in tuft but not in other intestinal progenitors. In this context, it is worth mentioning that the Atf4 target gene *Asns* is upregulated in both TA and immature tuft cells. *Asns* codes for Asparagine synthetase, which catalyzes the conversion of aspartate and glutamine to asparagine and glutamate in an ATP-dependent manner (Balasubramanian et al, 2013). How this metabolic reaction would block tuft cell differentiation is unknown but it may nevertheless suggest that metabolism is crucial in regulating epithelial differentiation in the intestine. In this context, we showed that tuft cells were enriched in enzymes involved in both oxidative phosphorylation and glycolysis. Moreover, specific glycolytic enzymes are robustly induced by IL-13, a cytokine that promotes tuft cell amplification. Therefore, the specific role of Elp3 on tuft cell differentiation may result from its control of metabolic pathways enriched in tuft cell progenitors. Besides a negative role of Aft4 in tuft cell differentiation, tuft cell progenitors may specifically express some candidates enriched in Lys^AAA, Gln^CAA, and Glu^GAA codons, which would make them more sensitive to Elp3 expression and activity.

The definition of Atf4 as a negative regulator of intestinal tuft cell differentiation opens the possibility that a variety of evolutionary conserved cascades known to be regulated by the host diet/nutritional or health status and connected to Atf4 may ultimately interfere with intestinal tuft cell differentiation with consequences on the anti-helminth immune response in the intestine. Our work may then contribute to better understand at the molecular level how the health status can influence the immune response in the intestine.

Multiple studies defined Elp3 as a promising target to fight against a variety of epithelial malignancies as well as metastatic melanomas (Ladang et al, 2015; Delaunay et al, 2016; Rapino et al, 2018). Our current study in which a role of Elp3 in the type 2 immune response is described, indicates that interfering with Elp3 may cause undesired consequences on the anti-helminth immune response. Future studies should tell us whether U_34 tRNA modifications have any role in the immune response to other pathogens.

# Methods

## Cell lines

Both HEK293 cells and mIECs (a gift from Dr. Thomas Marichal, GIGA, University of Liège, Liège, Belgium) were cultured in Dulbecco's Modified Eagle's Medium (DMEM) (Lonza, Basel Switzerland) supplemented with 10% fetal bovine serum (FBS) (Sigma-Aldrich, St-Louis, MO, USA), L-glutamine and antibiotics (Lonza). Flag-Nprl2 WT and Flag-Nprl2 Mut were generated by VectorBuilder using the pLV[Exp]-Hygro-EF1A as a lentiviral vector backbone.

## Animal strains

The *Elp3^fl/fl* strain, referred to as Elp3^WT throughout this study, was generated as previously described (Ladang et al, 2015). In order to delete *Elp3* in IECs, *Elp3^fl/fl* mice were crossed with the *Villin-Cre* strain (Elp3^ΔIEC). The *Rosa26^fl-STOP-fl-Atf4* mouse strain was purchased from Jackson Laboratory (B6;129×1-Gt(ROSA)26Sor^tm2(ATF4)Myz/J, strain # 029394). In order to overexpress Atf4 in IECs, *Rosa26^fl-STOP-fl-Atf4* mice were crossed with the *Villin-Cre* strain to generate the strain referred to as "TgAtf4^IEC". Mice were analyzed at 8–10 weeks of age, regardless of the gender. For comparisons of Elp3^WT and Elp3^ΔIEC mice or wild-type and TgAtf4^IEC mice, littermates were

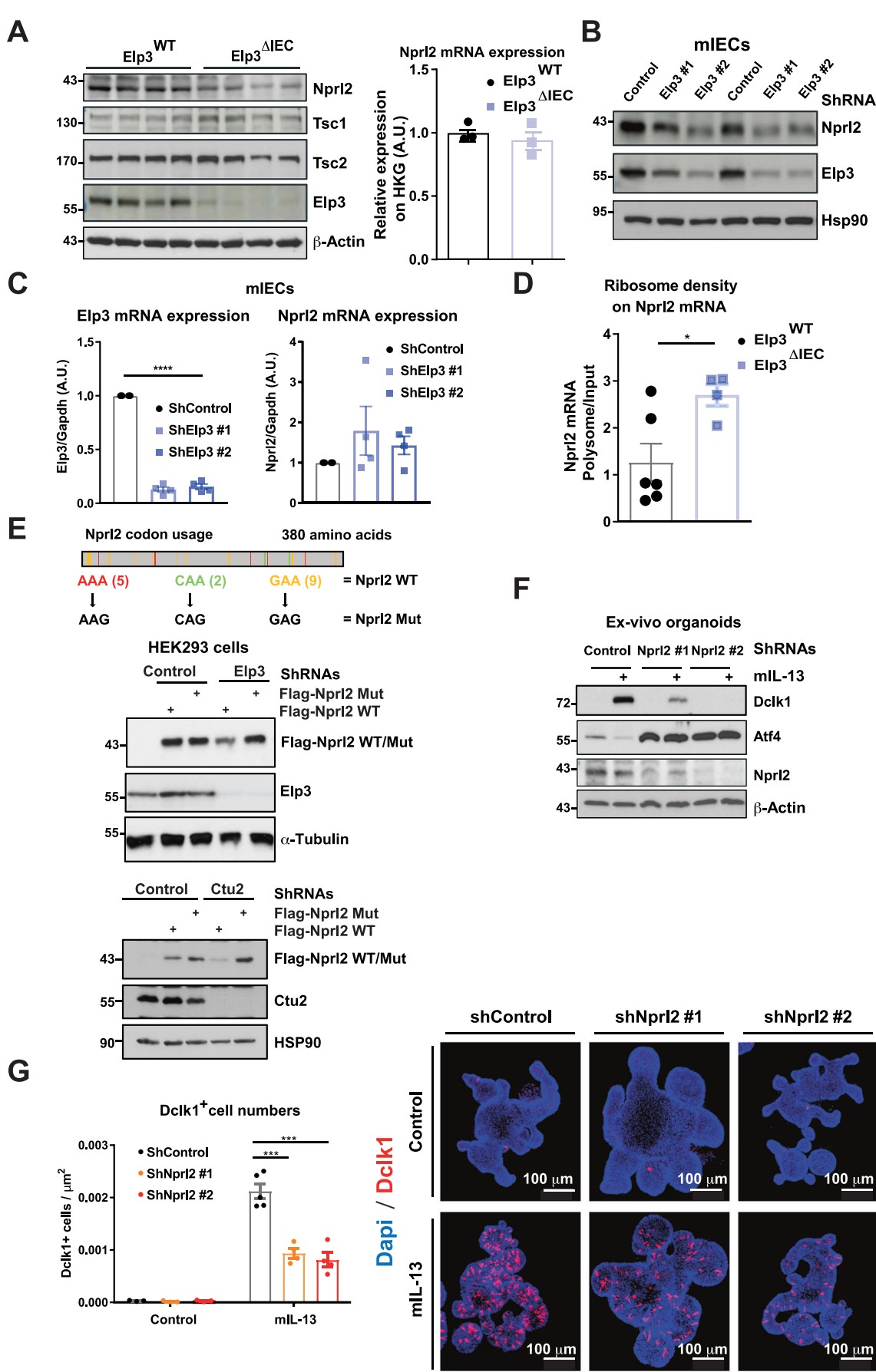

Figure 7. Elp3 promotes Nprl2 mRNA translation.

(A) Defective Nprl2 expression at the protein level upon *Elp3* deficiency in IECs. On the left, extracts from the intestines of both naive Elp3^WT and Elp3^ΔIEC mice were subjected to western blot analyses using the indicated antibodies. On the right, Nprl2 mRNA levels in the small intestine of naive Elp3^WT and Elp3^ΔIEC mice were quantified by real-time PCRs. Normalization was calculated on the housekeeping gene Gapdh (mean values ± SEM; $n = 3$). (B, C) Elp3 promotes Nprl2 expression. An immortalized mouse intestinal epithelial cell line (mIECs) was transfected with the indicated shRNA construct and extracts from the resulting cells were subjected to western blot analyses using the indicated antibodies (B). Elp3 and Nprl2 mRNA levels were also quantified by real-time PCRs in all indicated experimental conditions (C). mRNA levels in cells transfected with the shRNA control were set to 1 and levels in other conditions were relative to that after normalization with Gapdh mRNA levels (mean values ± SEM; Student *t*-test; $n = 4$; ****$p < 0.0001$). (D) Enhanced ribosome density on the Nprl2 transcript upon *Elp3* deficiency. A polysome profiling experiment was performed on IECs from both Elp3^WT and Elp3^ΔIEC naive mice. Ribosome density on the Nprl2 transcript was calculated by real-time PCRs (polysomal fraction/input ratio; mean values ± SEM; Student *t*-test; $n \geq 4$; *$p < 0.05$). (E) Elp3 promotes Nprl2 mRNA translation in a codon-dependent manner. HEK293 cells infected with the shRNA control or targeting either Elp3 or Ctu2 were transfected with the indicated expression construct (cf the schematic representation of both wild-type and mutated Nprl2 with the localization of all mutations) and the resulting extracts were subjected to western blot analyses using the indicated antibodies. (F, G) Nprl2 promotes tuft cell differentiation. Western blot analyses using the indicated antibodies were conducted using extracts from untreated or IL-13-stimulated ex-vivo organoids lacking or not Nprl2 (100 ng/ml, 3 days) (F). Anti-Dclk1 immunofluorescence analyses were also carried out to quantify the number of Dclk1^+ cells (in red) in all indicated experimental conditions (mean values ± SEM; Student *t*-test; $n \geq 3$ organoids; ***$p < 0.001$) (G). Source data are available online for this figure.

analyzed. All experiments were approved by the local Ethical Committee (University of Liège, Liège, Belgium). All mice strains used in this study are listed in Table EV2.

## Parasite infections

*N. brasiliensis* life cycle was maintained in male Wistar rats, as described (Rolot et al, 2018). Infectious L3 larvae were isolated from day 6 to 9 fecal cultures using a Baermann apparatus, washed in sterile PBS, counted and used for subcutaneous infections (550 L3 larvae/mouse). Fecal eggs, L4 larvae in the lung and adult worms in the intestine were collected and counted, as previously described (Camberis et al, 2003).

## ATP luciferase assay

Parasite viability was measured using the ATP CellTiter-Glo® Luminescent Cell Viability Assay (#G7572, Promega). Adult worms were collected and separated by gender after examination under the microscope. Groups of three intact worms were suspended in 120 μL of PBS, placed into 1.5 ml brown microtubes, and the same volume of the CellTiter-Glo® Reagent was added. Worms were homogenized using a motorized pestle and incubated for 10 minutes at room temperature. After centrifugation at $1000 \times g$ for 3 minutes, 100 μl of supernatant was transferred to a 96-well opaque-walled plate and applied to a luminometer (Enspire Multimode plate reader, Perkin Elmer) to measure luminescence for one second. A standard curve was generated using ATP (#A7699, Sigma-Aldrich) according to the manufacturer's instructions.

## Mice treatments

IL-13-anti-IL-13 complexes were prepared by mixing 5 μg of recombinant mouse IL-13 (#575908, Biolegend) with 25 μg of IL-13 monoclonal antibody (eBio1316H,# 16-7135-85, Invitrogen) in PBS. rIL-13 or vehicle (PBS) was injected intraperitoneally every 2 days for long treatments. Depending on the experiment, intestinal samples were collected after 2 hours or at day 4 after injection. For integrated stress response (ISR) inhibition, Isrib (#SML0843, Sigma-Aldrich) was resuspended in DMSO (5 mg/mL). The Isrib solution was heated at 40 °C and vortexed every 30 seconds until the solution became clear. Isrib (5 mg/kg) or vehicle (20% DMSO,

20% polyethylene glycol 400 (PEG-400) in saline solution) was administrated twice daily for 2 weeks to Elp3^ΔIEC mice by intraperitoneal injection. Four and 2 days before the end of the treatment, mice were injected with rIL-13 to induce tuft cell differentiation. For Notch inhibition in vivo, mice received a daily intraperitoneal injection of either vehicle (5% DMSO, 0.5% Hydroxypropylmethylcellulose, 0.1% Tween-80 in water) or Dibenzazepine (DBZ) (#HY-13526, MedChemExpress) for 5 days at a dose of 15 μmol/kg. For Rapamycin treatments, Elp3^ΔIEC mice received a daily intraperitoneal injection of either vehicle (4% Ethanol, 5% Tween-80, 5% polyethylene glycol 400 (PEG-400) in water) or Rapamycin 8 mg/kg (#R-5000, LC Laboratories) for 4 days. For amino acid deprivation in vivo, mice were fed with an RKV-deprived diet for 2 weeks. Body weight was monitored daily, and the experiment was stopped before the mice lost 20% of their body weight. 4 and 2 days before the end of the experiment, mice were treated with rIL-13 to induce tuft cell differentiation. All reagents used in this study are listed in Table EV2.

## Tissue processing and intestinal epithelial cell isolation

After euthanasia, proximal intestine was extracted from mice, washed with PBS and cut longitudinally. About 5 cm of the proximal tissue was rolled and used for histological purposes and 5 cm was used for intestinal epithelial cell extraction. In brief, intestine was incubated twice for 10 minutes at 37 °C in a HBSS-EDTA buffer (5 mM). Cells were harvested, washed twice in PBS and snap-frozen.

## Lamina propria isolation

To isolate leukocytes from the lamina propria, the first 2/3 of the small intestine was dissected and flushed with 20 mL PBS. Peyer's patches and fat tissue were carefully removed with scissors and the intestine was cut longitudinally before further washed in 50 mL tubes by shaking in cold PBS. The intestine was then cut into 5 cm pieces and epithelial cells were removed following incubation at 37 °C under gentle agitation in HBSS without calcium or magnesium and supplemented with 5 mM EDTA and 10 mM HEPES. Then, the tissue was gently vortexed, the supernatant removed, and a second incubation was performed to remove all epithelial cells. The intestine was then transferred in a Petri dish on ice and further cut into small pieces, minced with a scalpel, and

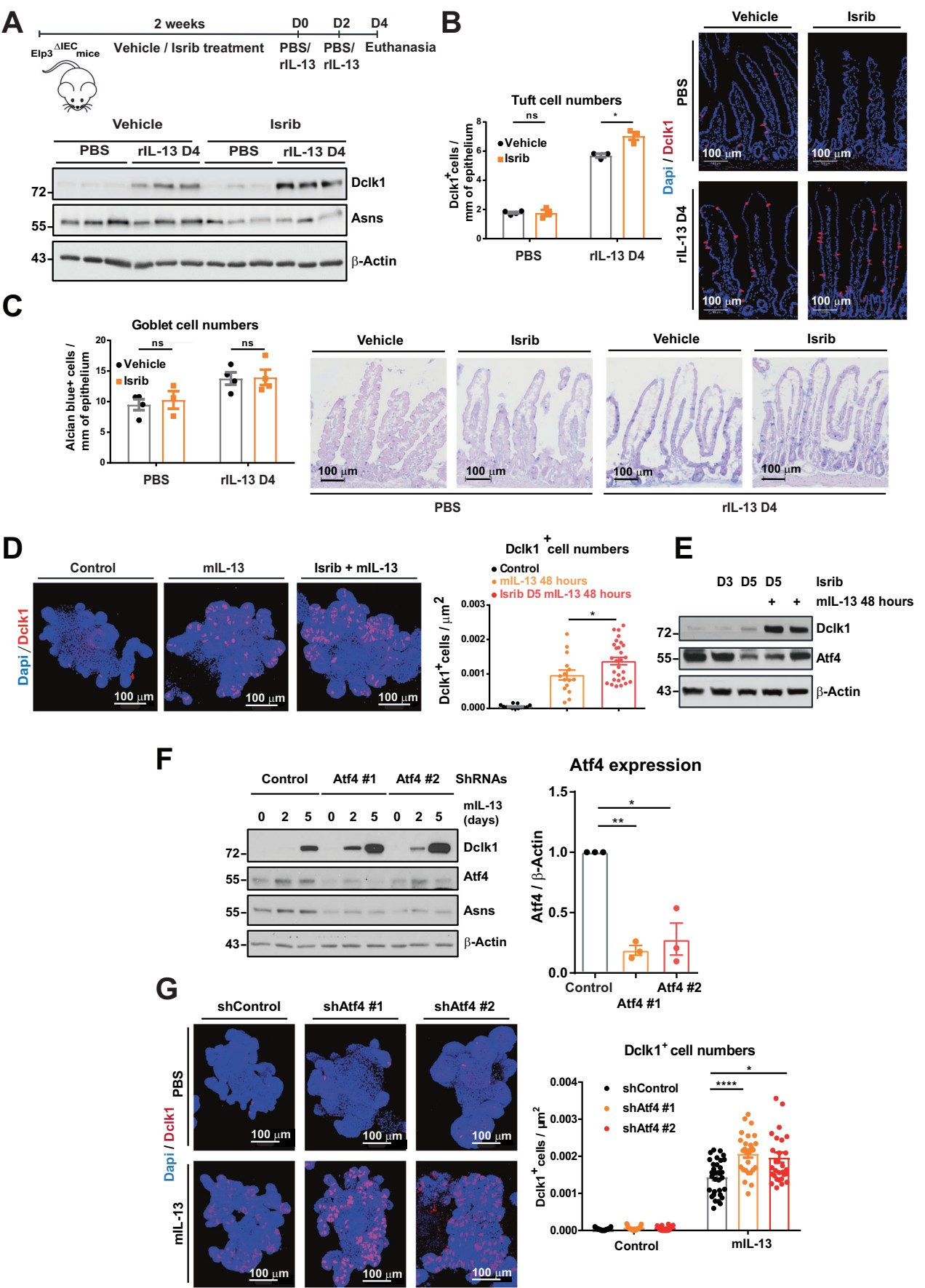

**Figure 8.  Atf4 inhibition promotes intestinal tuft cell differentiation.**

(A, B) Inhibition of Atf4 translation potentiates IL-13-dependent tuft cell differentiation in intestines lacking Elp3. Elp3$^{\Delta IEC}$ mice received twice daily Isrib injections for 2 weeks to reduce Atf4 expression and were injected with rIL-13 4 days before the end of the experiment. Western blot analyses with extracts from the intestines of the resulting mice were carried out with the indicated antibodies (A). Immunostainings and corresponding quantifications (left and right panels, respectively) of tuft cells (Dclk1$^+$ cells, in red) in the small intestine of Elp3$^{\Delta IEC}$ mice treated with rIL-13 and with Isrib or with the vehicle (mean values ± SEM; Student *t*-test; $n \geq 3$; *$p < 0.05$) are illustrated (B). (C) Inhibition of Atf4 translation does not have any impact on IL-13-dependent goblet cell differentiation. Immunostainings and corresponding quantifications (left and right panels, respectively) of goblet cells (Alcian Blue$^+$ cells) in the small intestine of Elp3$^{\Delta IEC}$ mice treated with rIL-13 and with Isrib or with the vehicle (mean values ± SEM; Student *t*-test; $n \geq 3$) are illustrated. (D, E) Inhibition of Atf4 translation potentiates IL-13-dependent tuft cell differentiation, as evidenced by immunofluorescence (D) and western blot (E) analyses. Immunostainings and corresponding quantifications (left and right panels, respectively) of Dclk1$^+$ cells (in red) in WT organoids treated with mIL-13 and with Isrib or with the vehicle (mean values ± SEM; Mann–Whitney test; $n \geq 13$ organoids; *$p < 0.05$) are illustrated (D). Expression levels of Dclk1, Atf4, and β-Actin in WT organoids treated with mIL-13 and with Isrib or with the vehicle are illustrated (E). (F, G) Atf4 deficiency potentiates IL-13-dependent intestinal tuft cell differentiation. Ex-vivo organoids generated with intestinal crypts from WT mice were infected with the indicated shRNA constructs, and the resulting ex-vivo organoids were treated or not with mIL-13 (100 ng/ml) for the indicated days. Extracts were subsequently subjected to western blot analyses using the indicated antibodies (F). A quantification of Atf4 relative expression (Atf4/β-Actin ratio) from several experiments is illustrated (paired Student *t*-test, mean values ± SEM; $n = 3$; *$p < 0.05$, **$p < 0.01$). Anti-Dclk1 immunofluorescence analyses were also carried out to quantify the number of Dclk1$^+$ cells (in red) in all experimental conditions (mean values ± SEM; Mann–Whitney test; $n \geq 16$ organoids; *$p < 0.05$, ****$p < 0.0001$) (G). Source data are available online for this figure.

incubated at 37 °C under agitation for 20 minutes in HBSS supplemented with calcium and magnesium, collagenase D (500 µg/mL), DNase I (100 µg/mL), dispase (0.5 U/mL) and 2% FBS. The resulting tissue digest was then vortexed for 20 seconds at maximal speed, and then filtered through 70-µm cell strainers. The remaining pieces of the intestine were further incubated in the digestion solution. The obtained single-cell suspensions were then centrifuged for 5 minutes at 1200 rpm at 4 °C and cells were counted before being processed for flow cytometry analyses.

## Lung isolation

To isolate leukocytes from the lungs, the vena cava was sectioned and lungs were perfused with 5 ml of ice-cold PBS through the right ventricle. Lungs were incubated in HBSS (Gibco), 5% FBS, 50 µg/mL Liberase TM, 50 µg/mL DNase I and dissociated with the gentle MACS dissociator (Miltenyi Biotec) in C-tubes (Miltenyi Biotec). Lungs were incubated for 45 minutes at 37 °C and further dissociated with the gentle MACS dissociator. The resulting suspension was centrifuged at 1200 rpm for 5 minutes at 4 °C and washed in cold PBS with 2 mM EDTA. Cell suspensions were filtered on a 100 µm cell strainer (Falcon) and centrifuged. Erythrocytes were then lysed in red cell lysis solution (155 mM $NH_4Cl$, 0.12 mM EDTA, 10 mM $KHCO_3$) for 2 minutes at room temperature. Cells were washed with PBS and counted before being processed for flow cytometry analyses.

## Flow cytometry

Incubations were performed in FACS buffer (PBS containing 0.5% BSA and 0.1% $NaN_3$) at 4 °C. Cells were first incubated with anti-mouse CD16/32 antibody (clone 93, BioLegend) before fluorochrome-conjugated antibodies against surface antigens were added and incubated during 20 minutes at 4 °C. Antibodies to CD3ε (clone 145-2C11, APC), F4/80 (clone BM8, APC), FcεR1 (clone MAR1, APC), CD11c (clone N418, APC), Siglec-F (clone S17007L, APC), CD4 (clone RM4-5, APC), CD8α (clone 53-6.7, APC), CD5 (clone 53-7.3, APC), CD49b (clone HMα2, APC), Gr-1 (clone RB6-8C5, APC), CD45 (clone 30-F11, PE-Cyanine 7), and Sca1 (W18174A, FITC) were obtained from Biolegend. Antibodies to B220/CD45R (clone RA3-6B2, APC), CD90.2 (clone 53-2.1, eF450), IL-13 (clone eBio13A, Alexa Fluor 488), and GATA3 (clone

TWAJ, Alexa Fluor 488) were obtained from eBioscience/Thermo Fisher Scientific. The anti-KLRG1 antibody (clone 2F1, BV711) was obtained from BD Biosciences. For intracellular staining, single-cell suspensions were stimulated for 3 hours at 37 °C in Iscove's Modified Dulbecco's Medium with 50 ng/mL phorbol 12-myristate 13-acetate (Sigma-Aldrich), 1 µg/mL ionomycin (Sigma-Aldrich), 25 µM monensin, and 5 µg/ml brefeldin A (Biolegend) and 2 mM β-mercaptoethanol (Thermo Fisher). Samples were analyzed on a BD LSR Fortessa X-20 flow cytometer (BD Biosciences).

## Single-cell RNA sequencing

Mice were treated with rIL-13 overnight. A total of two mice were used per condition. A 2-cm-long fragment of the proximal intestine was extracted from mice, washed with PBS, and cut longitudinally. This fragment was incubated for 20 minutes at 4 °C in a DPBS buffer supplemented with 30 mM EDTA and 1.5 mM DTT, then incubated for 10 minutes at 37 °C in a DPBS-EDTA buffer (30 mM). Cells were harvested, centrifuged at $300 \times g$ for 5 minutes at 4 °C, and washed with DPBS-FBS 10%. After centrifugation, samples were dissociated into single-cell suspensions by incubation in HBSS-dispase (0.8 mg/mL) for 12 minutes at 37 °C. Single-cell solutions were sequentially passed through 70- and 40-µm strainers. Red blood cells were lysed (#130-094-183, Miltenyi Biotec) for 2 minutes at room temperature. Cells were counted and 10,000 cells were sent for single-cell RNA sequencing experiments. For library preparation and sequencing, 10,000 cells were injected for each sample in four different reactions into the Chromium Single Cell Controller following the protocol of the Chromium NextGEM Single Cell 3' V3.1 Kit (10x Genomics). The protocol was followed according to the manufacturer's guidelines. The quality control of the DNA library was performed on Qiaxcel (QIAgen) and quantification by qPCR (KAPA SYBR Fast & LQ Stds+Primer (Illumina). All Single cell 3' Gene expression libraries were sequenced on a Novaseq 6000 System (Illumina) on an S1 flowcell. For bioinformatic analyses, we used the 10X genomics *CellRanger* software (v6.0.1 - https://support.10xgenomics.com/Single-cell-geneexpression/software/overview/welcome) to process the data after sequencing. CellRanger mkfastq pipeline was used to demultiplex raw base call (BCL) files generated by Illumina sequencers into FASTQ files. CellRanger count pipeline was then used to perform alignment, filtering, barcode counting, and UMI

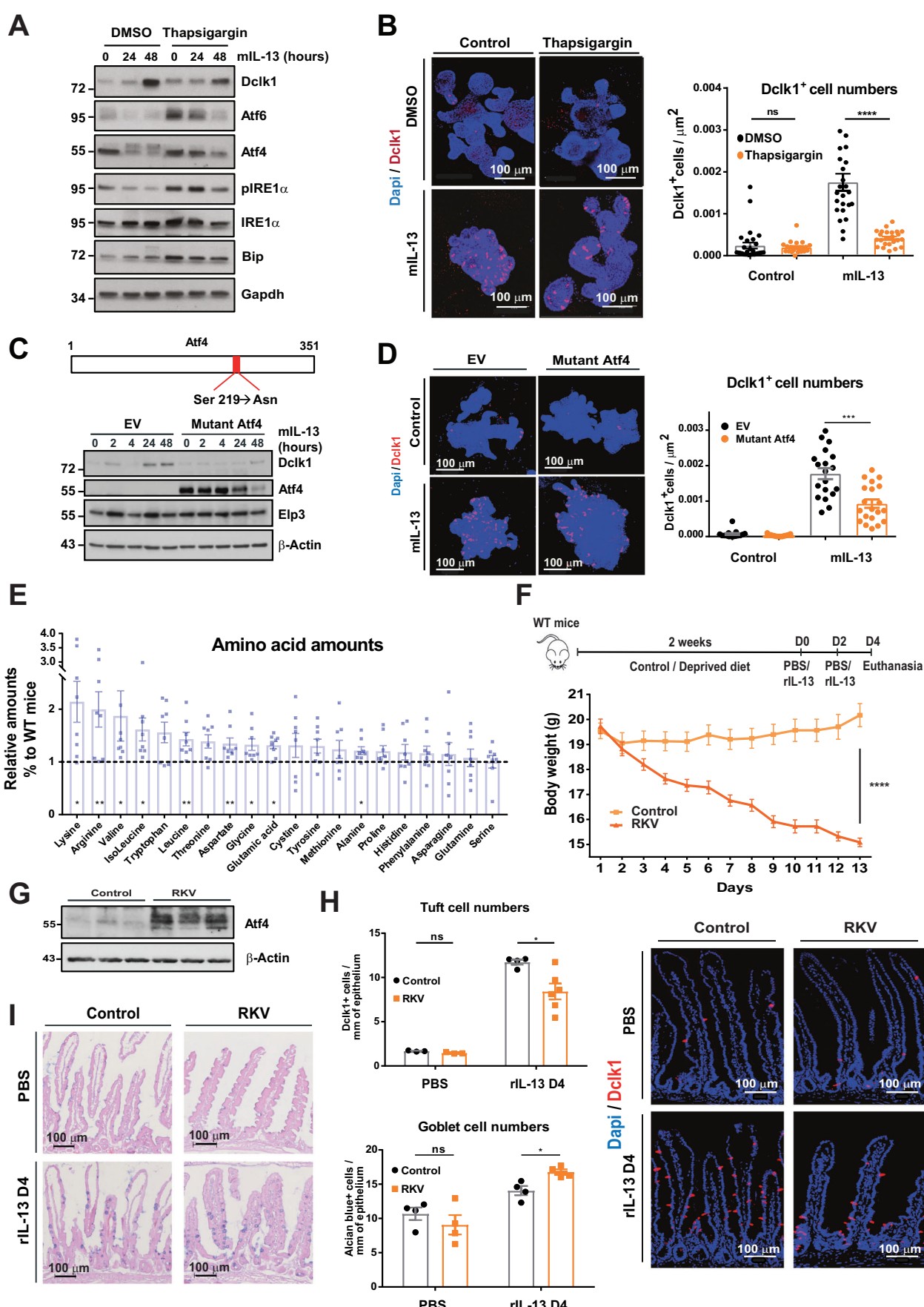

**Figure 9. Atf4 induction reduces intestinal tuft cell differentiation.**

(A, B) UPR activation impairs tuft cell differentiation in ex-vivo organoids. The UPR inducer Thapsigargin blocks IL-13-dependent Dclk1 induction, as evidenced by western blot (A) and immunofluorescence analyses (B). Expression levels of some UPR effectors, Dclk1 and Gapdh, in ex-vivo organoids generated with intestinal crypts from Elp3$^{WT}$ mice treated with mIL-13 and with Thapsigargin or with the vehicle (DMSO) are illustrated (A). Immunostainings and corresponding quantifications (left and right panels, respectively) of Dclk1$^+$ cells (in red) in WT organoids treated with mIL-13 and with Thapsigargin or with the vehicle (mean values ± SEM; Mann–Whitney test; $n \geq 22$ organoids from $n \geq 3$ mice; ****$p < 0.0001$) are illustrated (B). (C, D) Overexpression of a non-degradable Atf4 mutant inhibits tuft cell differentiation. A mutated form of Atf4 (serine 219 was replaced by an asparagine) was overexpressed in WT organoids (C, D). Expression levels of Dclk1, Atf4, Elp3 and β-Actin in WT organoids overexpressing or not (Empty vector, EV) mutated Atf4 at indicated time points post-mIL-13 treatments were assessed by western blot analyses (C). Immunostainings and corresponding quantifications (left and right panels, respectively) of Dclk1$^+$ cells (in red) in WT organoids overexpressing or not mutated Atf4 and treated or not with mIL-13 (mean values ± SEM; Mann–Whitney test; $n \geq 15$ organoids; ***$p < 0.001$) are illustrated (D). (E) Enhanced levels of some amino acids in intestines lacking Elp3. A metabolomics analysis with extracts from small intestines of naive Elp3$^{WT}$ and Elp3$^{\Delta IEC}$ mice was conducted (mean values + SEM; Mann–Whitney test; $n > 5$; *$p < 0.05$, **$p < 0.01$). (F) Amino acid deprivation induces weight loss in mice. WT mice received a control or a RKV-deprived diet for 2 weeks and were injected with rIL-13 for 4 days before the end of the experiment. Body weight was monitored every day (mean values + SEM; two-way ANOVA test; $n = 8$; ****$p < 0.0001$). (G) Amino acid deprivation induces Atf4 expression. Mice were fed with a control or a RKV-deprived diet for 2 weeks and extracts were subjected to western blot analyses using the indicated antibodies. (H, I) Amino acid deprivation inhibits tuft but not goblet cell differentiation. Immunostainings and corresponding quantifications (left and right panels, respectively) of tuft cells (Dclk1$^+$ cells, in red) (H) or goblet cells (Alcian Blue$^+$ cells) (I) in the small intestine of WT mice after 2 weeks of control or RKV-deprived diet and 4 days of PBS/rIL-13 treatment (mean values ± SEM; Mann–Whitney test; $n \geq 3$ for (H) and $n = 4$ for (I); *$p < 0.05$) are shown. Source data are available online for this figure.

counting for each sample. Alignment was performed using the mice reference genome *GRCm38* with the gene set annotations from Ensembl release 97 (Ensembl.org) for gene quantification. Downstream analyses were performed in the R environment using the Seurat package from Satija's lab (https://satijalab.org/seurat/) (Hao et al, 2021). For each sample, we kept genes expressed in a minimum of ten cells and cells expressing more than 200 genes were selected. Additionally, cells whose percentage of counts originating from mitochondrial genes is higher than 25% were discarded. We used the classical method from Seurat to analyse each dataset, and we performed dataset integration using the SCT method (Hafemeister and Satija, 2019) from Seurat. Dataset were first annotated independently and again after integration. Differentially expressed genes were identified using FindMarkers/FindAllMarkers functions from the Seurat package, while plots were produced using a combination of the Seurat pre-build functions and self-written functions using the ggplot2 package. The annotation of datasets was performed using a combination of methods: the identification and investigation of markers coming from different clusters defined in each dataset; the evaluation of known cell-type signatures by creating and visualizing associated expression scores on the datasets; the evaluation of specific molecular functions / pathways by creating and visualizing associated expression score on datasets. Our single-cell RNA Sequencing data were deposited with the following accession number: GSE241625.

## Gene set enrichment analysis

GSEA was performed to determine gene sets (minimal size = 7 genes) involving the differentially expressed genes identified between Elp3$^{WT}$ and Elp3$^{\Delta IEC}$ mice. The statistically significant enrichment of gene set was selected based on a false discovery rate (FDR) ≤0.25 and a normalized enriched score (NES) was calculated for each gene set. Heatmaps were used based on GSEA analysis in order to visualize the expression levels of genes from these signatures across cell populations. In Figs. 5B, 6F showing signatures of oxidative phosphorylation and mTORC1-Atf4, respectively, only genes that were significantly up- or down-regulated between Elp3$^{\Delta IEC}$ mice compared to Elp3$^{WT}$ mice in one or more of the represented populations were selected (based on $p$_val_adj <0.05 and logFC >0.25). Differentially expressed genes

were identified using FindMarkers/FindAllMarkers functions from the Seurat package. The statistical test done was the Wilcoxon rank-sum test.

## Polysome profiling

mIECs were extracted from mice as usual, except that CHX (Cycloheximide) (0.1 mg/mL) was added in every step of the isolation process. Cells were then lysed in hypotonic buffer (2.5 mM MgCl$_2$, 5 mM TRIS pH 7.5, 1.5 mM KCl supplemented with protease inhibitors). To solubilize both cytosolic and endoplasmic reticulum-associated ribosomes, sequential addition of CHX, DTT, RNAse inhibitor, Triton X-100, and sodium deoxycholate was performed. After centrifugation, the concentration of cytosolic lysates was measured and similar amounts were loaded onto a non-linear sucrose gradient (5–34–55%) and ultracentrifuged at 220,000×$g$ for 2 hours at 4 °C. Approximately 10% of the cytosolic lysate was further kept for total RNA extraction (input). The different fractions were collected using a piston gradient fractionator (BioComp). Input, as well as gradient fractions corresponding to polysomal (efficiently translated mRNA; associated with >3 ribosomes), were then lysed in a Trizol solution (TriPure Isolation Reagent, #11667165001, Sigma). The RNA dosage was performed using the Ribogreen™ RNA assay and cDNAs were used for qPCR analyses. The ratio of polysomal to input fraction was calculated in all experimental conditions.

## Ex-vivo organoid cultures

Mouse intestines were opened longitudinally, washed with cold PBS and cells from villi were scraped with a coverslip. Tissue was cut into small pieces, washed five to ten times with cold PBS, and incubated in 2 mM EDTA-PBS for 30 minutes at 4 °C. After removal of PBS-EDTA, cold PBS-FBS 10% was added and tissue was vigorously suspended with a 10-ml pipette to obtain a crypt-enriched fraction. This fraction was centrifuged at 600 rpm at 4 °C for 5 minutes. Crypts were washed with cold PBS and passed through a 70-µm cell strainer. After a second centrifugation, crypts were suspended in Matrigel (#354230, Corning) and cultured in a 24-well plate. IntestiCult™ Organoid Growth Media (#06005, Stemcell) was added and fresh media was changed every 4 days. Organoids were passaged every 7 days and the first experiments

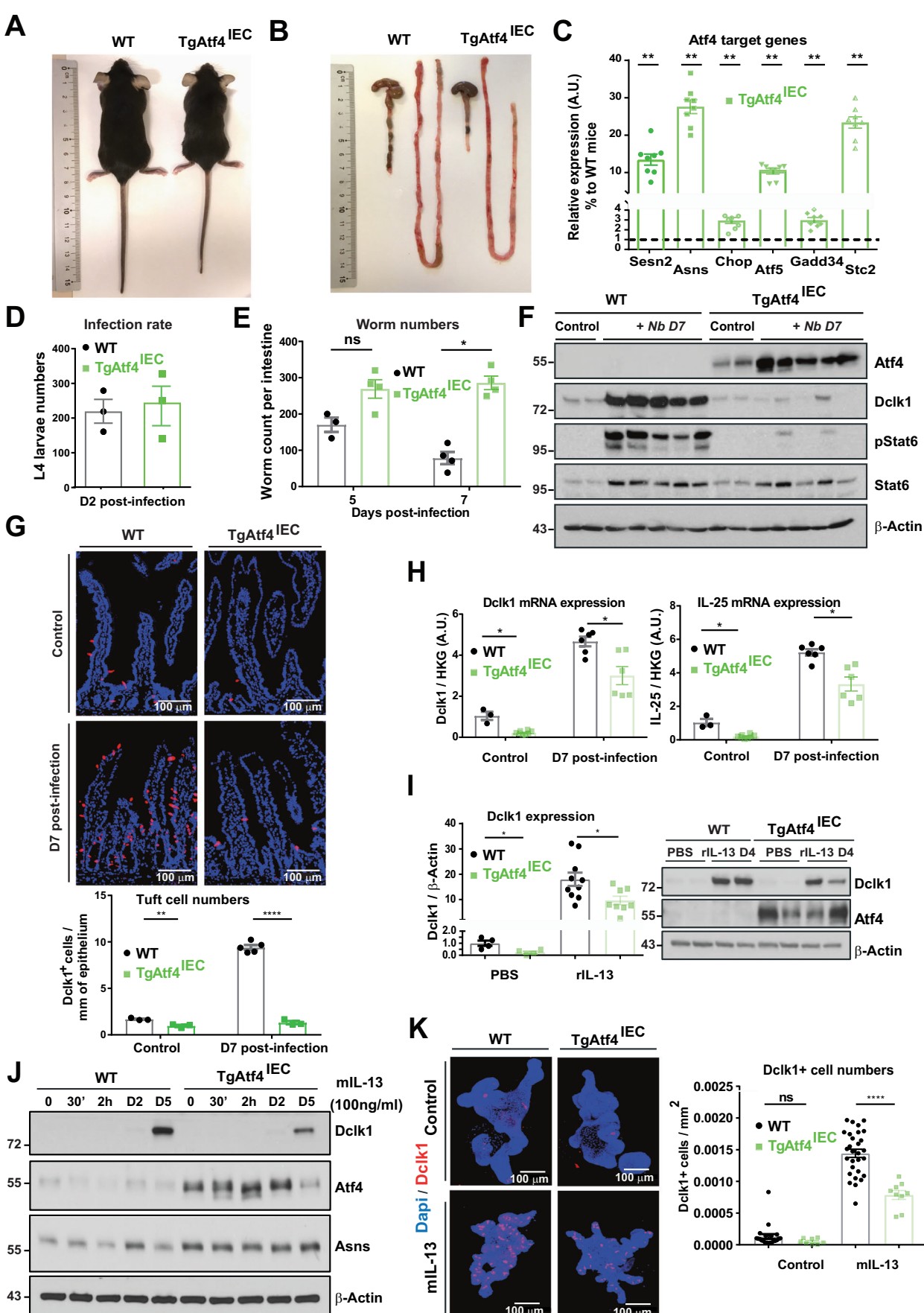

**Figure 10. Atf4 overexpression in mouse IECs blocks tuft cell differentiation.**

(A) TgAtf4$^{IEC}$ mice are smaller than WT mice. Representative pictures of both WT and TgAtf4$^{IEC}$ mice are illustrated. (B) Atf4 overexpression in intestinal epithelium shortens the length of both small intestines and colons. Representative organs of both genotypes are illustrated. (C) TgAtf4$^{IEC}$ mice overexpress some Atf4 target genes. mRNA levels of Atf4 target genes in the small intestine of naive WT and TgAtf4$^{IEC}$ (ratio to WT mice) were quantified by Real-Time PCRs. Normalization was calculated on the average of the two housekeeping genes Gapdh and 36b4 (mean values ± SEM; Mann–Whitney test; $n \geq 4$; **$p < 0.01$). (D) WT and TgAtf4$^{IEC}$ mice share similar infection rates in the lung. The infection rate assessed by L4 larvae counts in lungs at day 2 after *N. brasiliensis* infection is illustrated (mean values ± SEM; Mann–Whitney test; $n = 3$). (E) Delayed parasite clearance in TgAtf4$^{IEC}$ mice infected with *N. brasiliensis*. The worm burden across the entire small intestine of both WT and TgAtf4$^{IEC}$ at the indicated time points post-*N. brasiliensis* infection was quantified (mean values ± SEM; Mann–Whitney test; $n \geq 3$; *$p < 0.05$). (F–H) Atf4 overexpression in the intestine interferes with tuft cell differentiation. Both WT and TgAtf4$^{IEC}$ mice were infected or not with *N. brasiliensis*. 7 days post-infection, extracts from the resulting small intestines were subjected to western blot (F), IF (G), or real-time PCR analyses (H), respectively. Immunostainings and corresponding quantifications (upper and lower panels, respectively) of tuft cells (Dclk1$^+$ cells, in red) in the small intestine of Elp3$^{WT}$ and TgAtf4$^{IEC}$ mice naive or 7 days post-*N. brasiliensis* infection are shown (mean values ± SEM; Student $t$-test; $n \geq 3$; **$p < 0.01$, ****$p < 0.0001$) (G). For real-time PCRs, Dclk1 and IL-25 mRNA levels were quantified, and normalization was calculated on the average of two housekeeping genes Gapdh and 36b4 (mean values ± SEM; Mann–Whitney test; $n \geq 3$; *$p < 0.05$) (H). (I) Tuft cell differentiation is specifically impaired in TgAtf4$^{IEC}$ mice upon rIL-13 treatment. Mice of the indicated genotypes were treated or not with rIL-13 for 4 days, and the resulting extracts from small intestines were subjected to western blot analyses (right panels). A quantification of Dclk1 relative expression (Dclk1/β-Actin ratio) from several experiments is illustrated (mean values ± SEM; Mann–Whitney test; $n \geq 4$; *$p < 0.05$). (J, K) Atf4 overexpression in ex-vivo organoids impairs IL-13-dependent Dclk1 induction. Ex-vivo organoids generated with intestines from mice of the indicated genotypes were treated or not with mIL-13 for 30 minutes to 5 days, and the resulting extracts were subjected to western blot analyses (J) or to anti-Dclk1 (in red) immunofluorescence analyses (K). A quantification of Dclk1$^+$ cells in all experimental conditions is illustrated (right panel) (mean values ± SEM; Mann–Whitney test; $n \geq 8$ organoids from $n \geq 2$ different mice; ****$p < 0.0001$). Source data are available online for this figure.

were done after ten passages to be sure to obtain a pure epithelial culture. When indicated, cultures were treated with recombinant murine IL-13 (100 ng/ml, #413-ML, R&D Systems), Isrib (10 µM, #SML0843, Sigma-Aldrich) and Thapsigargin (200 nM, #SML1845, Sigma-Aldrich). Lentiviral-based organoid infection was performed, as previously described (Maru et al, 2016). Infected organoids were selected upon puromycin treatment (2 µg/mL, ant-pr-1, InvivoGen). All reagents used in this study are listed in Table EV2.

## Immunofluorescence and immunohistochemistry analyses

Immunohistochemistry and immunofluorescence stainings were performed on cryosections. Intestinal tissues were fixed overnight in a 4% formaldehyde solution and incubated overnight in 20% sucrose. After a second incubation overnight in 30% sucrose, tissues were frozen in Tissue-Tek O.C.T. 5-µm-thick sections were permeabilized and blocked in PBS supplemented with 0.2% Triton X-100 and 3% BSA for 1 hour. After incubation overnight with primary antibody at 4 °C, sections were washed and incubated 1 hour at room temperature in secondary antibody, stained with DAPI and mounted. For Dclk1-IL-13Rα1 co-stainings, sections were first stained for Dclk1. Sections were subsequently fixed again in 4% formaldehyde, permeabilized, blocked, and stained for IL-13Rα1. Both stainings were also performed separately to confirm their specificity and to exclude any possible nonspecific reactions. For pHH3, Ki67, pSTAT6, and ChgA stainings, frozen sections were first incubated in antigen retrieval solution (Dako) at 95 °C for 30 minutes. For ex-vivo organoid stainings, organoids were cultured in eight-well chamber slides (#354118, Falcon). After treatment, organoids were fixed for 20 minutes in 4% formaldehyde solution, permeabilized for 20 minutes in PBS with 0.5% Triton X-100, and blocked for 30 minutes in PBS with 0.2% Triton X-100, 0.1% Tween, and 1% BSA. After incubation overnight with primary antibody at 4 °C, organoids were washed and incubated 1 hour at room temperature with the secondary antibody. For Paneth cell detection, UEA1 (Sigma, 1 mg/ml stock) was added at 1:200 dilution with secondary antibodies. After further washes, organoids

were stained with DAPI and mounted. For intestine stainings, $n \geq 4$ pictures per mice were acquired, and the average quantification was shown as a representative dot in the graph. For organoids, each organoid was represented in the graph. Slides were analyzed on a Nikon A1R and Leica SP5 microscope.

## Western blots

Snap-frozen intestinal epithelial cells and organoids were lysed in 1% SDS lysis buffer at 98 °C for 10 minutes. Protein concentrations were quantified by using a BCA protein kit (Thermo Fisher). Protein samples were analyzed on SDS polyacrylamide gel electrophoresis (SDS–PAGE) and transferred onto nitrocellulose membranes (#10600002, Cytiva). These membranes were blocked with 10% milk and 0.1% Tween 20 in Tris-buffered saline for 1 hour and incubated overnight at 4 °C with primary antibodies. The appropriate HRP-coupled secondary antibody was then added and was detected with chemiluminescent substrate ECL (Thermo Fisher). The antibodies used in this study are listed in the Table EV2.

## Targeted LC-MS metabolomics analyses

For metabolomic analyses, the extraction solution was composed of 50% methanol, 30% acetonitrile (ACN), and 20% water. The volume of the extraction solution was adjusted to a cell number (1 ml per $10^6$ cells). After the addition of the extraction solution, samples were vortexed for 5 minutes at 4 °C and centrifuged at 16,000×g for 15 minutes at 4 °C. The supernatants were collected and stored at −80 °C until analysis. LC/MS analyses were conducted on a QExactive Plus Orbitrap mass spectrometer equipped with an Ion Max source and a HESI II probe coupled to a Dionex UltiMate 3000 uHPLC system (Thermo). External mass calibration was performed using a standard calibration mixture every 7 days, as recommended by the manufacturer. The 5 µl samples were injected onto a ZIC-pHILIC column (150 mm × 2.1 mm; i.d. 5 µm) with a guard column (20 mm × 2.1 mm; i.d. 5 µm) (Millipore) for LC separation. Buffer A was 20 mM ammonium carbonate, 0.1% ammonium hydroxide (pH 9.2), and buffer B was ACN. The chromatographic gradient was run at a flow rate of 0.200 µl min$^{-1}$ as follows: 0–20 minutes, linear

gradient from 80 to 20% of buffer B; 20–20.5 minutes, linear gradient from 20 to 80% of buffer B; 20.5–28 minutes, 80% buffer B. The mass spectrometer was operated in full scan, polarity switching mode with the spray voltage set to 2.5 kV and the heated capillary held at 320 °C. The sheath gas flow was set to 20 units, the auxiliary gas flow to 5 units, and the sweep gas flow to 0 units. The metabolites were detected across a mass range of 75–1000 $m/z$ at a resolution of 35,000 (at 200 $m/z$) with the automatic gain control target at $10^6$ and the maximum injection time at 250 ms. Lock masses were used to ensure mass accuracy below 5 ppm. Data were acquired with Thermo Xcalibur software (Thermo). The peak areas of metabolites were determined using Thermo TraceFinder software (Thermo), identified by the exact mass of each singly charged ion and by the known retention time on the HPLC column.

### RNA extractions and real-time PCRs

Total RNAs from ex-vivo intestinal organoids were isolated using an RNeasy mini kit (QIAGEN) according to the manufacturer's protocol. Total RNAs from snap-frozen intestinal epithelial cells were isolated using a Trizol-chloroform extraction. cDNAs were synthesized using the Revert Aid H Minus First Strand cDNA Synthesis kit (Fermentas). Subsequent PCRs were performed using the SYBR Green (Sybr Premix Taq, ROX Plus; Takara Bio Inc.) on the LightCycler 480 (Roche). Depending on the experiment, normalization was calculated on the average of 2 to 4 housekeeping genes (Gapdh, Actin, 36b4, and β2m—detailed in corresponding legends) and relative expression was obtained using the ΔΔCt method. All primer sequences used in this study are listed in Table EV2.

### Statistical analyses

The ImageJ software was used for both western blot quantification and immunofluorescence countings. The NDP.View2 software was used for immunohistochemistry countings. The GSEA 4.1.0 software was used for Gene Set Enrichment Analyses. The R software was used for single-cell RNA sequencing analyses. Graphs and statistical analyses were performed using the GraphPad Prism 7.0 Software. All data were presented as the mean and standard error of the mean (mean ± SEM), and each experiment was performed at least in triplicates. Depending on the experiment, the Student $t$-test or Mann–Whitney test was used for comparison between both groups. The two-way ANOVA test was used to compare the mean of a continuous variable between two groups. $p < 0.05$ was considered statistically significant. $*p < 0.05$; $**p < 0.01$; $***p < 0.001$; $****p < 0.0001$. Blinding was systematically done.

## Data availability

Our single-cell RNA sequencing data were deposited with the following accession number: GSE241625.

The source data of this paper are collected in the following database record: biostudies:S-SCDT-10_1038-S44318-024-00184-4.

## Peer review information

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

## Acknowledgements

The authors are grateful to Dr. Thomas Marichal (GIGA, University of Liege, Liege, Belgium) for the gift of the immortalized mouse intestinal epithelial cell line and to the GIGA Imaging platform for IF analyses. This work was supported by grants from WELBIO (WELBIO-CR-2019C-02R), FNRS/TELEVIE, the Belgian Foundation against Cancer (FBC/2020/1323), the University of Liege (ARC UBICOREAR) and by the Leon Fredericq Foundation (Faculty of Medicine, CHU of Liege). NEH, CM, PC, and AC are Senior Research Assistants, Senior Research Associate, and Research Director at the FNRS, respectively.

## Author contributions

**Caroline Wathieu**: Conceptualization; Data curation; Formal analysis; Validation; Investigation; Methodology. **Arnaud Lavergne**: Conceptualization; Data curation; Formal analysis; Validation; Visualization. **Xinyi Xu**: Data curation; Formal analysis. **Marion Rolot**: Data curation; Formal analysis. **Ivan Nemazanyy**: Conceptualization; Data curation; Formal analysis; Validation. **Kateryna Shostak**: Data curation; Formal analysis; Validation; Investigation. **Najla El Hachem**: Data curation; Formal analysis. **Chloé Maurizy**: Data curation; Formal analysis. **Charlotte Leemans**: Data curation; Formal analysis. **Pierre Close**: Resources; Formal analysis; Methodology. **Laurent Nguyen**: Resources; Formal analysis. **Christophe Desmet**: Resources; Formal analysis. **Sylvia Tielens**: Conceptualization; Data curation; Formal analysis; Supervision; Validation; Investigation; Visualization; Methodology; Writing—review and editing. **Benjamin G Dewals**: Conceptualization; Resources; Data curation; Formal analysis; Supervision; Funding acquisition; Validation; Investigation; Methodology; Writing—review and editing. **Alain Chariot**: Conceptualization; Resources; Formal analysis; Supervision; Funding acquisition; Validation; Investigation; Visualization; Methodology; Writing—original draft; Project administration; Writing—review and editing.

Source data underlying figure panels in this paper may have individual authorship assigned. Where available, figure panel/source data authorship is listed in the following database record: biostudies:S-SCDT-10_1038-S44318-024-00184-4.

## Disclosure and competing interests statement

The authors declare no competing interests.

# Expanded View Figures

**Figure EV1.  Intestinal homeostasis is unaffected upon *Elp3* inactivation.**

(**A**) Intestines lacking Elp3 properly proliferate. Immunolabelings of intestinal sections from mice of the indicated genotypes and infected or not with *N. brasiliensis* are illustrated (KI67 and pHH3 stainings are in red and white, respectively). On the right, histograms show corresponding quantifications ($n = 5$ mice per genotype, mean value ± SEM; Mann–Whitney test shows no significant difference between Elp3$^{WT}$ and Elp3$^{\Delta IEC}$ mice). (**B**) *Elp3* deficiency impairs the induction of FFAR3. Mice of the indicated genotypes were infected with *N. brasiliensis* and protein extracts from intestines 7 days post-infection were subjected to western blot analyses to assess FFAR3 and β-actin expression. (**C**, **D**) Defa5 and ChgA are properly expressed in intestinal epithelial cells lacking Elp3. Mice of the indicated genotypes were infected or not with *N. brasiliensis* and RNAs from the resulting intestines were subjected to real-time PCR analyses to quantify both Defa5 (**C**) and ChgA (**D**) mRNA levels. Expression levels of these candidates were normalized on the average of four housekeeping genes, including Gapdh, β-Actin, 36B4 and β2-Microglobulin ($n \geq 6$ mice per genotype; mean ± SEM; a Mann–Whitney test shows no significant differences between Elp3$^{WT}$ and Elp3$^{\Delta IEC}$ mice). Source data are available online for this figure.

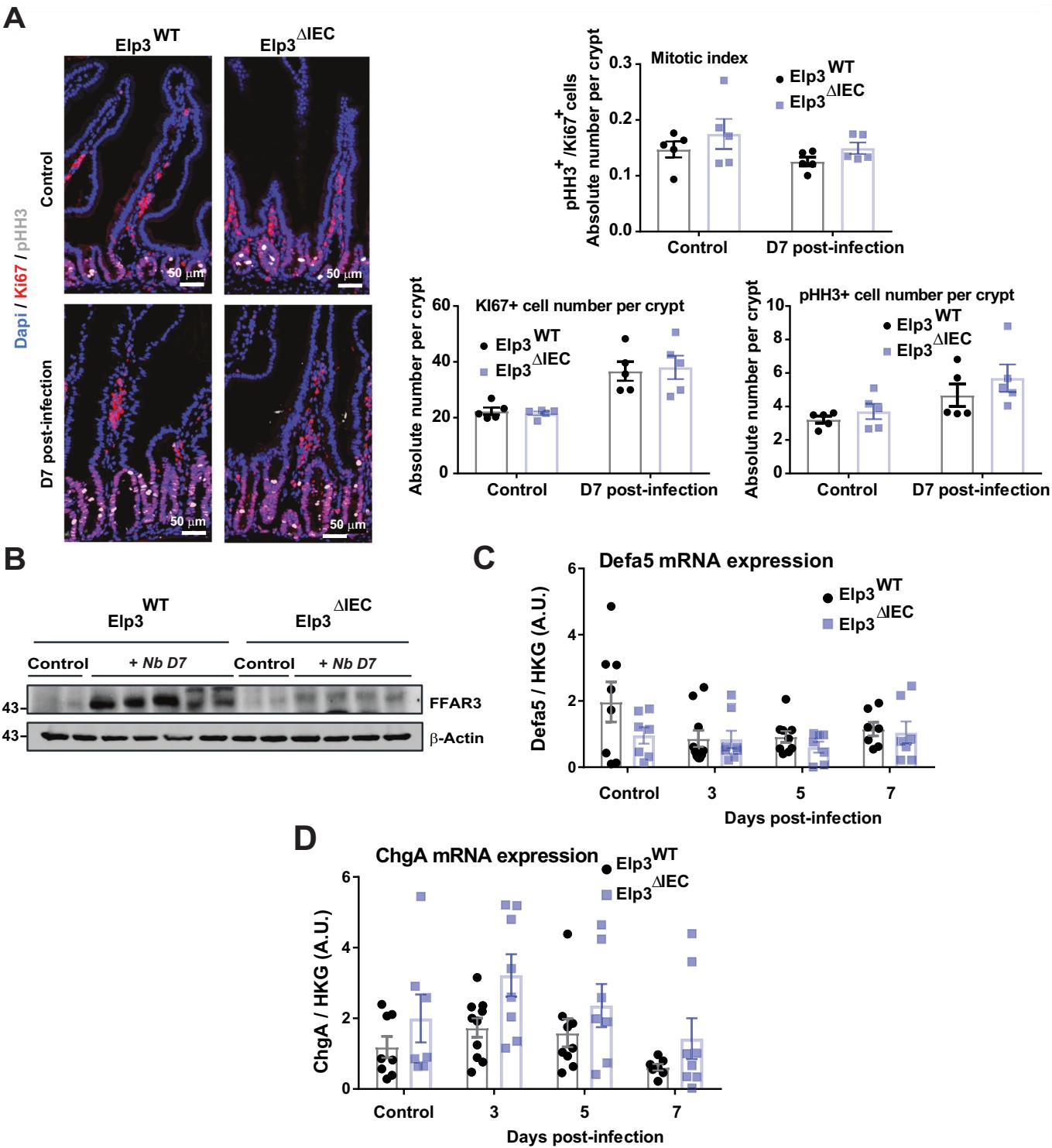

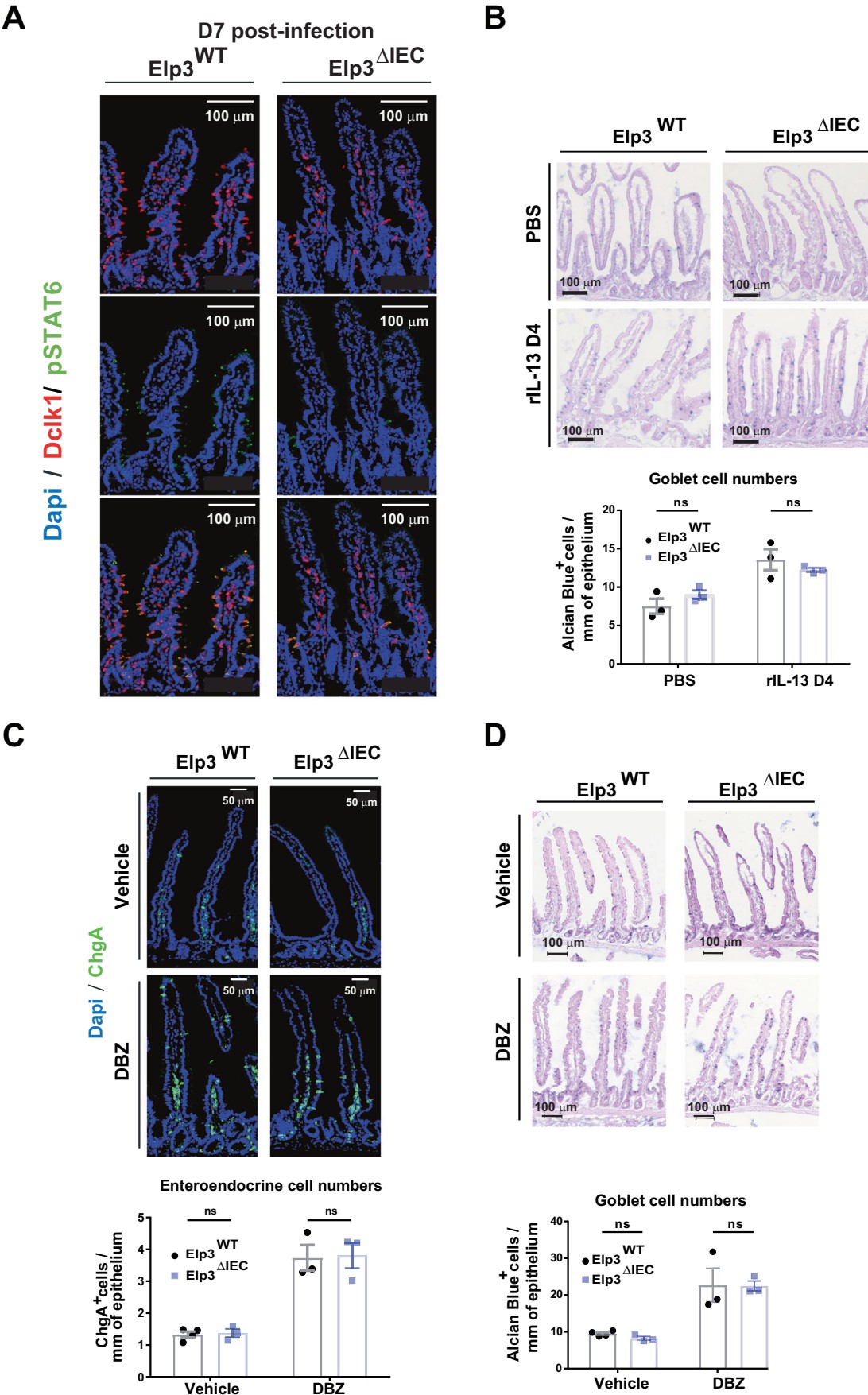

◄ **Figure EV2. Elp3 is dispensable for the expansion of both enteroendocrine and goblet cells.**

(**A**) Mice of the indicated genotypes were infected with *N. brasiliensis* and immunofluorescence analyses in the intestine were conducted 7 days post-infection to detect tuft cells (Dclk1+ cells in red) as well as Stat6 phosphorylation (in green). (**B–D**) Mice of the indicated genotypes were treated or not with recombinant IL-13 (rIL-13) for 4 days (**B**) or with the Notch inhibitor DBZ (**C, D**), and the resulting intestines were subjected to immunohistochemistry analyses to quantify goblet cells (Alcian Blue+ cells) (**B, D**) or enteroendocrine cells (Chromogranin A+ cells (**C**) (top panels). At the bottom, a quantification is provided (mean values ± SEM; Mann–Whitney test, $n = 3$ and $n \geq 3$ for IL-13 and DBZ treatments, respectively). Source data are available online for this figure.

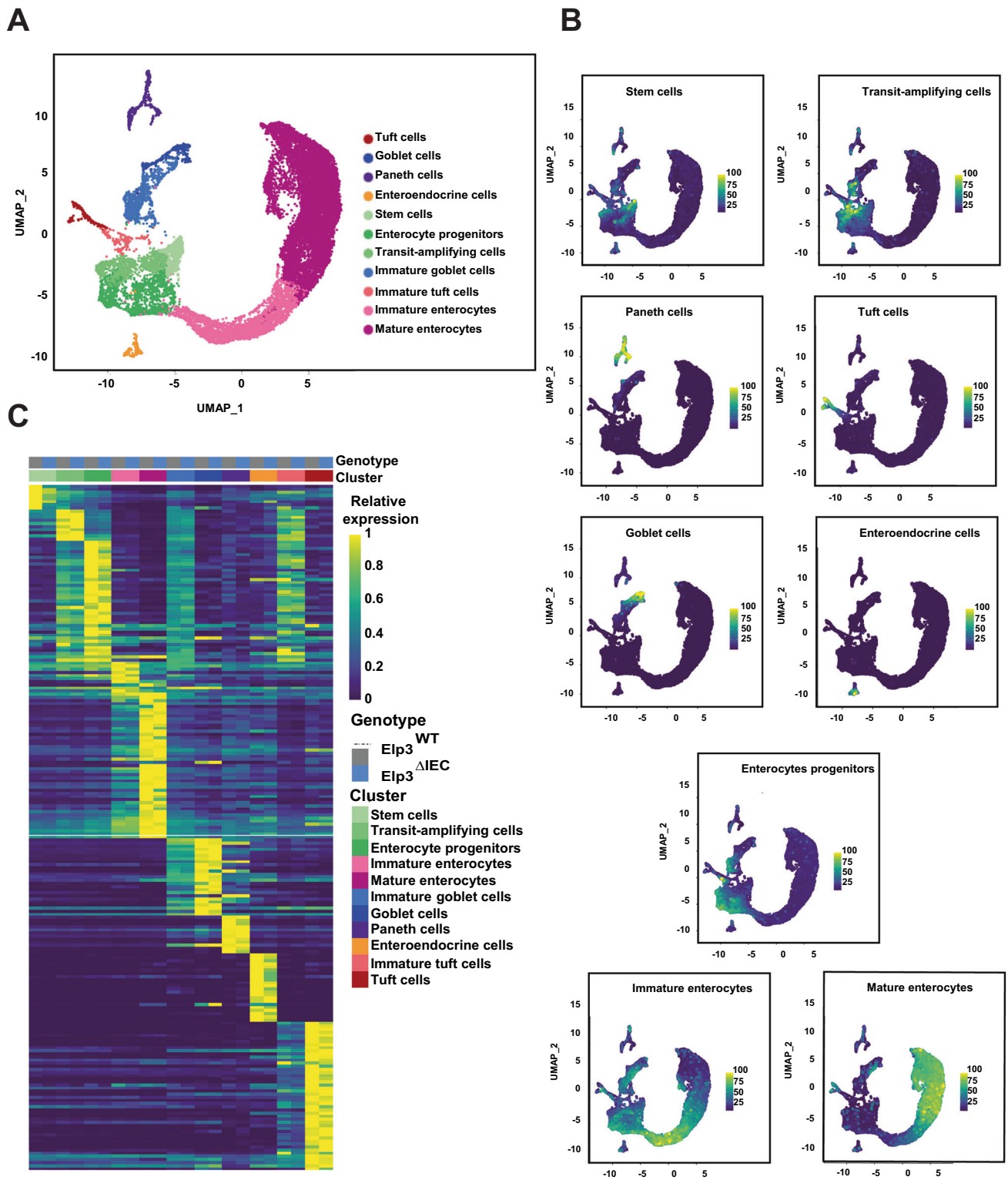

**Figure EV3. Single-cell RNA sequencing analysis of mouse intestinal epithelium.**

(A) Cell-type clustering. Two-dimensional graphical representation of the cell-type clustering in the small intestine of both Elp3$^{WT}$ and Elp3$^{\Delta IEC}$ mice after overnight rIL-13 treatment ($n = 4$ pooled mice). (B, C) Cell-type signatures in mouse intestinal epithelium. UMAP showing expression and distribution of representative genes in clusters from A (B). Heatmap of cluster marker genes in the small intestine of both Elp3$^{WT}$ and Elp3$^{\Delta IEC}$ mice after overnight rIL-13 treatment ($n = 2$ pooled mice per genotype) (C).

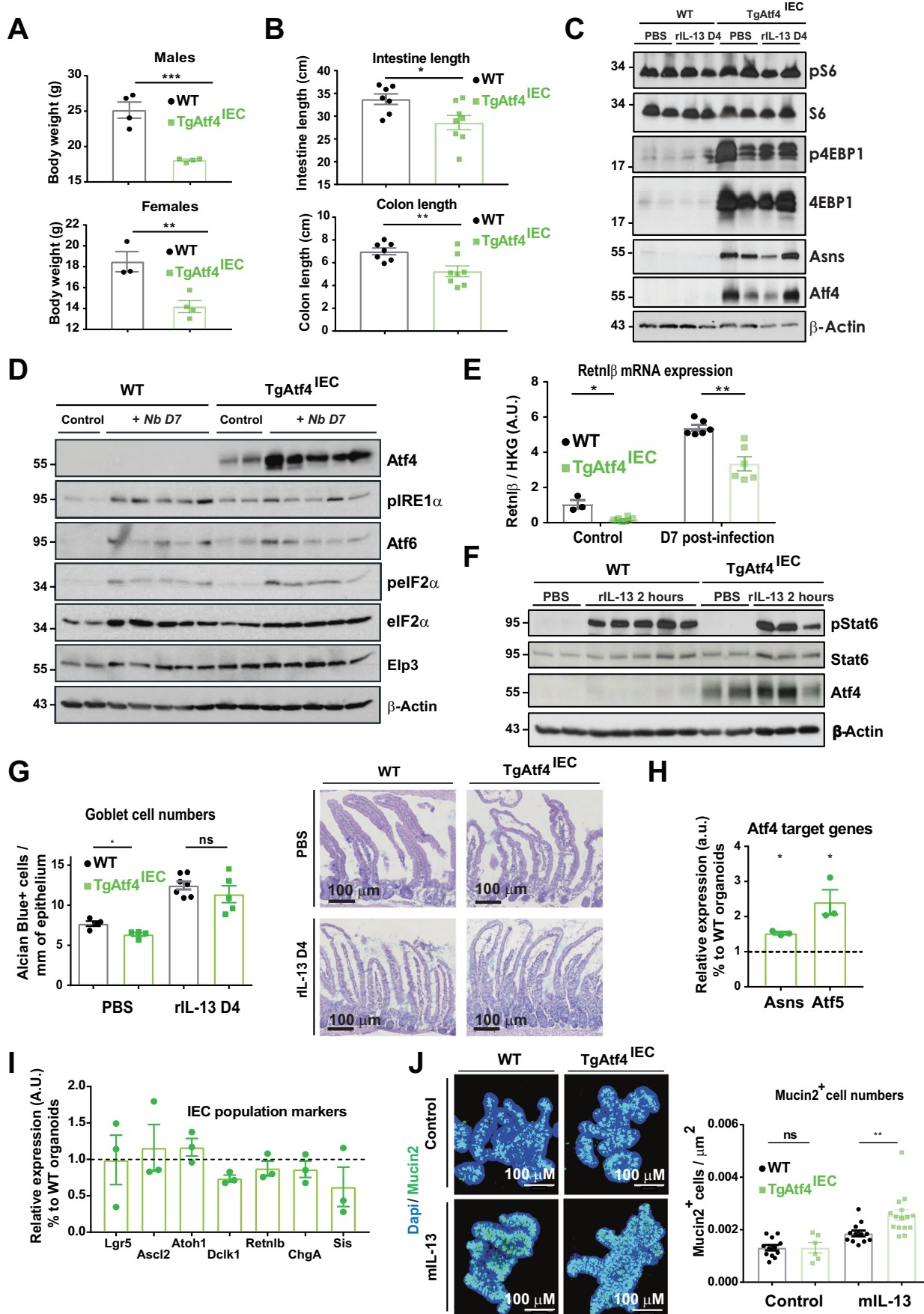

◄  **Figure EV4. Atf4 overexpression in mouse intestinal epithelium inhibits tuft cell differentiation.**

(A) Atf4 overexpression in IECs deregulates the body weight. A quantification of 8 weeks old WT and TgAtf4[IEC] mice body weight (mean values ± SEM; Student *t*-test; $n \geq 3$ for females and $n = 4$ for males; **$p < 0.01$, ***$p < 0.001$) is illustrated. (B) Atf4 overexpression in IECs shortens the length of both small intestines and colons. A quantification of both intestine and colon lengths in WT ($n = 7$) and TgAtf4[IEC] ($n = 8$) mice (mean values ± SEM; Student *t*-test; *$p < 0.05$, **$p < 0.01$) is illustrated. (C) Atf4 overexpression in the intestinal epithelium does not change S6 phosphorylation but enhances 4EBP1 protein levels. Extracts from the intestinal epithelium of both WT and TgAtf4[IEC] mice were subjected to western blot analyses. (D) Atf4 overexpression in IECs does not trigger the canonical UPR pathway upon helminth infection. Extracts from both WT and TgAtf4[IEC] mice naive or infected with *N. brasiliensis* for 7 days were subjected to western blot analyses using the indicated antibodies. Note that the anti-Atf4 blot is identical to the one illustrated in Fig. 10F. (E) Defective mRNA induction of Retnlβ upon helminth infection in Atf4-overexpressing IECs. Mice of the indicated genotypes were infected or not with *N. brasiliensis* for 7 days and total RNAs were subjected to real-time PCRs. Retnlβ mRNA levels were quantified and normalization was calculated on the average of the two housekeeping genes Gapdh and 36b4 (mean values ± SEM; Mann–Whitney test; $n \geq 3$; *$p < 0.05$, **$p < 0.01$). (F) IL-13-dependent Stat6 phosphorylation does not change upon Atf4 overexpression. WT and TgAtf4[IEC] mice were treated or not with rIL-13 for 2 hours, and the resulting extracts were subjected to western blot analyses. (G) IL-13-dependent goblet cell expansion is similar in intestines from WT and TgAtf4[IEC] mice. Immunostainings and quantifications (right and left panels, respectively) of goblet cells (Alcian Blue[+] cells) in intestines from WT and TgAtf4 mice treated or not with rIL-13 for 4 days (mean values ± SEM; Mann–Whitney test; $n \geq 3$ mice; *$p < 0.05$) are illustrated. (H, I) Atf4 overexpression in ex-vivo organoids induces the expression of Atf4 target genes but does not change levels of epithelial cell markers. mRNA levels of Atf4 target genes (Asns and Atf5) (H) and IEC population markers (I) in ex-vivo organoids extracted from WT and TgAtf4[IEC] mice (ratio to WT organoids) were quantified by real-time PCRs. Normalization was calculated on the average of the three housekeeping genes Gapdh, β-Actin and 36b4 (mean values ± SEM; Student *t*-test ; $n = 3$; *$p < 0.05$). (J) IL-13-dependent goblet cell expansion is not reduced in ex-vivo organoids from TgAtf4[IEC] mice. Immunostainings and quantifications (right and left panels, respectively) of goblet cells (Mucin2[+] cells in green) in ex-vivo organoids of WT and TgAtf4 mice treated or not with mIL-13 (mean values ± SEM; Mann–Whitney test; $n \geq 6$ organoids from $n = 2$ mice; **$p < 0.01$) are illustrated. Source data are available online for this figure.

