## [Peer Review File · The EMBO Journal]

Loss of Elp3 blocks intestinal tuft cell differentiation via an mTORC1-Atf4 axis

Caroline Wathieu, Arnaud Lavergne, Xinyi Xu, Marion Rolot, Ivan Nemazanyy, Kateryna Shostak, Najla El Hachem, Chloé Maurizy, Charlotte Leemans, Pierre Close, Laurent Nguyen, Christophe Desmet, Sylvia Tielens, Benjamin Dewals, and Alain Chariot

Corresponding author: Alain Chariot (alain.chariot@uliege.be)

Review Timeline:

Submission Date:	19th Sep 23
Editorial Decision:	14th Nov 23
Revision Received:	29th Mar 24
Editorial Decision:	10th May 24
Revision Received:	25th Jun 24
Accepted:	12th Jul 24

Editor: Daniel Klimmeck

Transaction Report:

Dear Alain,

Thank you again for the submission of your manuscript (EMBOJ-2023-115652) to The EMBO Journal and in addition discussing your views on the critique and providing a preliminary revision plan. As mentioned earlier, your study was assessed by three reviewers with expertise in tRNA biology and intestinal homeostasis, whose comments are enclosed below.

As you will see from their comments, the referees acknowledge the analysis and potential interest and value of your findings. However, referee #2 also expresses major concerns i.p. regarding the degree of mechanistic insights into the role of Elp3 and tRNA alterations in steering tuft cell differentiation in Npr12-TOR-ATF4-dependent manner (ref#2, pt.2). This referee also expresses substantial reservations on remaining ambiguities of Elp3's cell type- and tuft cell stage-specific roles in this context (ref#2, pt.1; see also ref#3, pt.3). Further, the reviewers raise a number of points related to the overall structure of the manuscript and presentation of the findings, additional controls required, improved methods annotation and data processing and overall discussion of related literature, that would need to be conclusively addressed to achieve the level of robustness and clarity needed for The EMBO Journal.

Given the overall interest stated and broader angle of your findings, we are able to invite you to revise your manuscript experimentally to address the referees' comments, along the lines sketched in your outline. I need to stress though that we do require strong support from the referees on a revised version of the study in order to move on to publication of the work.

Please feel free to contact me if you have any questions or need further input on the referee comments.

We generally allow three months as standard revision time. As a matter of policy, competing manuscripts published during this period will not negatively impact on our assessment of the conceptual advance presented by your study. However, we request that you contact the editor as soon as possible upon publication of any related work, to discuss how to proceed.

Should you foresee a problem in meeting this three-month deadline, please let me know in advance and I may be able to grant an extension.

When submitting your revised manuscript, please carefully review the instructions below.

Please feel free to approach me any time should you have additional questions related to this.

Thank you for the opportunity to consider your work for publication.

I look forward to your revision.

Best regards,

Daniel

Daniel Klimmeck, PhD
Senior Editor
The EMBO Journal

Instruction for the preparation of your revised manuscript:

2) individual production quality figure files as .eps, .tif, .jpg (one file per figure).

3) a .docx formatted letter INCLUDING the reviewers' reports and your detailed point-by-point response to their comments. As part of the EMBO Press transparent editorial process, the point-by-point response is part of the Review Process File (RPF), which will be published alongside your paper.

4) a complete author checklist, which you can download from our author guidelines ([https://wol-prod-cdn.literatumonline.com/pb-assets/embo-site/Author Checklist%20-%20EMBO%20J-1561436015657.xlsx](https://wol-prod-cdn.literatumonline.com/pb-assets/embo-site/Author%20Checklist%20-%20EMBO%20J-1561436015657.xlsx)). Please insert information in the checklist that is also reflected in the manuscript. The completed author checklist will also be part of the RPF.

6) It is mandatory to include a 'Data Availability' section after the Materials and Methods. Before submitting your revision, primary datasets produced in this study need to be deposited in an appropriate public database, and the accession numbers and database listed under 'Data Availability'. Please remember to provide a reviewer password if the datasets are not yet public (see <https://www.embopress.org/page/journal/14602075/authorguide#datadeposition>).

7) Our journal encourages inclusion of *data citations in the reference list* to directly cite datasets that were re-used and obtained from public databases. Data citations in the article text are distinct from normal bibliographical citations and should directly link to the database records from which the data can be accessed. In the main text, data citations are formatted as follows: "Data ref: Smith et al, 2001" or "Data ref: NCBI Sequence Read Archive PRJNA342805, 2017". In the Reference list, data citations must be labeled with "[DATASET]". A data reference must provide the database name, accession number/identifiers and a resolvable link to the landing page from which the data can be accessed at the end of the reference. Further instructions are available at .

8) At EMBO Press we ask authors to provide source data for the main and EV figures. Our source data coordinator will contact you to discuss which figure panels we would need source data for and will also provide you with helpful tips on how to upload and organize the files.

Numerical data can be provided as individual .xls or .csv files (including a tab describing the data). For 'blots' or microscopy, uncropped images should be submitted (using a zip archive or a single pdf per main figure if multiple images need to be supplied for one panel). Additional information on source data and instruction on how to label the files are available at .

9) We replaced Supplementary Information with Expanded View (EV) Figures and Tables that are collapsible/expandable online (see examples in <https://www.embopress.org/doi/10.15252/embj.201695874>). A maximum of 5 EV Figures can be typeset. EV Figures should be cited as 'Figure EV1, Figure EV2" etc. in the text and their respective legends should be included in the main text after the legends of regular figures.

11) For data quantification: please specify the name of the statistical test used to generate error bars and P values, the number (n) of independent experiments (specify technical or biological replicates) underlying each data point and the test used to calculate p-values in each figure legend. The figure legends should contain a basic description of n, P and the test applied. Graphs must include a description of the bars and the error bars (s.d., s.e.m.).

We realize that it is difficult to revise to a specific deadline. In the interest of protecting the conceptual advance provided by the work, we recommend a revision within 3 months (12th Feb 2024). Please discuss the revision progress ahead of this time with the editor if you require more time to complete the revisions.

Referee #1:

In this manuscript, the authors suggest that tuft cell differentiation is regulated partially via an mTORC1-ATF4 dependent pathway that is dependent on U34 tRNA modification, Elp3. The authors draw comparisons of this ATF4 mechanism to those found in Elp3 deficiency in yeast in terms of amino acid (metabolism) changes. The authors showed that molecular signals from Elp3 changes are integrated via ATF4 either through the translation of Nprl2 which is an mTORC1 inhibitor or the amino acid response pathway. This manuscript seems quite interesting and well done, with multiple perturbations in vitro and in mouse models to demonstrate the mechanism of perturbation. There are some claims that the authors made that are questionable and should be addressed.

Major comments

1. The authors claim, "egg numbers significantly increased from day 6 to 11 post-infection..." However, this is not what the data shows, in the Elp3 deficient mice, there's an egg increase from day 6 to 7, then numbers start to decrease at day 8, suggesting that there is egg clearance occurring as indicated by days 11-15. With this observation, it seems like the mice can indeed clear the worms without Elp3, and thus, the conclusions regarding defective worm clearance may be overstated. The authors also tend to change the days in different plots (7 or 9 days) which is confusing to the readers
2. I think clarification is needed in the early figures regarding the downregulation of immune cascade effectors from tuft cells. It is unclear with the transcript of those effectors (IL-25) is directly regulated by Elp3 deficiency or whether that is merely a result of insufficient tuft cell hyperplasia. There are many instances like this (for example 3g), where the authors would use a western blot (for chga) and claims certain features on cell numbers
3. Figure 3C is not convincing as an IF (staining Paneth cell granules), please validate with FISH or another approach.
4. With regards to single-cell analysis, a major issue is that there are no statistical considerations regarding differences. The authors show a heatmap of signatures and claims increases or decreases, and a quantitative analysis would help strengthen this figure because the authors call attention to the differences in immature tuft and tuft cell populations between the elp3 deficient and the WT mice. Regarding differences in glycolytic and ox phos scores. There are no significant differences in the data shown to make those conclusions (even the Western Blots are not convincing). In fig. 4f authors claimed HK1 and Aldoa are not properly induced, but the western blots look very similar for both WT and Elp3 deficient. Western blot quantification should be used to demonstrate significance due to there not being a profound difference visually.
5. The authors reference both unstimulated and Il-13 treated mice in fig.5h but 5h data is comparing WT vs elp3 deficient, doesn't have any indication in the figure regarding Il-13 treatments.
6. Consider reformatting fig.6e, currently as is, looks like Atf4 expression seems to be increasing under Isrib and il-13 treatment at day 5 which is counter-intuitive to the authors proposed mechanism of action. Additionally, the shRNA for Atf4 seem only to be marginally effective via western blots.
7. The author did not provide any mechanistic insight into why the regulation of the Stat6 signaling pathway is altered in late response but not in early response, and how would that alter the phenotype of the animals.

Minor comments

In the intro, the authors say that tuft cell development does not depend on Atoh1. I don't think that statement is completely true given the references cited.

For Fig. 1A, although there is a bar plot showing quantification, at first glance the microscopy images look cherry picked. The authors should consider showing the entire image like demonstrated in fig.1F

For Fig.4e, if the claim is that metabolism is altered in Tuft cells in Elp3 deficient mice, identifying the tuft cells in the UMAP will better call our attention in the comparison.

In the section for figure 5, need to provide a source after this statement. "likewise, the expression of Atf4, an effector of both UPR and mTORC1 signaling". Additionally, in the text, the flow jumps from fig. c to h. Consider reordering figures.

In fig.7b the authors claim that Thapsigargin interfered with the production of DCLK1+ cells. The authors should use images to reflect the bar plots. Current images suggest that Thapsigargin has no effect or encourages tuft cell expansion because there are more numbers than in the control organoids. This is also counter-intuitive to the proposed mechanism explained by the authors.

Referee #2:

In their manuscript "Loss of Elp3 blocks intestinal tuft cell differentiation through Atf4 upregulation" Wathieu et al. show that Elp3 deficiency specifically impairs tuft cell differentiation and the correlated immune response against helminth infections in the intestine.

The manuscript is nicely written and the presentation is mostly clear, nevertheless, I have concerns about the interpretation of the results and their significance in terms of mechanisms, since the conclusions are often based on associative assumptions, or contradictory observations of which the significance remains to be proven. In addition, a direct mechanism through which some U34 tRNA modifications specifically promote tuft cell fate determination and regulate the immune response in the intestine is missing. Therefore, I consider the study preliminary and don't support the publication of this manuscript in The EMBO Journal at this stage.

Major points:

What is primarily missing is the direct mechanism through which Elp3 deficiency and U34 tRNA modifications impairs tuft cell differentiation.

- 1) Is it a cell autonomous mechanism in immature tuft cells that fails to start in absence of Elp3 or a systemic signal from the epithelium that is missing? This remains completely unclear, as Elp3 Δ IEC mice lack Elp3 expression in the entire intestinal epithelium.
- 2) More importantly, what is the role of ELp3 and modified tRNA in the differentiation program? Which RNAs are not properly translated in the absence of Elp3 that impair Tuft cells differentiation? A direct mechanism of Elp3 modified U34 tRNAs that affect the tuft cells differentiation program is missing here.
- 3) The single-cell RNA analysis shows increased mRNA expression of genes involved in oxidative phosphorylation and glycolysis in a variety of the epithelial cells, and this was validated in a Western blot (figure 4F) of the entire intestine. This poses again the questions why only tuft cells are affected. In addition, this dysregulation is mainly at the mRNA level, with the mRNA of these enzymes overexpressed in IL-13-stimulated Elp3 Δ IEC mice. This seems to be a secondary effect, since it is unlikely that the lack of ELp3 modified tRNAs can directly produce a stabilization of these mRNA during translation. Why these enzymes are overexpressed in absence of U34 tRNA modifications and why their function is important in the differentiation of tuft cells remains unclear.
- 4) While it is demonstrated nicely that Atf4 is a negative regulator of intestinal tuft cell differentiation (Figures 6, 7 and 8), the part of the manuscript addressing the mechanism of Atf4 activation and overexpression in the absence of ELp3 modified tRNAs is less convincing. The authors suggest that Atf4 is activated through the amino acid response signalling pathway, nevertheless they observe only some amino acids modestly elevated in IECs from Elp3 Δ IEC mice (figure 7E), and there is no sign of starvation which could have triggered the activation of Atf4, as in the study the authors cite (Kilberg et al, 2009). This point should be clarified.
- 5) The data in Figure 5D and 6A-E are contradictory and needs to be clarified. On one hand (figure 5D) the lack of induction of eIF2 α phosphorylation and UPR in Elp3 Δ IEC intestine excludes the activation of the integrated stress response as mechanism of upregulation of Atf4. On the other hand (Figure 6A-E) and in spite of this observation, the authors use Isrib as inhibitor of the integrated stress response to blocks Atf4 translation and rescues tuft cell differentiation. These are contrasting results which need more mechanisms inside to be explained.

Minor points:

1. In some Figures (e.g. 1 and 5) the order of panels is confusing. A better layout and organization are needed.

Referee #3:

Wathieu et al. provide a comprehensive report detailing the involvement of Elp3, a tRNA-modifying enzyme, in the expansion of tuft cells in response to Type 2 stimuli. This defect in tuft cell development in Elp3-Vil-Cre mice leads to a blunted Type 2 immune response and delayed worm clearance during infection with *Nippostrongylus brasiliensis*. The tuft cell activation circuit is complex and heavily regulated, indicating the importance of the homeostasis of the intestinal barrier in health and disease. This work from Wathieu et al. is an important contribution to our understanding of the mechanisms that regulate the composition of the barrier site. This has implications for the therapeutic targeting of the intestine in health, infection, and cancer.

Major comments:

1. In the discussion, the authors do not really address why (evolutionarily) there is this requirement for Elp3 activity for the effective development of tuft cells in naïve and *N. brasiliensis* infected mice. Could they speculate on whether this axis is important when responding to changes in host diet/nutritional or health status that may affect host metabolism?
2. The authors should include enumeration of tuft cells (by Dcl1 staining, for example) in the intestines of TgAtf1IEC naïve and infected mice.
3. In figure 3c, the authors intimate that Il13Ra1 expression is decreased in the crypts - and therefore in intestinal epithelial stem

cells in mice lacking Elp3 expression in IECs. Can the authors use their scRNAseq data to establish whether Elp3 is expressed and/or active in specific subpopulations of IECs, in particular Lgr5+ stem cells? Does Elp3 activity correlate with Il13Ra1 gene expression in specific IEC subsets? Further to this, the authors show a deficit in IL-13 production by ILC2s at d3 of Nb infection. This is a very early timepoint (I understand the technical difficulties of looking at a later timepoint), is there also a defect in ILC2 IL-13 production in the steady-state? Could this account for a reduction in basal tuft cell numbers? Or is this also linked to changes in IL13Ra1 expression in the IEC compartment?

Minor comments:

1. For the readouts of parasite burden, the authors show increased worm and egg numbers. Is the increase in egg deposition simply associated with an increase in worm burden or are the worms also more fecund? At d9, some Elp3IEC mice still have worms in the intestine, is this demonstrative of a failure to completely clear parasites, or represents a temporal delay in total clearance?
2. Given the changes in ILC2 migration in Elp3IEC mice, is there a defect in PGD2 production in the intestine of Nb-infected KO mice?

Referee #1:

In this manuscript, the authors suggest that tuft cell differentiation is regulated partially via an mTORC1-ATF4 dependent pathway that is dependent on U34 tRNA modification, Elp3. The authors draw comparisons of this ATF4 mechanism to those found in Elp3 deficiency in yeast in terms of amino acid (metabolism) changes. The authors showed that molecular signals from Elp3 changes are integrated via ATF4 either through the translation of Npr12 which is an mTORC1 inhibitor or the amino acid response pathway. This manuscript seems quite interesting and well done, with multiple perturbations in vitro and in mouse models to demonstrate the mechanism of perturbation. There are some claims that the authors made that are questionable and should be addressed.

Major comments:

1. The authors claim, "egg numbers significantly increased from day 6 to 11 post-infection..." However, this is not what the data shows, in the Elp3 deficient mice, there's an egg increase from day 6 to 7, then numbers start to decrease at day 8, suggesting that there is egg clearance occurring as indicated by days 11-15. With this observation, it seems like the mice can indeed clear the worms without Elp3, and thus, the conclusions regarding defective worm clearance may be overstated. The authors also tend to change the days in different plots (7 or 9 days) which is confusing to the readers.

Our answer:

We agree with Reviewer 1 that Elp3-deficient mice can ultimately clear worms, although much less efficiently, as they fail to properly amplify the pool of tuft cells upon *N. brasiliensis* infection. We rephrased the conclusion accordingly in the revised version of our paper. We added the following sentence: "Despite similar infection rates in the lung by day 2 post-infection, Elp3^{ΔIEC} mice showed higher egg numbers from day 6 to day 11 post-infection as well as higher numbers of worms in their intestine, 5 or 7 days post-infection (Figs. 1B to 1D, respectively). Nevertheless, no eggs could be detected by day 12 in Elp3 deficient mice, suggesting effective but delayed worm clearance (Fig. 1C)".

2. I think clarification is needed in the early figures regarding the downregulation of immune cascade effectors from tuft cells. It is unclear with the transcript of those effectors (IL-25) is directly regulated by Elp3 deficiency or whether that is merely a result of insufficient tuft cell hyperplasia. There are many instances like this (for example 3g), where the authors would use a western blot (for chga) and claims certain features on cell numbers.

Our answer:

The defective induction of IL-25 in Elp3^{ΔIEC} mice infected with *N. brasiliensis* and illustrated in Figure 2A indeed results from an impaired tuft cell amplification. We rephrased this conclusion to avoid any confusion in the revised version of our paper. Regarding Figure

3G, we now provide an IF analysis (Chromogranin A staining) in which we show that the number of enteroendocrine cells is similarly increasing in $Elp3^{WT}$ and $Elp3^{\Delta IEC}$ mice subjected to DBZ for 5 days, which is in agreement with our western blot data (Figure EV2C and here below). We systematically checked our conclusions on western blot data and on cell numbers throughout the manuscript to avoid any confusion. Data obtained through both western blots and immunofluorescence are consistent all the time.

Figure: $Elp3$ deficiency does not regulate the number of enteroendocrine cells in naive mice as well as upon Notch inhibition. The number of enteroendocrine cells (Chromogranin A⁺) was assessed by immunofluorescence analyses in the small intestine of the indicated mice treated or not with the Notch inhibitor DBZ (mean values \pm SEM; Mann-Whitney test; $n \geq 3$).

3. Figure 3C is not convincing as an IF (staining Paneth cell granules), please validate with FISH or another approach.

Our answer:

Our result is in agreement with a previous study illustrating that IL-13R α 1 expression is enriched in stem cells, i.e. at the bottom of intestinal crypts, as well as in tuft cells (our figure 3D): <https://www.nature.com/articles/s41590-018-0297-6#Sec2>;

4. With regards to single-cell analysis, a major issue is that there are no statistical considerations regarding differences. The authors show a heatmap of signatures and claims increases or decreases, and a quantitative analysis would help strengthen this figure because the authors call attention to the differences in immature tuft and tuft cell populations between the $elp3$ deficient and the WT mice. Regarding differences in glycolytic and ox phos scores. There are no significant differences in the data shown to

make those conclusions (even the Western Blots are not convincing). In fig. 4f authors claimed HK1 and Aldoa are not properly induced, but the western blots look very similar for both WT and Elp3 deficient. Western blot quantification should be used to demonstrate significance due to there not being a profound difference visually.

Our answer:

For our single cell analysis, we used GSEA analysis to identify pathways that were statistically significantly up- or downregulated in Elp3 deficient mice (Figures 5A and 6A). GSEA was performed to determine gene sets (minimal size = 7 genes) showing differentially expressed genes identified between Elp3^{WT} and Elp3^{ΔIEC} mice. The statistical significant enrichment of the gene set was selected based on false discovery rate (FDR) ≤0.25 and the normalized enriched score (NES) was calculated for each gene set. This additional information is now mentioned in the revised version of our manuscript.

In Figure 5A, GSEA was performed to highlight signaling pathways downregulated in tuft cells from Elp3^{ΔIEC} mice compared to tuft cells from Elp3^{WT} mice. In Figure 6A, GSEA was performed to highlight signaling pathways up- or downregulated in different immature cell populations from Elp3^{ΔIEC} mice compared to Elp3^{WT} mice. In Figures 5B and 6F showing oxidative phosphorylation and upregulation of mTorc1-Atf4 target genes in Elp3^{ΔIEC} mice compared to Elp3^{WT} mice, respectively, only genes that were significantly up- or downregulated in one or more of the represented populations were selected (based on $p_{val_adj} < 0.05$ and $logFC > 0.25$). We now provide all heatMaps and tables with p-values as Supplementary information. Importantly, it is very difficult to obtain a sufficient amount of tuft cells (they are rare cells in the intestinal epithelium), which impacts the statistical strength of our observations, as indicated by the adjusted p-value in our comparisons.

Please find here a quantification of our western blot data (ImageJ/Fiji) illustrated in Figure 5D. These quantifications were added on the revised manuscript ($n \geq 4$ independent experiments for both glycolytic enzymes).

Figure: *Elp3* deficiency in the intestine interferes with IL-13 induction of specific glycolytic enzymes. Elp3^{WT} and Elp3^{ΔIEC} mice were treated or not with rIL-13 for 4 days and the resulting extracts from the intestines were subjected to western blot analyses using the indicated antibodies. A quantification of relative expression (ratio on β-Actin) in IL-13-stimulated mice from several experiments is illustrated (mean values ± SEM; Student t test; $n \geq 4$).

5. The authors reference both unstimulated and Il-13 treated mice in fig.5h but 5h data is comparing WT vs elp3 deficient, doesn't have any indication in the figure regarding Il-13 treatments.

Our answer:

We apologize for this mistake. Former Figure 5H (now Figure 6H) is indeed exclusively showing data with both Elp3^{WT} and Elp3^{ΔIEC} naive mice. We corrected the revised manuscript accordingly.

6. Consider reformatting fig.6e, currently as is, looks like Atf4 expression seems to be increasing under Isrib and il-13 treatment at day 5 which is counter-intuitive to the authors proposed mechanism of action. Additionally, the shRNA for Atf4 seem only to be marginally effective via western blots.

Our answer:

We confirm that Atf4 protein levels decrease upon Isrib treatment at day 5 (please compare lanes 1 and 3 of former Figure 6E, now Figure 8E). At the same time, Dclk1 levels increase. Regarding effects of IL-13 on both Atf4 and Dclk1 levels, Dclk1 levels increases in ex-vivo organoids treated with IL-13 alone, which is expected as this cytokine promotes tuft cell differentiation (please compare lanes 1 and 5). In those circumstances, IL-13 alone slightly decreases Atf4 levels, which supports the notion that Atf4 acts as a negative regulator of tuft cell differentiation. If we compare lanes 4 and 5 (IL-13 treatments), we clearly see that Isrib decreases Atf4 levels in IL-13-stimulated ex-vivo organoids. Meanwhile, Dclk1 levels are higher in ex-vivo organoids treated with both Isrib and IL-13 compared to organoids treated with IL-13 alone (compare lanes 4 and 5). Collectively, these results fit with our conclusion: Atf4 is a negative regulator of tuft cell differentiation.

Regarding the efficiency of ShRNA-mediated Atf4 downregulation, let's keep in mind that Atf4 levels are low in ex-vivo organoids not subjected to any type of stress. As a result, we do not expect a huge downregulation with both shRNAs targeting Atf4. Yet, the consequences of Atf4 deficiency on Dclk1 in this experiment are obvious to us and this is an additional demonstration that Atf4 acts as a negative regulator of tuft cell differentiation. A quantification of western blot signals from several experiments is illustrated here after and was also added in the revised manuscript.

Figure: Depletion of Atf4 in ex-vivo organoids. Western blot data were quantified (paired Student T-test, mean values \pm SEM; n = 3).

7. The author did not provide any mechanistic insight into why the regulation of the Stat6 signaling pathway is altered in late response but not in early response, and how would that alter the phenotype of the animals.

Our answer:

We added some immunofluorescence analyses in which we showed that Stat6 phosphorylation was enriched in tuft cells (Fig. EV2A). An impaired Stat6 phosphorylation seen in the intestinal epithelium of $Elp3^{\Delta IEC}$ mice infected with *N. brasiliensis* is the indirect consequence of a lower number of tuft cells in these mice. Therefore, we believe that these additional immunofluorescence analyses will somehow clarify this issue.

Minor comments:

In the intro, the authors say that tuft cell development does not depend on Atoh1. I don't think that statement is completely true given the references cited.

Our answer:

We thank Reviewer 1 for pointing out this issue. It is indeed true that the involvement of Atoh1 in intestinal tuft cell lineage has been controversial for many years. We re-wrote the introduction and gave more details on the roles of Atoh1 in tuft cell differentiation: “Intestinal tuft cells, a rare population of IECs, derive from $Lgr5^+$ stem cells and $Gfi1b$ -expressing progenitors and rely on $Pou2f3$, $Sox4$ and $Stat6$ for their development (Gerbe et al, 2016; Gracz et al, 2018; Howitt et al, 2016). Some studies demonstrated that the transcription factor Atoh1 was dispensable for tuft cell differentiation (Bjerknes et al, 2012; Gracz et al, 2018) while another study actually demonstrated that Atoh1 was required in this process (Gerbe et al., 2011). It is now established that tuft cell differentiation occurs through both

Atoh1-dependent and independent pathways (Gerbe et al, 2011; Herring et al, 2018). Indeed, tuft cell differentiation is Atoh1-independent in the small intestine but Atoh1-dependent in the colon.”

For Fig. 1A, although there is a bar plot showing quantification, at first glance the microscopy images look cherry picked. The authors should consider showing the entire image like demonstrated in fig.1F.

Our answer:

We now provide the requested IF images in the revised manuscript (see here after as well).

Figure: *Elp3* deficiency impairs intestinal tuft cell numbers Tuft cell numbers are reduced in intestines lacking *Elp3*. Immunostainings and corresponding quantifications (left and right panels, respectively) of tuft cells (Dclk1⁺ cells, in red) in the small intestine of naive *Elp3*^{WT} and *Elp3*^{ΔIEC} mice (mean values ± SEM; Mann-Whitney test; n = 8).

For Fig.4e, if the claim is that metabolism is altered in Tuft cells in *Elp3* deficient mice, identifying the tuft cells in the UMAP will better call our attention in the comparison.

Our answer:

We now provide the requested information in our revised version (Former Figure 4E now Figure 5C) as well as here after.

Figure: Gene candidates involved in both oxidative phosphorylation and glycolysis are enriched in tuft cells compared to other secretory cells and both signatures are impaired upon *Elp3* deficiency. An UMAP showing both metabolic pathways enriched in tuft cells (n = 2 pooled mice per genotype) is illustrated (data with enterocytes are removed for clarity purposes). Tuft cell population is circled in red.

In the section for figure 5, need to provide a source after this statement. "likewise, the expression of Atf4, an effector of both UPR and mTORC1 signaling".

Our answer:

We added the following reference right after this sentence:

Torrence M, MacArthur M, Hosios A, Valvezan A, Asara J, Mitchell J & Manning B (2021)

The mTORC1-mediated activation of ATF4 promotes protein and glutathione synthesis downstream of growth signals. *eLife* 10: e63326.

As stated later in the manuscript, Atf4 is an effector of mTORC1 and a specific transcriptional signature can be defined downstream of this pathway.

Additionally, in the text, the flow jumps from fig. c to h. Consider reordering figures.

Our answer:

We thank Reviewer 1 for pointing out this issue. We moved a sentence related to Atf4 target genes (Asns) later in the manuscript in order to resolve this issue.

In fig.7b the authors claim that Thapsigargin interfered with the production of DCLK1+ cells. The authors should use images to reflect the bar plots. Current images suggest that Thapsigargin has no effect or encourages tuft cell expansion because there are more numbers than in the control organoids. This is also counter-intuitive to the proposed mechanism explained by the authors.

Our answer:

We thank Reviewer 1 for highlighting this confusion. We now provide other IF images of ex-vivo organoids treated with Thapsigargin which better reflect our quantification of Dclk1⁺ cells. It is fair to say that the inhibitory effect of Thapsigargin on the number of tuft cells is more dramatic upon IL-13 stimulation.

Figure: UPR activation impairs tuft cell differentiation in ex-vivo organoids. The UPR inducer Thapsigargin blocks IL-13-dependent Dclk1 induction, as evidenced by immunofluorescence analyses. Immunostainings and corresponding quantifications (left and right panel, respectively) of Dclk1⁺ cells (in red) in WT organoids treated with mL-13 and with Thapsigargin or with the vehicle (mean values ± SEM; Mann-Whitney test; n ≥ 22 organoids from n ≥ 3 mice) are illustrated.

Referee #2:

In their manuscript "Loss of Elp3 blocks intestinal tuft cell differentiation through Atf4 upregulation" Wathieu et al. show that Elp3 deficiency specifically impairs tuft cell differentiation and the correlated immune response against helminth infections in the intestine.

The manuscript is nicely written and the presentation is mostly clear, nevertheless, I have concerns about the interpretation of the results and their significance in terms of mechanisms, since the conclusions are often based on associative assumptions, or contradictory observations of which the significance remains to be proven. In addition, a direct mechanism through which some U34 tRNA modifications specifically promote tuft cell fate determination and regulate the immune response in the intestine is missing. Therefore, I consider the study preliminary and don't support the publication of this manuscript in The EMBO Journal at this stage.

Major points:

What is primarily missing is the direct mechanism through which Elp3 deficiency and U34 tRNA modifications impairs tuft cell differentiation.

Our answer:

Our revised manuscript now includes more results on molecular mechanisms through which U₃₄ tRNA modifications promote intestinal tuft cell differentiation, which is indeed very important to resolve. Our new data are described in the newly generated Figure 7 (these results are illustrated here after as well). We first showed that the Gator1 subunit and mTORC1 inhibitor Nprl2 is not properly expressed in the intestine of Elp3^{ΔIEC} mice (Figure 7A). Importantly, Nprl2 protein but not mRNA levels were decreased upon *Elp3* deficiency (Figure 7A, left and right panels, respectively). Importantly, this conclusion is also relevant in another experimental system. Indeed, an immortalized intestinal epithelial cell line lacking Elp3 also showed decreased Nprl2 protein but not mRNA levels (Figures 7B and 7C, respectively).

Figure: Elp3 promotes Nprl2 mRNA translation. **A.** Defective Nprl2 expression at the protein level upon *Elp3* deficiency in IECs. On the left, extracts from the intestines of both naive $Elp3^{WT}$ and $Elp3^{\Delta IEC}$ mice were subjected to western blot analyses using the indicated antibodies. On the right, Nprl2 mRNA levels in the small intestine of naive $Elp3^{WT}$ and $Elp3^{\Delta IEC}$ mice were quantified by Real-Time PCRs. Normalization was calculated on the house keeping gene *Gapdh* (mean values \pm SEM; $n = 3$). **B.** *Elp3* promotes Nprl2 expression. An immortalized mouse intestinal epithelial cell line (mIECs) was transfected with the indicated shRNA construct and extracts from the resulting cells were subjected to western blot analyses using the indicated antibodies.

Figure: Elp3 regulates protein but not mRNA levels of Nprl2. *Elp3* and *Nprl2* mRNA levels were quantified by Real-Time PCRs in all indicated experimental conditions (c). mRNA levels in cells transfected with the ShRNA Control were set to 1 and levels in other conditions were relative to that after normalization with *Gapdh* mRNA levels (mean values \pm SEM; Student t test; $n = 4$).

To explore the underlying mechanism, we conducted two distinct experiments. We first carried out a polysome profiling experiment on the *Nprl2* transcript using extracts from both $Elp3^{WT}$ and $Elp3^{\Delta IEC}$ mice. We observed an increase in ribosome density on the *Nprl2* transcript upon *Elp3* deficiency (Fig. 7D). Considering that *Nprl2* protein but not mRNA levels are reduced in IECs lacking *Elp3*, these results suggest a ribosome pausing on *Nprl2* mRNA, leading to decreased translational efficiency.

Figure: Enhanced ribosome density on the Nprl2 transcript upon Elp3 deficiency. A polysome profiling experiment was performed on IECs from both $Elp3^{WT}$ and $Elp3^{\Delta IEC}$ naïve mice. Ribosome occupancy on the *Nprl2* transcript was calculated by Real-Time PCR (polysomal fraction/input ratio; mean values \pm SEM; Student t test; $n \geq 4$).

The next experiment was even more important to carry out. Elp3 promotes mRNA translation in a codon-dependent manner (see the reference by Damien Hermand and colleagues). Therefore, to assess whether Nprl2 is a direct target of Elp3, we generated a Nprl2 mutant in which Lys^{AAA}, Gln^{CAA} and Glu^{GAA} codons which rely on Elp3 to be decoded (“Nprl2 WT”) were replaced by synonymous Lys^{AAG}, Gln^{CAG} and Glu^{GAG} codons (“Nprl2 Mut”) which escape from Elp3 regulation (Fig. 7E). Both Nprl2 WT and Nprl2 Mut were expressed at very similar levels in the easily transfectable HEK293 cell line (Fig. 7E). Importantly, while Nprl2 WT expression was defective in HEK293 cells lacking Elp3 or the thiolase Ctu2 (which acts in the same enzymatic cascade as Elp3), Nprl2 Mut was properly expressed upon Elp3 or Ctu2 deficiency (Fig. 7E). Therefore, Nprl2 is a direct target of Elp3.

Figure: Elp3 promotes Nprl2 mRNA translation in a codon-dependent manner. HEK293 cells infected with the shRNA control or targeting either Elp3 or Ctu2 were transfected with the indicated expression construct (cf the schematic representation of both wild type and mutated Nprl2 with the localization of all mutations) and the resulting extracts were subjected to western blot analyses using the indicated antibodies.

At this time, we cannot rule out the possibility that Elp3 promotes the codon-dependent mRNA translation of additional candidates but we can nevertheless conclude that Nprl2 is a key candidate through which Elp3 promotes tuft cell differentiation. Indeed, we depleted Nprl2 in ex-vivo organoids from mouse intestinal crypts and noticed that IL-13-dependent Dclk1 induction was robustly impaired, at least through Atf4 stabilization (Fig. 7F). Likewise, the production of Dclk1⁺ cells upon IL-13 stimulation was severely impaired in ex-vivo organoids in which Nprl2 was depleted (Fig. 7G). Therefore, Elp3 promotes intestinal tuft cell differentiation, at least through Nprl2 mRNA translation.

Figure: Nprl2 promotes tuft cell differentiation. Western blot analyses using the indicated antibodies were conducted using extracts from untreated or IL-13-stimulated ex-vivo organoids lacking or not Nprl2 (100 ng/ml, 3 days).

Figure: Nprl2 promotes tuft cell differentiation. Anti-Dclk1 immunofluorescence analyses were also carried out with samples from untreated or IL-13-stimulated ex-vivo organoids lacking or not Nprl2 (100 ng/ml, 3 days) to quantify the number of Dclk1⁺ cells (in red) in all indicated experimental conditions (mean values \pm SEM; Student t test; $n \geq 3$ organoids).

1) Is it a cell autonomous mechanism in immature tuft cells that fails to start in absence of Elp3 or a systemic signal from the epithelium that is missing? This remains completely unclear, as Elp3^{AIEC} mice lack Elp3 expression in the entire intestinal epithelium.

Our answer:

The depletion of Elp3 in the entire intestinal epithelium leads to really mild effects on intestinal homeostasis. When we performed GSEA on data from Single cell RNA experiments, we observed activation of the mTORC1-Atf4 pathway mainly in immature cells, including TA cells, immature tuft cells, immature goblet cells and immature enterocytes (but not in stem cells). Atf4 acts as a “brake” for tuft cell differentiation and accumulation of Atf4 leads to impaired tuft cell differentiation, as demonstrated in Figures 8, 9 and 10. Whether the

activation of Atf4 uniquely in immature tuft cells would be sufficient to block tuft cell differentiation is still unclear.

In any case, Reviewer 2 is raising a key issue that would be easy to experimentally address if specific CRE lines for all epithelial subtypes would be available, which is not the case. Immature tuft cells are poorly characterized and there is currently not experimental way to specifically inactivate *Elp3* in immature tuft cells. Our data nevertheless suggest the existence of a cell autonomous mechanism in early steps of tuft cell differentiation which relies on *Elp3* and on *Npr12* to prevent the aberrant activation of the mTORC1-Atf4 signaling axis. We actually observed less immature tuft cells in *Elp3*-deficient mice compared to *Elp3*^{WT} mice, suggesting that *Elp3* is required for cell lineage specification at this stage (Figure 4B). We now provide an additional result in which we established a specific gene signature of immature tuft cells using our single cell data (see our new Figure 4D illustrated here after). We could now rely on this signature to design a new CRE mouse strain in which any gene of interest would be genetically inactivated in these immature tuft cells but such experiment would be beyond the scope of this manuscript.

Figure: Identification of the transcriptional signature of immature tuft cells. Dotplot illustrating candidates whose mRNA levels are enriched in immature tuft cells (Single cell RNA sequencing data carried out with extracts from *Elp3*^{WT} mice subjected to rIL-13 treatment overnight).

We also performed qPCRs on samples from the entire intestine to further investigate differences in the immature tuft cell pool between $Elp3^{WT}$ and $Elp3^{\Delta IEC}$ mice. We now show multiple candidates whose expression is enriched in immature tuft cells (red circles), are not properly expressed in the intestine of $Elp3^{\Delta IEC}$ mice infected with *N.brasiliensis* (see our new Figure 4E illustrated here after). Collectively, our results demonstrate that *Elp3* deficiency causes a blockage at an early stage of differentiation of tuft cells.

Figure: Expression of candidates enriched in immature tuft cells is impaired in $Elp3^{\Delta IEC}$ mice infected with *N.brasiliensis*. mRNA levels of the indicated candidates in the small intestine of $Elp3^{WT}$ and $Elp3^{\Delta IEC}$ mice naive or 7 days post-*N. brasiliensis* infection were quantified by Real-Time PCRs. Normalization was calculated on the average of 2 house keeping genes *Gapdh* and *36b4* (mean values \pm SEM; Student t test; n = 3 and 2 for naive $Elp3^{WT}$ and $Elp3^{\Delta IEC}$ mice; n = 5 and 4 for infected $Elp3^{WT}$ and $Elp3^{\Delta IEC}$ mice).

2) More importantly, what is the role of *Elp3* and modified tRNA in the differentiation program? Which RNAs are not properly translated in the absence of *Elp3* that impair Tuft cells differentiation? A direct mechanism of *Elp3* modified U34 tRNAs that affect the tuft cells differentiation program is missing here.

Our answer:

Our revised manuscript now includes more results on molecular mechanisms through which U₃₄ tRNA modifications promote intestinal tuft cell differentiation, which is indeed very important to resolve. This key issue has been explained in details in the previous issue raised by Reviewer 2. We now show that *Nprl2* mRNA translation relies on *Elp3* and this the mechanism through which *Elp3* limits both mTORC1 activation and *Atf4* protein levels. Please see all corresponding figures here before. This is basically our model:

Elp3 deficiency leads to ribosome pausing on *Nprl2* mRNAs, which causes *Nprl2* downregulation at the protein level. *Nprl2* is a mTORC1 repressor. Therefore, its downregulation promotes mTORC1 activation and subsequent *Atf4* accumulation. *Atf4* acts as a “brake” for tuft cell differentiation and accumulation of *Atf4* leads to impaired tuft cell differentiation.

3) The single-cell RNA analysis shows increased mRNA expression of genes involved in oxidative phosphorylation and glycolysis in a variety of the epithelial cells, and this was validated in a Western blot (figure 4F) of the entire intestine. This poses again the questions why only tuft cells are affected. In addition, this dysregulation is mainly at the mRNA level, with the mRNA of these enzymes overexpressed in IL-13-stimulated *Elp3^{ΔIEC}* mice. This seems to be a secondary effect, since it is unlikely that the lack of *Elp3* modified tRNAs can directly produce a stabilization of these mRNA during translation. Why these enzymes are overexpressed in absence of U34 tRNA modifications and why their function is important in the differentiation of tuft cells remains unclear.

Our answer:

We show for the first time that specific enzymes such as Hexokinase 1 and Aldolase A are robustly induced by IL-13 in IECs, which defines them as candidates enriched in tuft cells. The specific expression of these enzymes in tuft cells is supported by our Single Cell RNA Sequencing data. Therefore, the defective expression of both enzymes at the mRNA levels in IECs lacking *Elp3* is an indirect consequence of impaired tuft cell differentiation, similarly to *Dclk1*, we believe. As tuft cell differentiation is impaired but not totally blocked upon *Elp3* deficiency, we still see some IL-13-dependent induction of these enzymes in IECs lacking *Elp3*.

Figure: Expression profile of the glycolytic enzymes Hexokinase 1 and Aldolase A showing enrichment in tuft cell population. A heatmap showing Hexokinase 1 and Aldolase A enrichment in tuft cell population based on transcriptomic analyses done with extracts from the small intestine of both $Elp3^{WT}$ and $Elp3^{\Delta IEC}$ mice subjected to rIL-13 stimulation overnight is illustrated (n = 2 pooled mice per genotype).

Whether both enzymes actively contributes to intestinal tuft cell differentiation remains an open issue. The depletion of each enzyme should be done in ex-vivo organoids or even in mice to experimentally address this issue but such experiments would bring us beyond the scope of the current manuscript. We felt that it was important to illustrate the notion that specific glycolytic enzymes such as Hexokinase 1 and Aldolase A are specifically expressed by tuft cells and robustly induced by IL-13, which opens new avenues for future scientific projects.

4) While it is demonstrated nicely that Atf4 is a negative regulator of intestinal tuft cell differentiation (Figures 6, 7 and 8), the part of the manuscript addressing the mechanism of Atf4 activation and overexpression in the absence of Elp3 modified tRNAs is less convincing. The authors suggest that Atf4 is activated through the amino acid response signalling pathway, nevertheless they observe only some amino acids modestly elevated in IECs from $Elp3^{\Delta IEC}$ mice (figure 7E), and there is no sign of starvation which could have triggered the activation of Atf4, as in the study the authors cite (Kilberg et al, 2009). This point should be clarified.

Our answer:

Can we invite Reviewer 2 to look at data illustrated in Figure 6I ? We show that the mTORC1 inhibitor Rapamycin, which downregulates S6 phosphorylation, interferes with Atf4 levels. As a consequence, levels of Asns, an Atf4 target gene, are also decreased upon Rapamycin treatment. While we do not rule out the possibility that Aft4 is stabilized through additional, mTORC1-independent pathways, we demonstrate that mTORC1 signaling is nevertheless a key pathway through which Aft4 protein levels are enhanced.

5) The data in Figure 5D and 6A-E are contradictory and needs to be clarified. On one hand (figure 5D) the lack of induction of eIF2alpha phosphorylation and UPR in $Elp3^{\Delta IEC}$

intestine excludes the activation of the integrated stress response as mechanism of upregulation of Atf4. On the other hand (Figure 6A-E) and in spite of this observation, the authors use Isrib as inhibitor of the integrated stress response to blocks Atf4 translation and rescues tuft cell differentiation. These are contrasting results which need more mechanisms inside to be explained.

Our answer:

We understand the confusion raised by Reviewer 2 but our wish is to use a variety of experimental approaches to increase or decrease Atf4 levels in order to assess the consequences on tuft cell differentiation. Using Isrib is a strategy to interfere with Atf4 levels and to show that tuft cell differentiation is consequently enhanced. These results, combined with data obtained with our Atf4-overexpressing mouse, demonstrate that Atf4 is a negative regulator of tuft cell differentiation. In other words, any drug that would influence Atf4 levels will impact on tuft cell differentiation, even if the « canonical » UPR cascade upstream of Atf4 is not activated upon *Elp3* deficiency. Any inhibitor of the integrated stress response which impacts on Atf4 protein levels will trigger consequences on tuft cell differentiation. It is true that *Elp3* deficiency does not show any sign of activation of the integrated stress response but nevertheless leads to enhances Atf4 protein levels, at least through enhanced mTORC1 activation. Therefore, consequences on tuft cell differentiation is observed. Atf4 is truly the key actor to focus on.

Minor points:

1. In some Figures (e.g. 1 and 5) the order of panels is confusing. A better layout and organization are needed.

Our answer:

We worked on this issue in the revised version of our paper.

Referee #3:

Wathieu et al. provide a comprehensive report detailing the involvement of Elp3, a tRNA-modifying enzyme, in the expansion of tuft cells in response to Type 2 stimuli. This defect in tuft cell development in Elp3-Vil-Cre mice leads to a blunted Type 2 immune response and delayed worm clearance during infection with *Nippostrongylus brasiliensis*. The tuft cell activation circuit is complex and heavily regulated, indicating the importance of the homeostasis of the intestinal barrier in health and disease. This work from Wathieu et al. is an important contribution to our understanding of the mechanisms that regulate the composition of the barrier site. This has implications for the therapeutic targeting of the intestine in health, infection, and cancer.

Our answer:

We thank Reviewer 3 for his/her nice comments on our study.

Major comments:

1. In the discussion, the authors do not really address why (evolutionarily) there is this requirement for Elp3 activity for the effective development of tuft cells in naïve and *N. brasiliensis* infected mice. Could they speculate on whether this axis is important when responding to changes in host diet/nutritional or health status that may affect host metabolism?

Our answer:

This is a very interesting issue raised by Reviewer 3. We discussed this issue in the revised version, keeping in mind that it will be speculative and will need some experimental validation. We added the following sentence: “The definition of Atf4 as a negative regulator of intestinal tuft cell differentiation opens the possibility that a variety of evolutionary conserved cascades known to be regulated by the host diet/nutritional or health status and connected to Atf4 may ultimately interfere with intestinal tuft cell differentiation with consequences on the anti-helminth immune response in the intestine. Our work may then contribute to better understand at the molecular level how the health status can influence the immune response in the intestine.”

2. The authors should include enumeration of tuft cells (by Dclk1 staining, for example) in the intestines of TgAtf4^{IEC} naïve and infected mice.

Our answer:

We now provide the requested experiment in the revised version of our manuscript (Figure 10G). In agreement with previous results, the amplification of Dclk1⁺ cells upon *N.brasiliensis* infection is severely impaired in TgAtf4^{IEC} mice.

Figure: Atf4 overexpression in the mouse intestine interferes with the amplification of tuft cells upon *N.brasiliensis* infection. Immunostainings and corresponding quantifications (upper and lower panels, respectively) of tuft cells (Dclk1⁺ cells, in red) in the small intestine of Elp3^{WT} and TgAtf4^{IEC} mice naive or 7 days post-*N. brasiliensis* infection are shown (mean values ± SEM; Student t test; n ≥ 3).

3. In figure 3c, the authors intimate that IL13Rα1 expression is decreased in the crypts - and therefore in intestinal epithelial stem cells in mice lacking Elp3 expression in IECs. Can the authors use their scRNAseq data to establish whether Elp3 is expressed and/or active in specific subpopulations of IECs, in particular Lgr5⁺ stem cells? Does Elp3 activity correlate with IL13Rα1 gene expression in specific IEC subsets? Further to this, the authors show a deficit in IL-13 production by ILC2s at d3 of Nb infection. This is a very early timepoint (I understand the technical difficulties of looking at a later timepoint), is there also a defect in ILC2 IL-13 production in the steady-state? Could this account for a reduction in basal tuft cell numbers? Or is this also linked to changes in IL13Ra1 expression in the IEC compartment?

Our answer:

We now provide our Single cell RNA Sequencing data obtained with extracts of IECs from $Elp3^{WT}$ mice treated with rIL-13 in which $Elp3$ expression in all epithelial subtypes is illustrated (Figure 4C). $Elp3$ expression was mainly detected in both immature tuft and goblet cells as well as in enterocyte progenitors and transit-amplifying and stem cells.

Figure: $Elp3$ expression profile in mouse intestinal epithelial cells revealed through single-cell RNA sequencing. Dot plot illustrating $Elp3$ expression among epithelial subtypes in the small intestine of $Elp3^{WT}$ mice subjected to rIL-13 treatment overnight.

We believe that the decreased IL-13R α 1 expression that we see upon $Elp3$ deficiency reflects the decreased number of tuft cells in $Elp3^{\Delta IEC}$ mice (please see Figure 3D showing that this receptor is expressed in tuft cells).

Indeed, we had strong difficulties to isolate alive ILC2s from WT mice at later time points. At day 4 post-infection, ILC2 viability was really low in WT mice. This explains why we did not include these data. Importantly, we were able to extract normal amounts of alive ILC2s in $Elp3^{\Delta IEC}$ mice, reflecting a delay in ILC2 activation in $Elp3^{\Delta IEC}$ mice upon *N.brasiliensis* infection.

To experimentally address Reviewer's 3 about homeostasis, we carried out additional FACS analyses on naïve mice and concluded that $Elp3$ deficiency in IECs of naive mice does not interfere with the percentage of activated ILC2s (see here after).

Figure: *Elp3* deficiency in IECs does not interfere with the percentage of IL-13-producing ILC2 cells under homeostasis. Lamina propria cells were isolated and subjected to PMA and Ionomycin restimulation in presence of brefeldin and monensin before intracellular staining for IL-13. ILC2s were gated as live CD45⁺Lin⁻KLRG1⁺ cells. Lineage included APC-labeled antibodies to CD3 ϵ , CD4, CD8 α , CD11c, siglec-F, Fc ϵ RI, B220/CD45R, Gr-1, CD5, CD49b and F4/80 (mean values \pm SEM; Mann-Whitney test; n \geq 8).

Minor comments:

1. For the readouts of parasite burden, the authors show increased worm and egg numbers. Is the increase in egg deposition simply associated with an increase in worm burden or are the worms also more fecund? At d9, some Elp3 Δ IEC mice still have worms in the intestine, is this demonstrative of a failure to completely clear parasites, or represents a temporal delay in total clearance?

Our answer:

We indeed believe that the increase in egg deposition is associated with an increase in worm burden due to decreased tuft and goblet cell numbers upon *Elp3* deficiency in infected mice. We do not have any evidence that worms are more fecund upon *Elp3* deficiency. This issue is indeed interesting but, at this stage, we do not see any molecular mechanism through which *Elp3* would regulate this process, other than promoting tuft cell differentiation. We believe that we are dealing with a temporal delay in parasite clearance as parasite numbers decrease over time in Elp3 Δ IEC mice. Indeed, few parasites were still present in the small intestine of *Elp3* deficient mice at D9 post-infection. In addition, we could not observe any egg in feces from *Elp3* deficient mice from D12 post-infection, reflecting total clearance of the parasites. This can be easily explained by the fact that Elp3 Δ IEC mice are not totally devoid of tuft cells. The immune cascade induced by fewer tuft cells is reduced and mice need more time to control the infection. A role for antigen-specific response of Th2 cells at these later time points post-infection might also be occurring in these mice.

2. Given the changes in ILC2 migration in $Elp3^{\Delta IEC}$ mice, is there a defect in PGD2 production in the intestine of Nb-infected KO mice?

Our answer:

Reviewer 3 is raising an interesting question. As enzymes involved in PGD2 production are mainly expressed by tuft cells, we assessed the expression of these enzymes (Hpgds, Ptgs1 and Ptgs2) on samples from both naïve $Elp3^{WT}$ and $Elp3^{\Delta IEC}$ mice infected or not with *N. brasiliensis*.

Figure: Impaired expression of enzymes involved in PGD2 production in $Elp3^{\Delta IEC}$ mice. mRNA levels of the indicated enzymes in the small intestine of $Elp3^{WT}$ and $Elp3^{\Delta IEC}$ mice naïve or 7 days post-*N. brasiliensis* infection were quantified by Real-Time PCRs. Normalization was calculated on the average of 2 house keeping genes *Gapdh* and *36b4* (mean values \pm SEM; Student t test ; $n = 3$ and 2 for naïve $Elp3^{WT}$ and $Elp3^{\Delta IEC}$ mice; $n = 5$ and $= 4$ for infected $Elp3^{WT}$ and $Elp3^{\Delta IEC}$ mice, respectively).

Dear Dr Alain Chariot,

Thank you for submitting your revised manuscript (EMBOJ-2023-115652R) to The EMBO Journal. Your amended study was sent back to the three referees for their scientific re-evaluation, and we have received detailed comments from all of them, which I enclose below. As you will see, the experts state that the work has been substantially improved by the revisions and they are now in favour of publication.

Thus, we are pleased to inform you that your manuscript has been accepted in principle for publication in The EMBO Journal.

Please consider the remaining minor concerns of referee #1 regarding IF data validity and statistical analysis of the single-cell data carefully and amend the manuscript figures and text accordingly, where to address the remaining comments.

Also, we now need you to take care of a number of issues related to formatting and data presentation as detailed below, which should be addressed at re-submission.

Please contact me at any time if you have additional questions related to below points.

Thank you for giving us the chance to consider your manuscript for The EMBO Journal. I look forward to your final revision.

Again, please contact me at any time if you need any help or have further questions.

Best regards,

Daniel Klimmeck

>> Author Contributions: Please remove the author contributions information from the manuscript text. Note that CRediT has replaced the traditional author contributions section as of now because it offers a systematic machine-readable author contributions format that allows for more effective research assessment. and use the free text boxes beneath each contributing author's name to add specific details on the author's contribution.

>> Section order should be corrected as follows: title page with complete author information, abstract, keywords, introduction, results, discussion, materials & methods, data availability section, acknowledgements, disclosure and competing interests statement, references, main figure legends, tables, expanded figure legends

>> Please limit the abstract length to maximally 175 words.

>> Data availability section: remove referee token and make sure privacy is released from the online dataset.

>> Source data: For EV figures, please zip together all source data.

>> Source data Figure 7E - Flag-Npr12 WT/Mut shows a light box around the last two lanes. This is not present in the source data that the Authors provided. Please provide a revised main figure without the grey box area.

>> Remove the synopsis text from the manuscript.

>> Author checklist: complete the section on sample information for replicates.

>> Consider additional changes and comments from our production team as indicated below:

- Figure Legends:

1. Please define the annotated p values ****/***/**/* in the legend of figure 1a, c-g; 2a-e, g; 3b, e, h; 4e; 5d; 7c-d, g; 8b, d, f-g; 9b, d, f, h; 10c, e, g-i, k; EV 4a-b; e, g-j; as appropriate.
2. Please indicate the statistical test used for data analysis in the legends of figures 6f-g.
3. Please note that in figure EV 4j the scale bar unit should be corrected from μM to μm (in the figure file).

Referee #1:

3. Figure 3C is not convincing as an IF (staining Paneth cell granules), please validate with FISH or another approach.

The response to this comment is not satisfactory. It is not convincing that the antibody is staining stem cells in the Figure. Referring to other papers does not address the issue.

4. With regards to single-cell analysis, a major issue is that there are no statistical considerations regarding differences. The authors show a heatmap of signatures and claims increases or decreases, and a quantitative analysis would help strengthen this figure because the authors call attention to the differences in immature tuft and tuft cell populations between the *elp3* deficient and the WT mice. Regarding differences in glycolytic and ox phos scores. There are no significant differences in the data shown to make those conclusions (even the Western Blots are not convincing). In fig. 4f authors claimed HK1 and Aldoa are not properly induced, but the western blots look very similar for both WT and *Elp3* deficient. Western blot quantification should be used to demonstrate significance due to there not being a profound difference visually.

Regarding the response to this comment, the authors should really explain their statistics on scRNA-seq analysis better. For example, claiming $n=3884$ and $n=4483$ in their scRNA-seq is not helpful as those are not really independent replicates. Thus, doing statistical testing using a cell as a replicate is a flawed method. However, other parts of the validation experiments are acceptable with $n=3$ and quantification.

Referee #2:

The authors have adequately addressed my issues and those of the other reviewers, and have improved the manuscript. I do not have additional comments.

Referee #3:

The authors have effectively addressed all the comments I raised in my original review report.

The authors have generated a comprehensive set of evidence to support their claim that *Elp3* supports tuft cell differentiation by regulating mTOR activity. Intestinal epithelial cells and metabolic pathways are both complex areas of study, thus the authors have utilized a number of complementary techniques to support their conclusions.

As previously stated in my original report, an understanding of the development and maintenance of the epithelial barrier is key to many aspects of human health and I believe that this work is an important contribution to that.

Referee #1:

3. Figure 3C is not convincing as an IF (staining Paneth cell granules), please validate with FISH or another approach.

The response to this comment is not satisfactory. It is not convincing that the antibody is staining stem cells in the Figure. Referring to other papers does not address the issue.

Our answer:

In order to provide convincing images for Reviewer 1, we took the time to redo multiple IFs using two distinct antibodies in all experimental conditions. Antibodies were from Abcam and LSBio (see methods and Table 2 for references). We first conclude that IL-13R α 1 was indeed expressed in tuft cells, as expected. Regarding IL-13R α 1 expression in the crypts, we also detected some signal in stem cells located in between Paneth cells (UEA1⁺ cells). We also detected IL-13R α 1 expression in cells located above the bottom of each crypt in all experimental conditions, i.e. where transit amplifying cells are known to be located. We would like to insist on the fact that this result was consistently obtained using two distinct antibodies. This result suggests that cells leaving the bottom of the crypt still express IL-13R α 1 at the protein level, which may differ from its mRNA profile established by Zhu and colleagues, *Nature Immunology*, 2019).

4. With regards to single-cell analysis, a major issue is that there are no statistical considerations regarding differences. The authors show a heatmap of signatures and claims increases or decreases, and a quantitative analysis would help strengthen this figure because the authors call attention to the differences in immature tuft and tuft cell populations between the *elp3* deficient and the WT mice. Regarding differences in glycolytic and ox phos scores. There are no significant differences in the data shown to make those conclusions (even the Western Blots are not convincing). In fig. 4f authors claimed HK1 and Aldoa are not properly induced, but the western blots look very similar for both WT and *Elp3* deficient. Western blot quantification should be used to demonstrate significance due to there not being a profound difference visually.

Regarding the response to this comment, the authors should really explain their statistics on scRNA-seq analysis better. For example, claiming n=3884 and n=4483 in their scRNA-seq is not helpful as those are not really independent replicates. Thus, doing statistical testing using a cell as a replicate is a flawed method. However, other parts of the validation experiments are acceptable with n=3 and quantification.

Our answer:

For our single cell RNA sequencing analysis, we used Gene Set Enrichment Analysis to identify pathways that were statistically significantly up- or downregulated in *Elp3*-deficient mice. GSEA was performed to determine gene sets (minimal size = 7 genes) showing differentially expressed genes identified between *Elp3*^{WT} and *Elp3*^{ΔIEC} mice. The statistical significant enrichment of the gene set was selected based on false discovery rate (FDR) ≤ 0.25 and the normalized enriched score (NES) was calculated for each gene set. In Figure 5A, GSEA was performed to highlight signaling pathways downregulated in tuft cells from *Elp3*^{ΔIEC} mice compared to tuft cells from *Elp3*^{WT} mice. In Figure 6A, GSEA was performed to highlight signaling pathways up- or downregulated in the different immature cell populations from *Elp3*^{ΔIEC} mice compared to *Elp3*^{WT} mice.

Heatmaps were used based on GSEA analysis in order to visualize the expression levels of genes from these signatures across cell populations. In Figures 5B and 6F showing signatures of oxidative phosphorylation and mTORC1-Atf4 respectively, only genes that were significantly up- or downregulated between *Elp3*^{ΔIEC} mice compared to *Elp3*^{WT} mice in one or more of the represented populations were selected (based on $p_val_adj < 0.05$ and $\log FC > 0.25$). Differentially expressed genes were identified using FindMarkers / FindAllMarkers functions from the Seurat package. The statistical test done was the Wilcoxon Rank Sum test. Tables resuming expression levels represented in heatmaps and corresponding p-values are provided as Supplementary information. Importantly, it is very difficult to obtain a sufficient amount of tuft cells (they are rare cells in the intestinal epithelium), which impacts the statistical strength of our observations, as indicated by the adjusted p-value in our comparisons.

Expression-UMAPs were used as a visualization tool for expression levels of gene set signatures across cell populations. No statistical information would be directly associated with these plots but the represented gene sets were previously selected as described using FindMarkers / FindAllMarkers functions from the Seurat package.

To define signature of genes preferentially expressed by immature tuft cells, FindMarkers / FindAllMarkers functions from the Seurat package was used to compare immature tuft cell population to other populations. Tables resuming expression levels represented in this dotplot and corresponding p-values are provided as Supplementary information. To compare both conditions between them, we used the default methodology provided by Seurat package for differential expression (DE) analysis. These tests treat each cell as an independent replicate and ignore inherent correlations between cells originating from the same sample. We recognize the p-values obtained from this analysis should be interpreted with caution, as also suggested by the Seurat package developers. A detailed table of cell distribution through cell types, genotypes and samples is provided as Supplementary table 1. This table shows that for some comparison, the amount of cells in each condition is already a limitation to statistical power, as reflected by the high p-value or adjusted p-value in some of the DE tables provided. Nevertheless, the results of our single cell RNA sequencing experiment are starting points for hypothesis and were complemented with specific validation experiments (Real-Time PCR analyses and western blots) (see Figure 4E for example in which the definition of a specific transcriptomic signature of immature tuft cells was confirmed through Real-Time PCR analyses).

Referees #2 and #3:

We thank both reviewers for their positive comments on our study.

Dear Dr Alain Chariot,

Thank you for submitting the revised version of your manuscript. I have now evaluated your amended manuscript and concluded that the remaining minor concerns have been sufficiently addressed.

I am thus pleased to inform you that your manuscript has been accepted for publication in the EMBO Journal.

On a different note, I would like to alert you that EMBO Press offers a format for a video-synopsis of work published with us, which essentially is a short, author-generated film explaining the core findings in hand drawings, and, as we believe, can be very useful to increase visibility of the work. Please see the following link for representative examples and their integration into the article web page:

<https://www.embopress.org/doi/full/10.15252/emj.2019103932>

Best regards,

Daniel Klimmeck

Daniel Klimmeck, PhD
Senior Editor
The EMBO Journal
EMBO
Postfach 1022-40
Meyershofstrasse 1
D-69117 Heidelberg
contact@embojournal.org
Submit at: <http://emboj.msubmit.net>
